

# Validation of Methane and Carbon Monoxide from Sentinel-5 Precursor using TCCON and NDACC-IRWG stations

Mahesh Kumar Sha[1], Bavo Langerock[1], Jean-François L. Blavier[2], Thomas Blumenstock[3], Tobias Borsdorff[4], Matthias Buschmann[5], Angelika Dehn[6], Martine De Mazière[1], Nicholas M. Deutscher[7], Dietrich G. Feist[8,9,10], Omaira E. García[11], David W. T. Griffith[7], Michel Grutter[12], James W. Hannigan[13], Frank Hase[3], Pauli Heikkinen[14], Christian Hermans[1], Laura T. Iraci[15], Pascal Jeseck[16], Nicholas Jones[7], Rigel Kivi[14], Nicolas Kumps[1], Jochen Landgraf[4], Alba Lorente[4], Emmanuel Mahieu[17], Maria V. Makarova[18], Johan Mellqvist[19], Jean-Marc Metzger[20], Isamu Morino[21], Tomoo Nagahama[22], Justus Notholt[5], Hirofumi Ohyama[21], Ivan Ortega[13], Mathias Palm[5], Christof Petri[5], David F. Pollard[23], Markus Rettinger[24], John Robinson[23], Sébastien Roche[25], Coleen M. Roehl[26], Amelie N. Röhling[3], Constantina Rousogenous[27], Matthias Schneider[3], Kei Shiomi[28], Dan Smale[23], Wolfgang Stremme[12], Kimberly Strong[25], Ralf Sussmann[24], Yao Té[16], Osamu Uchino[21], Voltaire A. Velazco[7], Mihalis Vrekoussis[27,5], Pucai Wang[29,30], Thorsten Warneke[5], Tyler Wizenberg[25], Debra Wunch[25], Shoma Yamanouchi[25], Yang Yang[31,29,1], and Minqiang Zhou[1]

[1]Royal Belgian Institute for Space Aeronomy (BIRA-IASB), Brussels, Belgium
[2]Jet Propulsion Laboratory, California Institute of Technology, Pasadena, California, USA
[3]Karlsruhe Institute of Technology, IMK-ASF, Karlsruhe, Germany
[4]SRON Netherlands Institute for Space Research, Utrecht, the Netherlands
[5]Institute of Environmental Physics, University of Bremen, Bremen, Germany
[6]European Space Agency, ESA/ESRIN
[7]Centre for Atmospheric Chemistry, School of Earth, Atmospheric and Life Sciences, University of Wollongong, Wollongong, Australia
[8]Ludwig-Maximilians-Universität München, Lehrstuhl für Physik der Atmosphäre, Munich, Germany
[9]Deutsches Zentrum für Luft- und Raumfahrt, Institut für Physik der Atmosphäre, Oberpfaffenhofen, Germany
[10]Max Planck Institute for Biogeochemistry, Jena, Germany
[11]Izaña Atmospheric Research Centre (IARC), State Meteorological Agency of Spain (AEMET), Santa Cruz de Tenerife, Spain
[12]Centro de Ciencias de la Atmósfera, Universidad Nacional Autonoma de Mexico, UNAM, Mexico
[13]National Center for Atmospheric Research, Boulder, CO, USA
[14]Finnish Meteorological Institute, FMI, Sodankylä, Finland
[15]NASA Ames Research Center, Moffett Field, CA, USA
[16]LERMA-IPSL, Sorbonne Université, CNRS, Observatoire de Paris, PSL Université, Paris, France
[17]Institut d'Astrophysique et de Géophysique, Université de Liège, Liège, Belgium
[18]Department of Atmospheric Physics, Faculty of Physics, St. Petersburg State University, Saint Petersburg, Russia
[19]Earth and Space Sciences, Chalmers University of Technology, Gothenburg, Sweden
[20]UAR 3365 – OSU Réunion, Université de La Réunion, Saint-Denis, Réunion, France
[21]National Institute for Environmental Studies (NIES), Tsukuba, Japan
[22]Institute for Space-Earth Environmental Research (ISEE), Nagoya University, Japan
[23]National Institute of Water and Atmospheric Research Ltd (NIWA), Lauder, New Zealand
[24]Karlsruhe Institute of Technology, IMK-IFU, Garmisch-Partenkirchen, Germany
[25]Department of Physics, University of Toronto, Toronto, Ontario, Canada
[26]Division of Geological and Planetary Sciences, California Institute of Technology, Pasadena, CA, USA





[27]Climate and Atmosphere Research Center (CARE-C), The Cyprus Institute, Nicosia, Cyprus
[28]Earth Observation Research Center (EORC), Japan Aerospace Exploration Agency (JAXA), Japan
[29]LAGEO, the Institute of Atmospheric Physics, Chinese Academy of Sciences, Beijing, China
[30]University of Chinese Academy of Sciences, Beijing, China
[31]Shanghai Ecological Forecasting and Remote Sensing Center, Shanghai, China

**Correspondence:** Mahesh Kumar Sha (mahesh.sha@aeronomie.be)

**Abstract.** The Sentinel-5 Precursor (S5P) mission with the TROPOspheric Monitoring Instrument (TROPOMI) onboard has been measuring solar radiation backscattered by the Earth's atmosphere and its surface since its launch on 13 October 2017. Methane ($CH_4$) and carbon monoxide (CO) data with a spatial resolution (initially $7 \times 7$ km$^2$, upgraded to $5.5 \times 7$ km$^2$ on $6^{th}$ of August 2019) have been retrieved from shortwave infrared (SWIR) and near-infrared (NIR) measurements since the
end of November 2017 and made available to the experts for early validation and quality checks before the official product release. In this paper, we present for the first time the S5P $CH_4$ and CO validation results (covering a period from November 2017 to September 2020) using global Total Carbon Column Observing Network (TCCON) and Infrared Working Group of the Network for the Detection of Atmospheric Composition Change (NDACC-IRWG) network data, accounting for a priori alignment and smoothing uncertainties in the validation, and testing the sensitivity of validation results towards the application
of advanced co-location criteria.

    We found that the required bias (systematic error) of 1.5 % and random error of 1 % for the S5P standard and bias-corrected methane data are met for measurements over land surfaces with pixels having quality assurance (QA) value >0.5. The systematic difference between the S5P standard $XCH_4$ and TCCON data is on average -0.69±0.73 %. The systematic difference changes to a value of -0.25±0.57 % for the S5P bias-corrected $XCH_4$ data. We found a correlation of above 0.6 for most sta-
tions, which is mostly dominated by the seasonal cycle. The contributions of smoothing uncertainty at the individual stations are estimated and found to be dependent on the location. The highest contribution of the smoothing uncertainty is observed for mid-latitude TCCON stations and high latitude stations for NDACC. A seasonal dependency of the relative bias is seen. We observe a high bias during the springtime measurements at high SZA and a decreasing bias with increasing SZA for the rest of the year.

We found that the required bias (systematic error) of 15 % and random error of <10 % for the S5P carbon monoxide data are met in general for measurements over all surfaces with pixels having quality assurance value of >0.5. There are a few stations where this is not the case, mostly due to co-location mismatches and the limited availability of co-located data. We compared the S5P XCO data with respect to standard TCCON XCO and unscaled TCCON XCO (without application of the empirical scaling factor) data sets. The systematic difference between the S5P XCO and the TCCON data is on average
9.14±3.33 % (standard TCCON XCO data) and 2.36±3.22 % (unscaled TCCON XCO data). We found that the systematic difference between the S5P CO column and NDACC CO column data (excluding two stations that were obvious outliers) is on average 6.44±3.79 %. We found a correlation of above 0.9 for most TCCON and NDACC stations indicating that the temporal variations in CO column captured by the ground-based instruments are reproduced very similarly by the S5P CO column. The contribution of smoothing uncertainty at the individual stations is estimated and found to be significant. They are found to be



dependent on the location with large changes seen for stations located in the Southern Hemisphere as compared to the Northern Hemisphere and at highly polluted stations. A cone co-location criterion, which gives a better match between the ground-based instrument's line-of-sight and satellite pixels, seems to give better results for high latitude stations and stations located close to emission sources. The validation results for the clear-sky and cloud cases of S5P pixels are comparable to the validation results including all pixels with quality assurance value of >0.5. We observe that the relative bias increases with increasing SZA. We

estimated this increase is about 10 % over the complete range of measurement SZAs.

  The study shows the high quality of S5P $CH_4$ and CO data by validating the products against reference global TCCON and NDACC stations covering a wide range of latitudinal bands, atmospheric conditions, and surface conditions.

## 1 Introduction

The Sentinel-5 Precursor (S5P) mission with the TROPOspheric Monitoring Instrument (TROPOMI) onboard was launched

on 13 October 2017. The S5P is orbiting in a Sun-synchronous polar orbit with an equator crossing at 13:30 local solar time. The TROPOMI instrument is a nadir-viewing hyperspectral spectrometer measuring solar radiation reflected by the Earth's atmosphere and its surface from the ultraviolet-visible (270–495 nm), near-infrared (675–775 nm) and shortwave-infrared (2305-2385 nm) with daily global coverage for monitoring atmospheric trace gases and aerosol (Veefkind et al., 2012). Methane ($CH_4$) and carbon monoxide (CO) are retrieved from shortwave-infrared (SWIR) and near-infrared (NIR) measurements.

Methane is the second most important anthropogenic greenhouse gas (GHG) after carbon dioxide ($CO_2$). It has a global warming potential of about 28 times larger than $CO_2$ over a 100 year time period. It is less abundant in the atmosphere and has a significantly shorter lifetime than $CO_2$ (Stocker et al., 2013). Reduction in $CH_4$ will affect the Earth's radiation budget on a short time scale. $CH_4$ is also relevant in atmospheric chemistry, where it reacts with hydroxyl radicals (OH), thereby reducing the oxidation capacity of the atmosphere and producing ozone (Kirschke et al., 2013).

Carbon monoxide is a poisonous reactive gas considered principally an anthropogenic atmospheric pollutant. Volatile organic compounds (VOCs) are emitted to the atmosphere by incomplete combustion (e.g., vehicles, industry and biomass burning) and have an important role in the production of CO. The lifetime of CO is relatively short and ranges from weeks to months (Novelli et al., 1998). CO reacts with atmospheric oxidants, ozone ($O_3$), hydroperoxy ($HO_2$), and hydroxyl radicals (OH). It is the largest direct sink of OH affecting the self-cleansing capacity of the atmosphere. An increase in CO would imply a higher

OH loss through chemical reaction and therefore less availability of OH for the depletion of other atmospheric constituents such as $CH_4$. CO is therefore affecting the concentrations of primary greenhouse gases and has an indirect but important influence in determining the chemical composition and radiative properties of the atmosphere. It is therefore considered as an indirect greenhouse gas (Stocker et al., 2013).

  Continuous precise and accurate global measurements of these gases are very important for long-term monitoring and

60 their use by the inverse models such that the inferred surface fluxes can be better constrained. This paper focuses on the quality assessment of the operational S5P $CH_4$ and CO products by performing validation of the total columns of these two products with the reference data from all stations in the ground-based Total Carbon Column Observing Network (TCCON) and





Infrared Working Group of the Network for the Detection of Atmospheric Composition Change (NDACC-IRWG) networks. The systematic and random error requirements of the $CH_4$ and CO products are checked based on 2.8 years of S5P data and possible reasons are given where large deviations are observed.

The paper is organised as follows: Section 2 describes the satellite and ground-based reference data used in this study. Section 3 gives the details of the validation methodology. Section 4 gives the validation results for $CH_4$ and section 5 gives the validation results for CO. Section 6 summarizes our results and conclusions.

## 2 Data

In this section, we present an overview of the input data from the S5P and the reference ground-based data from the TCCON and NDACC-IRWG, herewith referred to as NDACC, which are used for the validation of the S5P operational $CH_4$ and CO products.

### 2.1 S5P Methane and Carbon monoxide data sets

TROPOMI is the unique payload of the ESA/Copernicus Sentinel-5 Precursor mission orbiting in a low-Earth Sun-synchronous polar orbit with a wide swath of 2600 km across track resulting in daily global coverage. The TROPOMI radiometric measurements of the Earth's radiance and solar irradiance are processed using on-ground data processor to retrieve the atmospheric abundances of ozone ($O_3$), nitrogen dioxide ($NO_2$), sulphur dioxide ($SO_2$), formaldehyde (HCHO), methane ($CH_4$), carbon monoxide (CO), as well as cloud and aerosol properties. The spatial resolution of the operational Level 2 (L2) $CH_4$ and CO products was originally $7 \times 7$ km$^2$ and was increased to $5.5 \times 7$ km$^2$ on 6th of August 2019.

The operational processing to retrieve the total column-averaged dry air mole fraction of methane ($XCH_4$) is performed by the RemoTeC-S5P algorithm. The details describing the theoretical baseline of the algorithm, the input and ancillary data needed, averaging kernel, and the output generated are described in detail in Hu et al. (2016) and Hasekamp et al. (2019). The use of satellite measurements for estimating sources and sinks of $CH_4$ strongly depends on the precision and accuracy achieved. Systematic biases or lower precision on regional or seasonal scales can jeopardise the usefulness of the satellite measurements for the estimation of source and sink estimates (Bergamaschi et al., 2007). The bias requirement for S5P $XCH_4$ is 1.5 % and the random error requirement is 1 % (as reported in the official ESA document ESA-EOPG-CSCOP-PL, 2017, Table 1, page 14). The current S5P $CH_4$ data are only processed for cloud-free measurements over land. Along with the standard $CH_4$ product, a bias-corrected $CH_4$ product is also made operationally available. We provide a brief summary of the $CH_4$ bias correction here and the details of the bias correction can be found in section 5.6 of the Algorithm Theoretical Baseline Document (ATBD) for S5P methane retrieval (Hasekamp et al., 2019). The operational S5P $CH_4$ product has been compared to co-located GOSAT proxy measurements. The S5P-GOSAT $XCH_4$ ratio shows a high correlation to the retrieved surface albedo in the SWIR. The highest correlation is for low surface albedo scenes. A posteriori bias correction has been applied to the S5P $CH_4$ product using a second-order polynomial fit. The effect of the bias correction is an increase of the retrieved $CH_4$ for scenes with relatively low albedo conditions (e.g., forest scenes) and a decrease of $CH_4$ for scenes with high albedo conditions (e.g., desert scenes). In the





paper, we will show the validation results of both standard and bias-corrected S5P CH$_4$ products. The latest product versions of S5P CH$_4$ data for the reprocessed (RPRO) + offline (OFFL) data from the start of the mission to 30 September 2020 are used in this work. The version numbers and the respective dates are listed in Table 1 and further details on the relevant improvements are given in the Product Readme File (PRF; https://sentinel.esa.int/documents/247904/3541451/Sentinel-5P-Methane-Product-Readme-File, last access 14 July 2020). The quality assurance (QA) value is provided as part of the CH$_4$ data product. QA

>0.5 is used as recommended by the PRF to filter out the S5P CH$_4$ data to be used for the validation studies. This selection filters out measurements performed with surface albedo <0.02, solar zenith angle (SZA) >70°, viewing zenith angle >60°, and some other criteria as mentioned in the PRF.

The operational processing to retrieve the total column density of carbon monoxide (CO) simultaneously with interfering trace gases and effective cloud parameters (cloud height and optical thickness) is performed by the Shortwave Infrared Carbon

Monoxide Retrieval (SICOR) algorithm (Landgraf et al., 2016). The details describing the theoretical baseline of the algorithm, the input and ancillary data needed, example plots of averaging kernel, and the output generated are described in details in Landgraf et al. (2018). The bias requirement for total column-averaged dry air mole fraction of carbon monoxide (XCO) is 15 % and the random error requirement is <10 % (as reported in the official ESA document ESA-EOPG-CSCOP-PL, 2017, Table 1, page 14). The CO total column L2 data products are available as the Offline (OFFL) and Near Real Time (NRTI)

timeliness data products. The version numbers and the respective dates are listed in Table 1 and further details on the relevant improvements are given in the Product Readme File (PRF; https://sentinel.esa.int/documents/247904/3541451/Sentinel-5P-Carbon-Monoxide-Level-2-Product-Readme-File, last access 14 July 2020). The latest product versions of S5P CO data for the reprocessed (RPRO) + offline (OFFL) data from the start of the mission to 30 September 2020 are used in this work. The NRTI data stream delivers the CO data product within 3 hours after sensing, whereas the OFFL data are available a few days

after sensing. Due to the different timeliness, the NRTI product are given in 5 min data granules, whereas the OFFL data product are given per satellite orbit. The consecutive data granules of the NRTI product show an overlap of about 12 scan lines. The NRTI processing chains employ the same algorithm as the OFFL since processor version 01.03.02 starting from orbit number 8906 on 3rd of July 2019 (see section 9.4 of Lambert et al. (2020) for validation results showing the equivalence of S5P NRTI and OFFL CO products). More details on the two processing streams of the two data sets are given in the Algorithm

Theoretical Basis Document (ATBD) (Landgraf et al., 2018). In this paper, we show the detailed validation results of the S5P OFFL CO product. Data with QA values >0.5 is used as recommended by the PRF. This selection filters out measurements performed with SZA ≥80°, sensor zenith angle ≥80°, two most westward pixels due to unresolved calibration issues and some other criteria as mentioned in the PRF. Furthermore, we also separated retrievals performed for measurements under clear-sky (CLSKY; cloud optical thickness <0.5 & cloud height <500 m, over land) and cloudy conditions (CLOUD; cloud

optical thickness ≥ 0.5 & cloud height <5000 m, over land and ocean) as suggested by Borsdorff et al. (2018b). The clear-sky observations over the ocean have too low signal intensities in the SWIR and therefore cannot be used for the data interpretation. Unlike the S5P CH$_4$ a priori profiles which are available in the L2 files, the S5P a priori profiles for CO were downloaded from ftp://ftp.sron.nl/pub/jochen/TROPOMI_apriori/ (last access 01 December 2020). Among the known data quality issues of the CO product, single overpasses of S5P show stripes of erroneous CO values <5 % in the flight direction, probably due to





calibration issues of S5P. We did not do any correction of this stripe pattern as we show the operational validation of the CO
product and also because of small number of pixels <5 % are affected by it. The striping effect is analysed in detail by Borsdorff
et al. (2019). The effect on the TCCON validation was small. The destriping approach suggested by this work is planned to be
implemented by the operational TROPOMI CO processing in the near future. Furthermore, the effect of updating the spectral
cross-sections in the TROPOMI CO processing for clear-sky and cloudy conditions was analysed with ground-based FTIR

measurements from 12 stations of the TCCON network (Borsdorff et al., 2019).

## 2.2 Ground-based TCCON reference data set

The Total Carbon Column Observing Network (TCCON) represents a network of ground-based Fourier transform spectrom-
eters (FTS), of the type Bruker IFS 125HR (some long-existing sites also use Bruker 120/5HR), that records direct solar
absorption spectra in the near-infrared (NIR) spectral range to retrieve accurate and precise column-averaged abundances of

atmospheric constituents including $CO_2$, $CH_4$ and CO amongst other species (Wunch et al., 2011, 2015). It is the current
state-of-the-art validation system for total column measurements of important greenhouse gases (GHGs) by remote sensing.
TCCON data from several stations have been used in previous studies for the validation of trace gas data products from satellite
platforms such as OCO-2 (O'Dell et al., 2018; Wunch et al., 2016), GOSAT (Iwasaki et al., 2017; Kulawik et al., 2016), S5P
(Sha et al., 2018a; Borsdorff et al., 2018a, 2019), MOPITT (Hedelius et al., 2019), SCIAMACHY (Borsdorff et al., 2016;

Hochstaffl et al., 2018). Data from all stations (23 in the Northern Hemisphere and 5 in the Southern Hemisphere) are used
in this study and are listed in Table 2. The stations cover various atmospheric conditions (humid, dry, polluted, presence of
aerosol), various surface conditions (range of albedo, flat terrain, high altitude locations), latitudinal distribution from 80° N to
45° S. The stations at Nicosia and Xianghe are not yet officially part of TCCON but performs observations and data analysis
fully compatible with TCCON guidelines. GGG2014 (the current standard TCCON retrieval code) $XCH_4$ systematic errors for

TCCON are below 0.5 % for SZAs below 85°. The XCO errors are below 4 % and decrease with SZA (Wunch et al., 2015). The
uncertainty in the scaling slope for XCO is 6 % ($2\sigma$) (Hedelius et al., 2019). Previous studies have shown that the scaling factor
of ∼7 % used in GGG2014 to tie the TCCON XCO measurements to the World Meteorological Organization (WMO) in situ
scale is large compared to the current uncertainty in spectroscopy (Sha et al., 2018b; Hedelius et al., 2019; Zhou et al., 2019).
A scaling factor of 7 % provided the best scaling to the in situ data available when the scaling for GGG2014 was calculated.

There is currently an ongoing effort within the TCCON community to determine whether the scaling factor is appropriate.
These results are very important to decide on the choice of spectroscopic cross-sections that should be implemented for the
future improved S5P CO product (Borsdorff et al., 2019). In this work, we use the official TCCON XCO product as well as
an XCO product without the application of the empirical scaling factor, herewith referred to as unscaled XCO. The validation
work is done using the standard and rapid delivery of TCCON data from the whole network. The publicly available TCCON

data can be accessed via https://tccondata.org/.





## 2.3    Ground-based NDACC-IRWG reference data set

The Infrared Working Group (IRWG) of the Network for the Detection of Atmospheric Composition Change (NDACC) represents a network of high-resolution Fourier transform spectrometers that records solar absorption spectra in the mid-infrared (MIR) spectral range. It is a multi-national collection of over twenty stations distributed from pole to pole (Eureka 80° N to Arrival Heights 77.8° S). The solar absorption spectra are used to retrieve the atmospheric concentrations of a number of gaseous atmospheric components, including ozone ($O_3$), nitric acid ($HNO_3$), hydrogen chloride ($HCl$), hydrogen fluoride ($HF$), carbon monoxide ($CO$), nitrous oxide ($N_2O$), methane ($CH_4$), hydrogen cyanide ($HCN$), ethane ($C_2H_6$) and chlorine nitrate ($ClONO_2$) (https://www2.acom.ucar.edu/irwg). NDACC $CH_4$ and CO data from several stations have been used in previous studies for satellites validation (Borsdorff et al., 2020; Hedelius et al., 2019; Hochstaffl et al., 2018; Sha et al., 2018b; Buchholz et al., 2017; Olsen et al., 2017). In this study, data from all stations (19 in the Northern Hemisphere and 4 in the Southern Hemisphere) are used and are listed in Table 3. Several of the stations are located in high latitude regions and many stations are located at high altitudes to reduce the interference of water vapour in the measurements. Some of these stations (e.g., Karlsruhe, Garmisch, Sodankylä) are not officially part of NDACC but performs observations and data analysis fully compatible with NDACC guidelines. The co-located NDACC and TCCON stations often share one FTIR instrument, applying the respective detector and filter settings. The spectra are analysed either with the SFIT4 algorithm, an evolution of SFIT2 (Pougatchev et al., 1995) or with the PROFFIT9 algorithm (Hase et al., 2004) to retrieve vertical profiles of $CH_4$ and CO. The retrieval allows the derivation of a tropospheric and a stratospheric column of the target gases (Sepúlveda et al., 2012, 2014). The NDACC CO column values can be used directly to validate the S5P CO column values. However, for the S5P $XCH_4$ validation, the NDACC $XCH_4$ values need to be calculated. Due to the NDACC measurements being performed in the MIR range, the oxygen ($O_2$) total column is not available from the spectrum for calculating the column-averaged dry air mole fractions of the target gas (Xgas), similar to what is done for TCCON (see Eq. A9 of Wunch et al. (2011)). Therefore, the total column of dry air is computed as described in Eq. 1 of Deutscher et al. (2010). The surface pressure ($P_s$) is recorded at the local weather station of the FTS stations and $H_2O$ total column ($TC_{H_2O}$) is derived from the National Centers for Environmental Prediction (NCEP) reanalysis data set. In case if there is no surface pressure available, then we extrapolate the pressure grid to the surface. The $XCH_4$ calculated values for NDACC measurements are then used for the validation of the S5P $XCH_4$ data. Unlike TCCON data, where a species specific scaling factor is applied to tie the measurements to the WMO in-situ scale, the NDACC data do not apply any scaling of the retrieved results. The typical accuracy and precision of the NDACC $CH_4$ data is about 3 % and 1.5 %, respectively. The typical accuracy and precision of the NDACC CO data is about 3 % and 1 %, respectively. High systematic uncertainty is mainly due to the too conservative spectroscopic uncertainty component. Both the consolidated data available via http://www.ndacc.org and the rapid delivery data supported by the CAMS27 project (https://cams27.aeronomie.be/) have been used in this study.



## 3 Validation methodology

S5P provides XCH$_4$ values but only column CO values and therefore XCO is calculated by taking the ratio of the total column of CO (TC$_{CO}$) divided by the total column of the dry air (TC$_{dry,air}$) (following Eq. 1 in Deutscher et al. (2010)) to compare to the TCCON XCH$_4$ and XCO data.

$$195 \quad XCO = \frac{TC_{CO}}{TC_{dry,air}} = \frac{TC_{CO}}{P_S/(g \times m_{dry,air}) - TC_{H_2O} \times (m_{H_2O}/m_{dry,air})}, \tag{1}$$

where P$_S$ is the surface pressure, TC$_{H_2O}$ is the total column of H$_2$O, g is the column-averaged acceleration due to gravity, m$_{dry,air}$, and m$_{H_2O}$ are the molecular masses of dry air and H$_2$O, respectively. P$_S$ and TC$_{H_2O}$ are taken from the S5P files.

The validation of the S5P methane and carbon monoxide data is performed based on the reference data sets from the ground-based TCCON and NDACC networks. We present the results for both the networks with different co-location criteria applied to the data sets. The differences in the validation results are also based on whether or not a common prior has been used for the satellite and ground-based Fourier transform infrared (FTIR) data sets, details are discussed in Appendix A.

S5P provides daily global coverage with a huge data set having a wide swath at a high spatial resolution for every overpass. Therefore, the selection of good co-location criteria is a crucial task in finding the best strict criteria while ensuring sufficient co-located data for a statistically significant validation. We tried several co-location criteria to test the sensitivity of the method in relation to the choice of the parameter (e.g., time, distance, line-of-sight, ...). The best co-location criteria will be such that the bias is robust and not sensitive to small changes in the co-location criteria. In the next sections, the results of the application of these criteria are shown for the case with the reduction of smoothing uncertainty and in relation to direct comparisons.

## 4 Validation of S5P methane products

The validation of the S5P methane products with the ground-based FTIR data is discussed in this section. The TCCON stations cover a wide range of varying ground conditions and topography. The high latitude stations (e.g., Eureka, Ny-Ålesund, Sodankylä, East Trout Lake, ...) challenge the satellite algorithm for measurements at very high SZAs, high air masses and scenes with snow or ice coverage. The Edward site is adjacent to a very bright playa. The Park Falls and Lamont stations have relatively uniform surface properties but the ground cover can vary seasonally. The TCCON stations at Izaña and Zugspitze are located at high altitude. Izaña along with Ascension, Réunion, and Burgos are located on small islands, remote from large landmasses but with significant topography. Several stations are located near or in urban regions with a large population (e.g., Pasadena, Paris, Tsukuba, ...). The Darwin site has the ocean to the north. The Wollongong site has the ocean on one side and sharp escarpment on the other. The Lauder site is surrounded by hills. Nicosia is a new site, operational since August 2019, using a FTIR which was moved from the Białystok TCCON station after its closure in October 2018. The TCCON observatory at Nicosia has been calibrated by vertical aircraft profiling at its former location Białystok, but not at its current location. Xianghe site in China, located in a heavily populated region, is a new site, which is operated following the recommendations of TCCON, but is not yet affiliated as a TCCON station. The NDACC stations are often located at high altitude (e.g., Altzomoni, Jungfraujoch, Mauna Loa, Zugspitze, Izaña, Maïdo, Boulder). Several of the NDACC stations are located at high latitudes





(e.g., Eureka, Ny-Ålesund, Thule, Kiruna, Sodankylä). Several of the NDACC stations are located near or in urban areas (e.g., Bremen, St.Petersburg, Toronto, Boulder, Altzomoni - close to Mexico City). The NDACC station at Arrival Heights is the
225 only site on the Antarctic continent. TCCON provides dry-air column-averaged mole fractions of methane similar to the S5P product, whereas NDACC provides concentration profiles of methane with sensitivities up to about 20 km. As the characteristics of the two reference ground-based data sets are different, two slightly different comparison methods were applied for the validation study which are discussed in this section.

### 4.1 Validation of S5P bias-corrected vs. standard methane data using TCCON and NDACC data sets

The validation results of the S5P bias-corrected and standard methane products with reference TCCON and NDACC data are discussed in this section. The S5P observations co-located with the ground-based reference measurements are found by selecting all filtered S5P pixels within a radius of 100 km around each site and with a maximal time difference of 1 h for TCCON and 3 h for NDACC observations. The 1 h time difference for TCCON can be justified by noting that TCCON instruments acquire only one type of spectra and from each good spectrum methane is retrieved, while NDACC instruments
are required to measure different types of spectra with different optical filter configurations, making the number of methane observations more sparse. An effective location of the FTIR measurement on the line-of-sight (i.e. at a 5 km altitude) is used to do the co-location. The co-located pixels can therefore differ from measurement to measurement. For each of the ground-based measurements which are co-located with the S5P measurements, an average of all S5P pixels is done. Co-located pairs are created between ground-based and averaged S5P only if a minimum of five pixels is found in applying the coincidence
criteria. In the comparison, the a priori in the TCCON and NDACC retrievals have been substituted with the S5P methane a priori following Eq. A1. The a priori alignment is done to compensate/correct its contribution to the smoothing equation (Rodgers and Connor, 2003). The TCCON results with the S5P prior substituted are then compared directly to the S5P $XCH_4$ data. However, the NDACC $CH_4$ concentration profile with the S5P prior substituted is additionally smoothed with the S5P column averaging kernel following Eq. A2. The NDACC $XCH_4$ is derived as discussed in section 2.3 and then compared to
the S5P $XCH_4$ data. Furthermore, each validation run also includes the adaptation of the S5P columns to the altitude of the ground-based FTIR instruments for cases where satellite averaging kernel is not applied or when column boundaries may differ (see Appendix B for details).

Table 4 provides the validation results for the S5P bias-corrected and standard $XCH_4$ data with the a priori aligned TCCON data at each TCCON station. The systematic difference (the mean of all relative differences) between the S5P and TCCON
data is on average -0.69±0.73 % (S5P standard $XCH_4$ product) and -0.25±0.57 % (S5P bias-corrected $XCH_4$ product). This is well within the mission requirements for a bias of 1.5 %. Only at a few TCCON stations (Sodankylä, East Trout Lake, Park Falls and Wollongong) the bias is slightly higher than 1.5 % for the S5P standard $XCH_4$ product. However, it never exceeds the mission requirements for the bias-corrected product. The standard deviation of the relative bias, which is a measure of the random error, is well below the mission requirement of 1 % for both standard (0.59±0.17 %) and bias-corrected (0.57±0.18 %)
S5P $XCH_4$ products.





Figure 1 shows the bar plots for the S5P XCH$_4$ relative bias with respect to the TCCON XCH$_4$ data at all stations (left panel) and the standard deviation of the relative bias (right panel). The comparisons relative to the S5P bias-corrected XCH$_4$ product (labeled – bcsm100k1hr) are the blue bars and those for the standard XCH$_4$ product (labeled – stdsm100k1hr) are the magenta bars. The bias-correction of the S5P XCH$_4$ product being a function of the surface albedo acts differently at the different

TCCON stations. Figure 2 shows the relative difference of the bias for the standard (top panel) and bias-corrected (bottom panel) S5P XCH$_4$ products as a function of the retrieved S5P SWIR surface albedo at the TCCON stations. The bias correction of the S5P XCH$_4$ product brings the high negative relative differences closer to zero for low surface albedo conditions and the high positive relative differences closer to zero for high surface albedo conditions. The low surface albedo conditions also show a high scatter in the relative difference plots. The latter is mainly because the scenes with low surface albedo are

challenging for satellite retrieved products due to large measurement noise. The difference of the relative bias between the S5P bias-corrected and the standard XCH$_4$ product for each TCCON station is shown as magenta bar in the middle panel plot (labeled – diff_bcvsstd) of Fig. 1. It shows the overall direction of change is positive for most stations (low surface albedo conditions) and negative for few stations like Edwards, JPL, Pasadena (high surface albedo conditions). The standard deviation of the relative bias for the S5P standard and bias-corrected XCH$_4$ products are comparable. Scenes with low and high albedos

pose specific challenges for S5P CH$_4$ retrieval. Validation of S5P CH$_4$ data at additional sites with different conditions (e.g. high surface albedo, high humidity, regions not covered by TCCON and NDACC) using portable FTIR spectrometers (Sha et al., 2020) will give further insight into the S5P CH$_4$ product quality.

The relative biases are plotted as mosaic plots and shown in Fig. 3, where the top panel shows the bias for S5P standard XCH$_4$ product while the bottom panel shows the bias for S5P bias-corrected XCH$_4$ product relative to TCCON. Each bar in

the mosaic plots represents the weekly averages of the relative bias values. The high latitude stations show a high positive bias during the spring, which is then reduced and even switched sign to show negative bias during the autumn. Lorente et al. (2021) also found similar seasonality in the bias at the high latitude sites of Sodankylä and East Trout Lake and indicated correlations of high bias during spring time with the presence of snow (low surface albedo in the SWIR but high surface albedo in the NIR). In addition, the high latitude sites are also influenced by the polar vortex, which is difficult to be represented by the a priori. The

difference of the a priori from the true atmospheric profile will also add to the bias. This will be discussed further in the next section. Since measurements rely on direct line-of-sight of the sun, data are not available during the winter months for high latitude stations. The time series of the S5P bias-corrected XCH$_4$ product and TCCON data for each site are shown in Figs. 4 and 5. The ground-based TCCON XCH$_4$ data are represented in grey and the S5P data during that period is shown in light blue. The S5P data co-located with TCCON data are shown in blue and the co-located TCCON data with a priori alignment are

shown in black. The amplitude of the CH$_4$ seasonal cycle is different at the different sites. This is related to the variability of the CH$_4$ concentrations in the atmosphere. The CH$_4$ concentration profile decreases rapidly with increasing altitude above the tropopause height. The concentration of CH$_4$ in the stratosphere, along with the troposphere, plays a key role in determining the total column of CH$_4$ at the given location. The CH$_4$ seasonal cycle in the troposphere is driven by the seasonality of both, CH$_4$ sources and its sinks (mainly due to the reaction with OH), while the CH$_4$ seasonal cycle in the stratosphere is dominated

by the vertical transport (Sepúlveda et al., 2012; Ostler et al., 2014; Bader et al., 2017; Zhou et al., 2018). The time series of





the relative bias plots shown in Figs. 6 and 7 indicate a seasonal cycle, which is clearly seen for stations with a high density of reference data with a low scatter e.g., Park Falls, East Trout Lake, Lamont, Edwards, Pasadena.

Taylor diagrams for the S5P XCH$_4$ and TCCON XCH$_4$ for the standard (top panel) and bias-corrected (bottom panel) S5P XCH$_4$ products are shown in Fig. 8. The correlation, represented by the angular coordinate of the S5P bias-corrected XCH$_4$
product improved in comparison to the S5P standard XCH$_4$ product. Most stations have a correlation above 0.6 (see Table 4 for exact values), and the distance to the origin of the ground-based dot relative to the satellite dot (ratio of std of ground-based/std of S5P) is below 1 for most stations implying that the satellite data are more variable than the ground-based data. The correlation is mostly dominated by the seasonal cycle and low correlations are seen for high latitude sites where a bias jump is seen between spring and summer periods. Outliers such as Ny-Ålesund, JPL, and Białystok are due to the limited
data sets available for the comparison. Ny-Ålesund station is located on the shore of a bay on the west coast of the island of Spitsbergen in Svalbard, Norway. As a result only a few valid S5P XCH$_4$ pixels are found around the station resulting in limited co-located data available for comparison. The TCCON instrument from JPL and Białystok stations were moved to Edwards and Nicosia, respectively. Thus resulting in a limited data sets available from these sites. The very low correlation for Darwin and Wollongong is due to the low satellite values for some days (see Fig. 5) and for high latitude sites is due to the jump in
the bias between spring and later months (see Fig. 6). The altitude correction of the pixels works well as can be seen by the relatively good correlation for Zugspitze, however, the scatter in the data is high.

Table 5 provides the validation results for the S5P bias-corrected and standard XCH$_4$ data with the smoothed NDACC data at each NDACC station. The systematic difference (the mean of all relative differences) between the S5P and NDACC data is on average $0\pm1.12$ % (S5P standard XCH$_4$ product) and $0.64\pm0.77$ % (S5P bias-corrected XCH$_4$ product). This is well within
the mission requirements for a bias of 1.5 %. The mean of all stations is calculated by excluding outliers, which are stations with a low number of co-locations (Ny-Ålesund, Rikubetsu), high scatter in the ground-based data (Toronto), and unexpected high bias (Thule, Arrival Height). Thule is located on the western coastline of Greenland. The valid S5P XCH4 pixels within the co-location radius around Thule show several pixels with high XCH$_4$ values. These high XCH$_4$ values are in general found along the coastline and regions with altitude variability. Although a filter for the variability of the terrain roughness is applied in
the QA filter options, these high values along the coastline of Greenland need detailed investigation and possible optimisation of the filter settings to remove the unexpected high values. We also observe valid pixels with unexpected high XCH$_4$ around the coastline and terrains with altitude variability in Antarctica. This is also the reason for the high bias observed at the Arrival Heights station located along the west side of Hut Point Peninsula in Ross Island, Antarctica. The bias at Altzomoni is worse than the requirement, while the random error is better than the requirement of 1 %. Bezanilla et al. (2014) found large variability
in CH$_4$ total columns measured at Mexico City Basin, pointing to significant local emissions affecting the natural background levels. A co-location mismatch would contribute partly to the bias seen with respect to S5P (see section 4.3 on how using an advanced co-location criterion reduces the bias at Altzomoni). The mean standard deviation of the relative bias which is a measure of the random error is about 1 % ($1.05\pm0.55$ %) for both S5P standard and bias-corrected XCH$_4$ products. The high latitude stations in the Northern Hemisphere show values slightly higher than 1 %.





The S5P XCH$_4$ relative bias and the standard deviation of the relative bias with respect to the NDACC stations as shown in Table 5 are shown as bar plots in Fig. 9. The comparisons relative to the S5P bias-corrected XCH$_4$ product (labeled – bcsm100k1hr) are the blue bars and those for the standard XCH$_4$ product (labeled – stdsm100k3hr) are the magenta bars. The standard deviation of the relative bias (right panel) for the S5P standard and bias-corrected XCH$_4$ products are comparable. Figure 10 shows the relative difference of the bias for the S5P standard (top panel) and bias-corrected (bottom panel) XCH$_4$

products as a function of the retrieved surface albedo at the NDACC stations. Similar to the TCCON comparison, we also see here that the bias correction of the S5P XCH$_4$ product brings the high negative relative differences closer to zero for low surface albedo conditions and the high positive relative differences closer to zero for high surface albedo conditions. The data at stations with low surface albedo conditions also show a high scatter in the relative difference plots. The difference of the relative bias between the S5P bias-corrected and the standard XCH$_4$ product for each NDACC station is shown as magenta bar

in the middle plot (labeled – diff_bcvsstd) of Fig. 9. It shows the overall direction of change is positive for most stations (low surface albedo conditions) and negative for few stations like Boulder, Altzomoni (high surface albedo conditions).

The relative biases are plotted as mosaic plots and are shown in Fig. 11, where the top panel shows the bias for the S5P standard XCH$_4$ product while the bottom panel shows the bias for the S5P bias-corrected XCH$_4$ product relative to NDACC. Each bar in the mosaic plots represents the weekly averages of the relative bias values. The high latitude stations show a high

positive bias during the spring, which is then reduced and even switched sign to show negative bias during the autumn. This is the reason for the high standard deviation of the relative difference seen for the high latitude stations having measurements during the spring and summer or autumn. Since measurements rely on direct line-of-sight of the sun, the data are not available during the winter months for high latitude stations. The time series of the S5P bias-corrected XCH$_4$ product and the NDACC data for each site are shown in Figs. 12 and 13, and the respective relative bias are shown in Fig. 14 and 15. In the plots, the

NDACC data are shown in grey and the S5P data are shown in light cyan. The S5P data co-located with NDACC data are shown in cyan and the co-located NDACC data are shown in black.

Taylor diagrams for the S5P XCH$_4$ and NDACC XCH$_4$ for the standard (top panel) and bias-corrected (bottom panel) S5P XCH$_4$ products are shown in Fig. 16. The correlation, represented by the angular coordinate, of the S5P bias-corrected XCH$_4$ product improved for most sites in comparison to the S5P standard XCH$_4$ product. Most stations have a correlation above

0.5 (see Table 5 for exact values). No clear conclusion can be drawn as to if the satellite data are more variable than the ground-based NDACC data, as we find quite some stations where the distance to the origin of the ground-based dot relative to the satellite dot is both below 1 and above 1. The correlation is mostly dominated by the seasonal cycle and low correlations are seen for high latitude sites where a bias jump is seen between spring and summer periods. Outliers such as Ny-Ålesund, Rikubetsu are due to the limited data sets available for the comparison. The ground-based data set from Toronto show a high

scatter, while a high unexpected bias for Thule and Arrival Heights indicates some problem with the data set. The ground-based data set from Harestua show a high scatter for few co-locations. The low correlation for the high latitude stations Sodankylä and Kiruna is due to the jump in bias between spring and later months (see Figs. 12 and 14).





Eight ground-based stations contributed to the validation study by providing XCH$_4$ data from both TCCON and NDACC measurements performed at the sites. The differences in the relative bias of the S5P bias-corrected XCH$_4$ product with respect to the TCCON and NDACC (bias$_\text{NDACC}$–bias$_\text{TCCON}$) for these stations are the following: 0.15 % (~2.9 ppb) for Eureka, 1 % (~19 ppb) for Sodankylä, 1.62 % (~30.8 ppb) for Bremen, 0.68 % (~12.9 ppb) for Karlsruhe, 0.16 % (~3.0 ppb) for Garmisch, 0.57 % (~10.8 ppb) for Zugspitze, 0.84 % (~16.0 ppb) for Wollongong, and 0.26 % (~5.0 ppb) for Lauder. Ostler et al. (2014) in a multistation (five) intercomparison study of column-averaged methane from NDACC and TCCON showed that there is no overall bias between MIR (NDACC) and NIR (TCCON) XCH$_4$ retrievals in general. However, dynamical variability can cause NDACC-TCCON differences in the XCH$_4$ values at the sites, with values up to 30 ppb. The high latitude stations are affected by the stratospheric subsidence induced by the polar vortex, whereas for other locations, a deep stratospheric intrusion event can be the cause for the difference. Our study also shows differences between the bias$_\text{NDACC}$–bias$_\text{TCCON}$ of the same order (up to ~31 ppb) for the co-located stations. In the next section, we show detailed results of the a priori alignment and smoothing correction at the individual stations.

## 4.2 Smoothing effect in the validation of S5P methane data

The validation of the S5P bias-corrected XCH$_4$ data relative to the TCCON and NDACC XCH$_4$ data with and without (i.e., direct comparison) a priori alignment and smoothing correction are discussed in this section. S5P, TCCON and NDACC all have different vertical sensitivities and use different a priori profiles for their retrievals. In the case of similar vertical sensitivities, we can assume that the smoothing effects from satellite and ground-based retrievals are of nearly equal magnitude. However, the vertical sensitivities and the a priori used are different, which means that the a priori profiles and the averaging kernels should be taken into account. For the case of TCCON, only an a priori alignment is done. The S5P prior is used as the common prior in our validation study. Smoothing effects are most relevant for cases with strong dynamic variability in the atmosphere. TCCON performs a profile scaling retrieval on the measurements performed in the NIR spectral region, whereas NDACC performs a profile retrieval in the MIR spectral region. The altitude of perturbation of the CH$_4$ profile plays a significant role on smoothing correction and is different for NIR and MIR retrievals. Ostler et al. (2014) showed that TCCON retrievals are more accurate when perturbations are due to stratosphere–troposphere exchanges in the upper troposphere/lower stratosphere (UTLS) region, whereas NDACC retrievals are more accurate for cases of stratospheric subsidence. In order to ascertain the effect of a priori alignment and smoothing, the validation results of the direct comparison are compared against the validation results with a priori alignment and smoothing as discussed in the previous section.

The validation results of the S5P bias-corrected XCH$_4$ data relative to the TCCON and NDACC data without a priori alignment and smoothing correction (direct comparison) are shown in columns 12–15 of Tables 4 and 5, respectively. The S5P XCH$_4$ relative bias and the standard deviation of the relative bias with respect to TCCON and NDACC are shown as grey bars in the left panel and right panel plots of Figs. 1 and 9, respectively. The standard deviation of the relative bias without smoothing correction is similar to the standard deviation of the relative bias for the case with smoothing correction. The differences between the relative bias with and without smoothing correction for the S5P bias-corrected XCH$_4$ data for each TCCON and NDACC station are shown as grey bars in the middle panel plot (labeled – diff_smvsnosm) of Figs. 1 and 9,





respectively. The difference plot relative to TCCON shows that the overall direction of change is negative for all stations, with high values for most stations in the Northern Hemisphere corresponding to regions with high dynamic variability. We observe

a maximum difference of -0.25 % (~-4.8 ppb) and a mean difference of -0.14±0.07 % (~-2.7±1.3 ppb) across all TCCON sites for the duration of available measurements used in this study. The a priori alignment correction for the Southern Hemisphere sites is low where we observe on average a difference of about -0.07 % (~-1.3 ppb). The difference plot relative to NDACC shows that the overall direction of change is positive for all stations. Ny-Ålesund, which has the lowest number of collocations, shows the highest difference of 2.18 % (~41.4 ppb). Thule, which has an unexpected high bias, shows the second highest

difference of 1.86 % (~35.3 ppb), and Toronto, which has a high scatter in the ground-based data, shows a high difference of 1.05 % (~20 ppb). The difference at all other stations is below 1 %, with the high values seen for high latitude sites, the mean difference of the selected NDACC sites shown in Table 5 is 0.37±0.28 % (~7±5.3 ppb).

As pointed out in section 4.1, the difference of smoothing (only a priori alignment for TCCON) vs. no smoothing for the eight co-located stations is observed highest for mid-latitude TCCON stations and that for the NDACC stations, we observe the

highest difference for the high latitude stations. It is therefore important to use a realistic a priori profile for scaling retrievals, especially for cases of stratospheric subsidence or stratosphere–troposphere exchanges. For such cases, improved a priori profiles representing the realistic atmospheric state will reduce the difference.

### 4.3 Comparison of circular vs. cone co-location criterion for validation of S5P methane data

In our standard S5P CH$_4$ validation settings with or without smoothing, we have used a co-location radius of 100 km around

each ground-based site. As the operational S5P CH$_4$ pixels are currently provided only over land, the circular co-location criterion may not be optimal to be applied for all sites. Ground-based sites located close to a sea/ocean coast will always lack S5P CH$_4$ pixels over water. Furthermore, for sites located close to regions with high emission sources, there are possible scenarios when the ground-based FTIR line-of-sight is not covering all pixels observed by the satellite using the circular co-location criterion. This is also relevant for high latitude sites where the ground-based FTIRs, mostly measuring at high solar

zenith angles, are always looking south for Northern Hemispheric sites and are looking north for Southern Hemispheric sites. We have implemented a cone selection criterion where we follow the ground-based FTIR line-of-sight with a 1° opening angle of the cone at the highest altitude. Using the cone co-location criterion, we have done the validation of the S5P bias-corrected CH$_4$ data with smoothing and compared to the validation results using circular co-location criterion using the same settings as discussed in section 4.1.

The validation results of the S5P bias-corrected XCH$_4$ data relative to the TCCON and NDACC data applying cone co-location criterion are shown in columns 16–20 of Tables 4 and 5, respectively. Using the cone co-location criterion reduces the number of S5P co-locations with ground-based FTIRs significantly (see column 16 in relation to column 3). The S5P XCH$_4$ relative bias and the standard deviation of the relative bias with respect to TCCON and NDACC using the cone co-location criterion are shown as orange bars in the left panel and right panel plots of Figs. 1 and 9, respectively. The standard deviation

of the relative bias with the cone co-location criterion is smaller than the standard deviation of the relative bias for the circular co-location criterion for sites with significantly reduced co-locations and is similar for other sites with small reduction in





the number of co-locations. The difference between the relative bias with circular and cone co-location criterion for the S5P bias-corrected XCH$_4$ data for each TCCON and NDACC station is shown as orange bars in the middle panel plot (labeled – diff_circvscone) of Figs. 1 and 9, respectively. The difference plot relative to TCCON shows the magnitude of change in bias, with values for some stations being negative while for others stations being positive. We observe a maximum difference
of 0.3 % (~5.7 ppb) and a mean difference of -0.02±0.12 % (~-0.4±2.3 ppb) across all TCCON sites for the duration of available measurements used in this study. The high latitude sites in the Northern Hemisphere show a significantly low number of co-locations for the cone criterion. The relative bias for these sites, Eureka, Ny-Ålesund, Sodankylä, and East Trout Lake, shows a slight increase for the cone co-location criterion in comparison to the circular co-location criterion. Sites where the
relative bias using the cone criterion as compared to the circular criterion is lower by at least 2 ppb are the following: JPL (-0.2 %), Pasadena (-0.18 %), Lamont (-0.11 %), and Białystok (-0.11 %). While the sites where the cone criterion as compared to the circular criterion is higher by at least 2 ppb are the following: Lauder (0.3 %), Saga (-0.18 %), and Orléans (0.12 %). The difference plot relative to NDACC shows the magnitude of change in bias with values for some stations being negative while for other stations being positive. We observe a maximum difference of 0.48 % (~9.1 ppb) and a mean difference of
0.01±0.2 % (~0.2±3.8 ppb) across the selected NDACC sites (see Table 5) for the duration of available measurements used in this study. Several sites have few co-locations left upon selecting the cone criterion with Ny-Ålesund showing no match at all. Amongst the sites where a significant number of co-locations remains, the sites where the relative bias using the cone criterion as compared to the circular criterion is lower by at least 2 ppb are the following: Altzomoni (0.48 %), Sodankylä (0.14 %), and Jungfraujoch (-0.14 %). The sites where the cone criterion as compared to the circular criterion is higher by at least 2 ppb are
the following: Lauder (-0.30 %), Kiruna (0.25 %), Bremen (-0.15 %), and St. Petersburg (-0.12 %).

We have observed that applying the cone co-location criterion reduces the number of co-locations for all sites and quite significantly for some sites. There are seven TCCON stations and seven NDACC stations where the magnitude of the difference is above 2 ppb. Amongst all the stations, the magnitude of change in the relative bias between the two settings is the highest for Altzomoni station (see section 5.3 for further discussion on the site).

**4.4 Solar zenith angle dependence of the S5P methane bias relative to TCCON and NDACC**

The remote sensing measurements made either from the ground or satellite are known to be affected by the solar zenith angle (SZA) of the measurements. In this section, we show the S5P CH$_4$ bias relative to the ground-based reference data as a function of the measurement SZA. Figure 17 shows the S5P relative bias for the a priori aligned and smoothed case as a function of the measurement SZA against the reference ground-based TCCON (top panel) and NDACC (bottom panel) stations. As mentioned
in section 2.1, the S5P CH$_4$ data are only available for SZA≤ 70°. The upper limits of the plots therefore show values only until 70°. The S5P relative bias shows a high scatter for high SZAs. The high positive bias at high SZA is from the spring measurements at high latitude sites which are influenced by surface conditions with snow cover and polar vortex conditions. Whereas, the negative bias at high SZA is from the summer and autumn measurements (e.g., see Figs. 6 and 7). In order to see this effect in detail we plotted the S5P relative bias against SZA at few stations as shown in Fig. 18. Stations like Sodankylä,
East Trout Lake, and Park Falls show high scatter in the relative bias for measurements at high SZAs when measurements





are performed during spring months. At Lamont we observe a strong increase in bias with decreasing SZA for measurements performed during spring. This is seen particularly in the case where the bias correction due to the SWIR surface albedo change occurred between 0.25 and 0.1 for measurements performed in this period at the site. The bias increase with decreasing SZA is also seen for other months at the different sites. Except for the spring measurements, which show a high bias, we observe a

general decrease in relative bias with increasing SZA.

## 5   Validation of S5P carbon monoxide products

The validation of the S5P carbon monoxide data with the ground-based FTIR data from TCCON and NDACC stations is discussed in this section. The official S5P CO products are available over land as well as over water. As a result, in addition to the stations mentioned in the S5P methane validation results, co-locations with ground-based stations located on islands

(e.g., Ascension, Izaña, Réunion, and Mauna Loa) are found and discussed here. The NDACC station at Paramaribo is the only station in the South American continent currently contributing to the S5P CO validation study. As NDACC provides the CO column values, they are used directly to validate the S5P CO column values. Whereas for the validation using TCCON XCO data, the S5P CO columns are converted to XCO as described in section 3.

## 5.1   Validation of S5P XCO data using TCCON standard and unscaled XCO data and analysis of smoothing
uncertainty

As mentioned in section 2.2, the validation of the S5P XCO offline data is performed with the TCCON standard XCO data as well as the TCCON unscaled XCO data and the results are discussed in this section. The density of the official S5P valid CO pixels is higher as compared to the valid $CH_4$ pixels. As a result, we found that using a co-location radius of 50 km around each ground-based station gave a sufficient number of pixels for robust statistics. We have used a maximal time difference

of 1 h for TCCON observations, which is similar to the settings used for $CH_4$ validation. An effective location of the FTIR measurement on the line-of-sight is used to do the co-location. As a result, the co-located pixels can differ from measurement to measurement. For each of the ground-based measurements, which are co-located with the S5P measurements, an average of all S5P pixels is made. Co-located pairs are created between ground-based and averaged S5P pixels only if a minimum of five pixels is found in applying the coincidence criteria. In the comparison, the a priori in the TCCON retrievals have been

substituted with the S5P CO a priori following Eq. A1. The TCCON results with the S5P prior substituted are then compared directly to the S5P XCO data. Furthermore, each validation run includes the adaptation of the S5P columns to the altitude of the ground-based FTIR instruments.

   Table 6 provides the validation results using the a priori aligned TCCON unscaled and standard XCO data at each TCCON station. The systematic difference (the mean of all relative differences) between the S5P and TCCON data is on average

9.14±3.33 % (TCCON standard XCO data) and 2.36±3.22 % (TCCON unscaled XCO data). These results are well within the mission requirements for a bias of 15 %, also the relative bias at each ground-based station is below the requirements. While most stations show a positive relative bias of S5P XCO with respect to the TCCON unscaled XCO, there are few exceptions



that show high negative values (e.g., Xianghe, JPL, and Pasadena - all urban sites). This will be further discussed in detail later in this section. The standard deviation of the relative bias, which is a measure of the random error, is well below the mission

requirement for a random error of <10 % for comparison against both TCCON standard and unscaled XCO data at all stations except at Wollongong where the value is 18.12 % (for TCCON unscaled XCO) and 19.37 % (for TCCON standard XCO). The high standard deviation of the relative bias at this station is due to the co-location mismatch during the period of fire event in that region producing enhanced CO plume passing over/nearby the ground-based station at Wollongong. As a result for some of the days we found enhanced CO values in the S5P co-located pixels, which were not observed by the FTIR as the enhanced

CO plume is not directly in the line-of-sight of the FTIR, while for other days we found enhanced CO values varying during the day as the fire plume passes by the station and in comparison the satellite measures for a shorter duration during the local noon and therefore misses the variability of CO during the co-location time selected for the validation. We tested with a reduced time co-location criterion of 30 min and found that, for the Wollongong station, the standard deviation of the relative bias reduced marginally to 18.05 % and the relative bias reduced to 2.03 % (for TCCON unscaled XCO validation results). The CO plumes

emitted from the Australian fire during the summer of 2019/2020 were also observed at the Lauder station in New Zealand. The CO was well dispersed by the time the fire plumes were measured there, resulting in a better match between the S5P and ground-based FTIR measured XCO (see Figs. 21 and 23).

Figure 19 shows the bar plots for the S5P XCO relative bias (left panel) and the standard deviation of the relative bias (right panel) with respect to the TCCON XCO data at all stations. The comparisons relative to the TCCON unscaled XCO

data (labeled – unscsm50k1h) are the blue bars and those for the TCCON standard XCO data (labeled – stdsm50k1hr) are the magenta bars. The relative bias of the S5P XCO data with respect to the TCCON unscaled XCO data is systematically lower than the relative bias with respect to the TCCON standard XCO data. The difference of the relative bias for S5P XCO data using the TCCON unscaled XCO and the standard XCO data for each station is shown as magenta bar in the middle panel plot (labeled – diff_unscvsstd) of Fig. 19. It shows the overall direction of change is negative with mean value of -6.78±0.57 %

for all stations. This result confirms the previously reported studies (Kiel et al., 2016; Sha et al., 2018b; Zhou et al., 2019). The standard deviation of the relative bias for the S5P XCO data relative to the TCCON unscaled and standard XCO data are comparable.

The time series of the S5P XCO and TCCON unscaled XCO data for each site are shown in Figs. 20 and 21. The ground-based TCCON XCO data are represented in grey and the S5P XCO data during that period are shown in light red. The S5P data

co-located with TCCON data are shown in red and the co-located TCCON data with a priori alignment are shown in black. The S5P and TCCON measurements observe the same seasonal cycle of CO. At the Northern Hemispheric sites, the high CO values are observed during winter and low values are observed during summer dominated by the OH variation (Té et al., 2016). At Southern Hemispheric sites, the high CO values are observed during September – November dominated by the influence of biomass burning (Duflot et al., 2010; Zeng et al., 2012). In addition to the seasonal cycle, we also see that at several of the

ground-based sites, S5P and TCCON observe sometimes very high values of CO. These enhanced CO concentrations are due to the passing of the plumes with elevated CO concentrations over/nearby the station location (e.g., high CO seen at Wollongong during the Australian forest fires November 2019 – February 2020). Yurganov et al. (2004) also reported enhanced CO buildup





measured at several sites with values much larger than the emission estimates. The time series of the relative bias plots shown in Figs. 22 and 23 indicate a seasonal cycle with a high bias seen during the high CO event and low bias seen during the low

CO event. Sometimes very low S5P XCO values are observed in the validation plots at some stations, which pass the quality filter and find a match with the reference TCCON XCO data following our selection criterion. In these particular cases, we observe very low values in the relative bias plots. However, there are only a few occurrences of such low S5P XCO values.

The relative biases are plotted as mosaic plots and shown in Fig. 24, where the top panel shows the S5P bias with respect to the TCCON standard XCO data while the bottom panel shows the S5P bias with respect to the TCCON unscaled XCO data.

Each bar in the mosaic plots represents the weekly averages of the relative bias values. We will focus on the comparison of the results using TCCON unscaled XCO data. As mentioned in the previous paragraph, we observe a high positive bias during the high CO event periods, which is then reduced and even switched sign to show a negative bias during the low CO event periods. As TCCON performs solar absorption measurements, data are not available during winter for high latitude stations.

Taylor diagrams for the S5P XCO and TCCON unscaled XCO data are shown in Fig. 25. The correlation, represented by the

540 angular coordinate, is above 0.9 for most stations (see Table 6 for exact values), and the distance to the origin of the ground-based dot relative to the satellite dot is below 1 for most stations implying that the satellite data are more variable than the ground-based data. The good correlation indicates that the short scale temporal variations in the XCO column captured by the ground-based instruments are moderately reproduced by S5P. Outliers such as Ascension, Zugspitze, and JPL are due to the limited data sets available for the comparison. The altitude correction of the pixels works well as can be seen by the relatively

good correlation for Zugspitze, however, the scatter in the data is high.

In this section, we further show the results focusing on the effect of smoothing while doing the S5P XCO validation against TCCON unscaled XCO data. S5P and TCCON have different vertical sensitivities (averaging kernels) and use different a priori profiles for their retrievals. The different a priori and vertical sensitivities should be taken into account in the validation. In case of TCCON only an a priori alignment is done. Smoothing corrections are most relevant for cases with strong dynamic variability

in the atmosphere. TCCON performs a profile scaling retrieval on the measurements performed in the NIR spectral range and provides XCO. In order to ascertain the effect of smoothing correction, the results of the S5P validation using TCCON unscaled XCO are compared to the S5P validation results using a priori aligned TCCON unscaled XCO data.

The validation results of the S5P XCO data relative to the TCCON unscaled XCO data without smoothing correction (direct comparison) are shown in columns 12–15 of Table 6. The S5P XCO relative bias and the standard deviation of the relative bias

with respect to the TCCON unscaled XCO data are shown as grey bars (labeled – unsc50k1h) in the left panel and right panel plots of Fig. 19. It can be seen that there exists an apparent interhemispheric difference in the bias for the direct comparison case (grey bars) between the Southern Hemispheric and Northern Hemispheric sites. This difference is greatly reduced when smoothing uncertainties are correctly accounted (blue bars) in the validation results (see left panel of Fig. 19). The difference between the relative bias with and without a priori alignment for the S5P XCO data for each TCCON station are shown as grey

bars in the middle panel plot (labeled – diff_smvsnosm) of Fig. 19. The magnitude of change between the smoothed and direct comparison is larger in the Southern Hemisphere than in the Northern Hemisphere with exception for sites located in high polluted regions. The change at some stations (e.g., the Southern Hemispheric sites and high polluted sites) is significant as it



is larger than the XCO error estimated in Wunch et al. (2015). Zhou et al. (2019) reported similar findings for a comparison between six co-located sites, where both NDACC and TCCON CO measurements were performed. The difference plot shows
the highest value of -17.43 % for Xianghe, a station located in a polluted area, due to a very high a priori difference from the true atmospheric state. As a result, the CO volume mixing ratio (VMR) at the surface is relatively high but it is not represented by the TCCON a priori, leading to an underestimation from the smoothing uncertainty. The same is true for other stations like Karlsruhe (change of -5.71 %), and Pasadena (change of -3.65 %). We observe a mean difference of 0.33±4.32 % across all TCCON stations. Figure 19 shows the TCCON stations where the a priori alignment uncertainty plays an important role in the
bias and needs to be accounted for in the CO validation studies.

## 5.2  Validation of S5P CO column data using NDACC CO column data and analysis of smoothing uncertainty

In this section, the validation results of the S5P CO columns using NDACC CO columns are discussed. The S5P observations co-located with the NDACC measurements are found by selecting all filtered S5P pixels within a radius of 50 km around each site and with a maximal time difference of 3 h. An effective location of the measurement on the line-of-sight is used
to do the co-location. The co-located pixels can therefore differ from measurement to measurement. For each of the NDACC measurements, co-located with the S5P measurements, an average of all S5P pixels is done. Co-located pairs are created between NDACC and averaged S5P only if a minimum of five pixels is found in applying the coincidence criteria. In addition to the direct comparison of the S5P and NDACC CO columns (referred to as NDACC CO un-smooth), the NDACC CO column values are additionally aligned with the S5P prior (referred to as NDACC CO ap-smooth) and used for the S5P validation, and
in a further step the NDACC CO column values with the S5P prior substituted are additionally smoothed with the S5P column averaging kernel (referred to as NDACC CO smooth) following Eq. A2 and used for S5P validation. Each validation run also includes the adaptation of the S5P columns to the altitude of the ground-based FTIR instruments.

Table 7 provides the validation results for the S5P CO columns using smooth, un-smooth, and ap-smooth NDACC CO column data at each NDACC station. The systematic difference (the mean of all relative differences) between the S5P and NDACC data is on average 6.86±4.7 % (NDACC CO un-smooth), 4.37±5.88 % (NDACC CO ap-smooth), and 7.62±5.27 %
(NDACC CO smooth). This is well within the mission requirements for a bias of 15 %. However, the values are outside the requirements at Altzomoni and Arrival Heights stations. Eliminating the results of these two stations from the statistics of the overall stations, we observe the systematic difference between the S5P and NDACC data is on average 5.75±3.09 % (NDACC CO un-smooth) 3.18±4.5 % (NDACC CO ap-smooth), and 6.44±3.79 % (NDACC CO smooth). The NDACC station
at Altzomoni is located at a high altitude in the south-west direction of the Mexico City (Plaza-Medina et al., 2017; Baylon et al., 2017). The station is located <60 km from the city center. As a result, the emission from the world's eighth-largest megacity, with >22 million population in its metropolitan area, plays a significant role in the satellite footprint (Stremme et al., 2013; Borsdorff et al., 2018a, 2020). In the example plot shown in Fig. 26, we can see that the ground-based FTIR located at Altzomoni, with the line-of-sight to the south indicated by the yellow line, is not able to observe the high CO values located to
the north-west of the station, which are selected for S5P using our co-location criterion. However, using the cone co-location criterion as described in section 4.3 we can eliminate the pixels with high CO values that are not in the line-of-sight of the





FTIR instrument and thereby reduce the co-location mismatch. The bias at Arrival Heights, high latitude background station located on the Antarctic continent showing very low values of CO, is slightly worse than the requirement, while the random error is better than the requirement of 10 %. The mean standard deviation of the relative bias, which is a measure of the random

error, is well below the requirements of <10 % for validation using both smoothed and direct NDACC CO data. However, there are few exceptions for stations like Altzomoni, Wollongong, and Boulder. The high values are due to the co-location mismatch during the high CO events (e.g., passage of a plume with a high CO concentration in the vicinity of the site) observed at these sites.

Figure 27 shows the bar plots for the S5P CO relative bias (left panel) and the standard deviation of the relative bias (right

panel) with respect to the NDACC CO column data at all stations. The comparisons relative to the NDACC smoothed CO data (labeled – ALLsm50k3h) are the blue bars, those for the NDACC un-smooth CO data (labeled – ALL50k3hr) are the magenta bars, and those for the NDACC ap-smooth CO data (labeled – ALLap50k3h) are the grey bars. The high latitude stations show a high bias while some stations like Paramaribo, Izaña, Mauna Loa show a low bias. The difference of the relative bias for S5P CO data for the NDACC smoothed CO (labeled – diff_smvsnosm) and NDACC ap-smooth (labeled – diff_apvsnosm)

relative to the un-smooth CO data for each station are shown as magenta and grey bars in the middle panel plot of Fig. 27. It shows the magnitude of change in bias with values for some stations being positive while for other stations being negative. The effect of smoothing appears to be dependent on the station location. We observe a maximum difference of -6.89 % and a mean difference of 0.69±2.79 % for all stations for the diff_smvsnosm case. And we observe a maximum difference of -9.38 % and a mean difference of -2.57±2.79 % for all stations for diff_apvsnosm case. The changes at some stations are significant as it is

larger than the CO column error estimated in NDACC. The standard deviation of the relative bias for the S5P CO data relative to the NDACC CO data with and without smoothing is comparable.

The time series of the S5P CO column and NDACC smoothed CO column data for each site are shown in Figs. 28 and 29. The ground-based NDACC CO data are represented in grey and the S5P data during that period are shown in light red. The S5P data co-located with NDACC data are shown in red and the co-located NDACC smoothed data are shown in black. The

implication of the altitude correction can easily be seen for stations located at high altitude (e.g., Zugspitze, Jungfraujoch, Izaña, Mauna Loa, Altzomoni, Maïdo). The S5P and NDACC measurements observe the same seasonal cycle of CO. Similar to the TCCON results, we also see that at several of the NDACC sites, S5P and NDACC sometimes observe very high values of CO columns due to the passing of the plumes with elevated CO concentrations over/nearby the station location (e.g., Wollongong, Boulder, St. Petersburg). The time series of the relative bias plots shown in Figs. 30 and 31 indicate a seasonal cycle with a

high bias seen during the high CO event and low bias seen during the low CO event. The high scatter observed at the Toronto site is related to the scatter observed in the ground-based NDACC CO column data at the site.

The relative biases of the S5P CO column and NDACC smoothed CO column data for each site are shown as a mosaic plot in Fig. 32. Each bar in the mosaic plot represents the weekly averages of the relative bias values. The plot shows high positive bias during the high CO event periods, which is then reduced and even switched sign to show negative bias during the low CO

event periods. The bias at few stations like Toronto, Altzomoni, and Arrival Heights appear as outliers in the plot. As NDACC





CO column data are retrieved from solar absorption measurements, the data are not available during few weeks in winter for high latitude stations when the sun is very low on the horizon.

Taylor diagrams for the S5P CO column and NDACC smoothed CO column data are shown in Fig. 33. The correlation, represented by the angular coordinate, is above 0.9 for most stations (see Table 7 for exact values), and the distance to the origin of the ground-based dot relative to the satellite dot is below 1 for most stations (except at Paramaribo and Rikubetsu which is due to the limited data sets available for the comparison) implying that the satellite data is more variable than the ground-based data. The good correlation indicates that the temporal variations in the CO column captured by the ground-based instruments are reproduced very similarly by S5P. Outliers such as Wollongong, Boulder, and Altzomoni are due to the co-location mismatch during the high CO events (e.g., passage of a plume with a high CO concentration in the vicinity of the site) observed at these sites. The altitude correction of the pixels works well as can be seen by the relatively good correlation at the high altitude stations.

Eleven ground-based stations (Eureka, Ny-Ålesund, Bremen, Karlsruhe, Garmisch, Zugspitze, Rikubetsu, Izaña, Réunion–Maïdo, Wollongong, and Lauder) contributed to the validation study by providing CO data from both TCCON and NDACC measurements performed at the sites. The mean difference in the relative bias of the S5P CO data with respect to the smoothed NDACC and TCCON (bias$_{\text{S5PvsNDACC}}$–bias$_{\text{S5PvsTCCON}}$) for these eleven stations is -4.31±3.7 %. This indirectly implies that the NDACC CO is 4.31±3.7 % larger than TCCON CO data. The ground-based data available for these eleven stations do not always cover the same period. Therefore, this is only a qualitative estimate indicating the mean difference between NDACC and TCCON CO data at these eleven sites. Zhou et al. (2019) showed that the bias between co-located and smoothed TCCON and NDACC XCO data products for six stations has a mean value of 6.8 % (range 5.6 %–8.6 %). Our indirect comparison results for more sites and not exactly co-located ground-based data for the TCCON and NDACC show similar differences.

### 5.3 Comparison of circular vs. cone co-location criterion for validation of S5P carbon monoxide data

In our standard S5P CO validation settings with or without smoothing, we have used a co-location radius of 50 km around each ground-based site. In this section, we will discuss the validation results of the S5P CO column data with the smoothed ground-based data following the cone co-location criterion as described in section 4.3. These results are further compared to the circular co-location criterion using the same settings.

The application of the cone co-location criterion is shown in Fig. 26 for one sample day. The top-left panel plot shows all available S5P pixels containing CO column number density data in the overpass file. The Altzomoni station is marked at the center of the plot. The high CO values to the north-west of the station are the footprint of the CO from Mexico City. Towards the northeast side of the station some missing pixels are filtered due to clouds. The top-right panel plot shows the co-located S5P pixels with circular co-location criterion with a radius of 50 km as used for the CO validation study. As seen in the plot, there are few pixels with high CO values in the north-west, which are included in the selected pixels. The yellow line in the plot represents the line-of-sight of the ground-based FTIR at Altzomoni. Therefore, the high CO values in the north-west will not be observed by the FTIR measurement. This mismatch is a cause of the potential bias. The bottom panel plot shows the



co-located S5P pixels with the cone co-location criterion with 1° opening angle of the cone at the highest altitude. The selected S5P pixels using the cone co-location criterion are in the line-of-sight of the ground-based FTIR instrument and will potentially reduce a mismatch and therefore lowering the potential bias between the satellite and ground-based data.

The validation results of the S5P CO data relative to the TCCON and NDACC data with smoothing and applying cone co-location criterion are shown in columns 16–20 of Tables 6 and 7, respectively. Using the cone co-location criterion only 670 marginally reduces the number of S5P co-locations with ground-based FTIRs (see column 16 in relation to column 3). This is due to the high density of the official S5P valid CO pixels availability. The S5P CO relative bias and the standard deviation of the relative bias with respect to TCCON and NDACC using the cone co-location criterion are shown as orange bars in the left panel and right panel plots of Figs. 19 and 27, respectively. The S5P CO relative bias is comparable or slightly smaller for the cone co-location criterion as compared to the circular co-location criterion. The standard deviation of the relative bias 675 with the cone co-location criterion is similar to the standard deviation of the relative bias for the circular co-location criterion. The difference between the relative bias with circular and cone co-location criterion for the S5P CO data for each TCCON and NDACC station is shown as orange bars in the middle panel plot (labeled – diff_circvscone) of Figs. 19 and 27, respectively. The difference plot relative to TCCON shows the magnitude of change in bias, with values for some stations being negative while for other stations being positive. We observe a maximum difference of 0.6 % and a mean difference of -0.02±0.24 % 680 across all TCCON sites for the duration of available measurements used in this study. Sites where the relative bias using the cone criterion as compared to the circular criterion is outside the 1 $\sigma$ limit of the mean are Eureka (0.6 %), Garmisch (0.48 %), Paris (0.23 %), Ny-Ålesund (-0.37 %), Xianghe (-0.37 %), JPL (-0.49 %), Pasadena (-0.43 %). The difference plot relative to NDACC shows the magnitude of change in bias, with values for some stations being negative while for other stations being positive. We observe a maximum difference of -1.24 % and a mean difference of -0.05±0.49 % across the selected NDACC 685 sites for the duration of available measurements used in this study. The sites where the relative bias using the cone criterion as compared to the circular criterion is outside the 1 $\sigma$ limit of the mean are Eureka (0.78 %), Harestua (-0.53 %), Zugspitze (-0.84 %), Jungfraujoch (-0.7 %), Boulder (0.64 %), Arrival Heights (-1.24 %). The high difference is observed mostly for the high latitude stations where the cone co-location criteria following the ground-based FTIR line-of-sight is the best choice.

## 5.4 Validation of S5P CO (CLSKY, CLOUD, and ALL) data using TCCON and NDACC data sets

As discussed in section 2.1, we separated S5P retrievals performed for measurements under clear-sky (CLSKY; cloud optical thickness <0.5 & cloud height <500 m, over land) and cloudy conditions (CLOUD; cloud optical thickness ≥ 0.5 & cloud height <5000 m, over land and ocean) in addition to our standard all case (ALL; cloud height <5000 m over land and ocean). The validation results of S5P CO for ALL settings have been discussed in detail in sections 5.1 – 5.3. In this section, we show the validation results of the S5P CO for CLSKY and CLOUD settings against TCCON unscaled XCO with a priori alignment 695 and NDACC CO column data with smoothing and compare the results in relation to the results of the ALL settings. Each validation run includes the adaptation of the S5P columns to the altitude of the ground-based FTIR instruments.

Tables 8 and 9 provide the validation results for the S5P CO data for ALL case, CLSKY case, and CLOUD case at each TCCON and NDACC station. The systematic difference (the mean of all relative differences) between the S5P and unscaled





TCCON data is on average 2.36±3.22 % (ALL case), 2.83±3.44 % (CLSKY case), and 1.91±3.08 % (CLOUD case). These
700 results are well within the mission requirements for a bias of 15 % as well as the relative bias at each ground-based station is
also below the requirements. The standard deviation of the relative bias which is a measure of the random error is well below
the mission requirement for a random error of <10 % for all sites except at Wollongong (ALL and CLOUD cases) and Pasadena
(CLOUD case).

Figure 34 shows the bar plots for the S5P XCO relative bias (left panel) and the standard deviation of the relative bias (right
panel) with respect to the TCCON XCO data at all stations. The comparisons relative to the TCCON unscaled XCO data for
ALL case (labeled – unscsm50k1hALL) are the blue bars, those for the CLSKY case (labeled – unscsm50k1hCLSKY) are the
red bars, and those for the CLOUD case (labeled – unscsm50k1hCLOUD) are the green bars. The middle panel plot of Fig. 34
shows for each TCCON station the difference of the relative bias for S5P XCO data using the TCCON unscaled XCO ALL case
and the CLSKY case (labeled – diff_ALLvsCLSKY) as red bars, as well as the CLOUD case (labeled – diff_ALLvsCLOUD)
as green bars. The overall direction of change for the CLSKY case is negative with few exceptions, the maximum value of
change is 2.21 % and a mean value of -0.48±0.89 % for all stations. The overall direction of change for the CLOUD case is
positive with few exceptions, maximum value of change is 1.58 % and a mean value of 0.45±0.59 % for all stations.

The systematic difference (the mean of all relative differences) between the S5P and NDACC data is on average 7.62±5.27 %
(ALL case), 7.8±5.11 % (CLSKY case), and 7.65±5.18 % (CLOUD case). This is well within the mission requirements for a
715 bias of 15 %. However, the values are outside the requirements for the validation results at the Altzomoni and Arrival Heights
stations. Eliminating the results of these two stations from the statistics of the overall stations, we observe that the systematic
difference between the S5P and NDACC data is on average 6.44±3.79 % (ALL case), 6.56±3.35 % (CLSKY case), and
6.53±3.91 % (CLOUD case). The bias at Arrival Heights, a high latitude station located on the Antarctic continent, is slightly
worse than the requirement, while the random error is better than the requirement of 10 %. The mean standard deviation of the
720 relative bias, which is a measure of the random error, is well below the requirements of <10 % for all three cases of validation
results with few exceptions for stations like Altzomoni, Wollongong, and Boulder. The high values are due to the co-location
mismatch during the high CO events (e.g., the passage of a plume with a high CO concentration in the vicinity of the site)
observed at these sites.

Figure 35 shows the bar plots for the S5P CO relative bias (left panel) and the standard deviation of the relative bias (right
panel) with respect to the NDACC CO column data at all stations. The comparisons relative to the NDACC CO column data
for ALL case (labeled – ALLsm50k3h) are the blue bars, those for the CLSKY case (labeled – ALLsm50k3hCLSKY) are
the red bars, and those for the CLOUD case (labeled – ALLsm50k3hCLOUD) are the green bars. The middle panel plot of
Fig. 35 shows for each NDACC station the difference of the relative bias for S5P CO column data using the NDACC CO
column ALL case and the CLSKY case (labeled – diff_ALLvsCLSKY) as red bars, as well as the CLOUD case (labeled –
730 diff_ALLvsCLOUD) as green bars. The direction of change for the CLSKY and CLOUD cases is negative for some stations
while for other stations it is positive. The maximum value of change is 2.68 % and a mean value of 0.17±1.0 % for CLSKY
case for all stations. The maximum value of change is 1.64 % and a mean value of -0.09±0.77 % for CLOUD case for all
stations.





The CLSKY and CLOUD selection criteria can be useful in case of specific applications. For example, the CLSKY case
helped to reduce the standard deviation of the relative bias for Wollongong's TCCON and NDACC validation results. This
is related to the significant filtering of the pixels over the ocean that are missing in the CLSKY case. The satellite clear-sky
observations made over ocean have too low signal in the SWIR spectral region and are therefore filtered out. However, the
ALL case results are quite comparable to the CLSKY and CLOUD cases and are therefore used as the general S5P CO data
set in our validation studies.

**5.5    Solar zenith angle dependence of the S5P carbon monoxide bias relative to TCCON and NDACC**

In this section, we show the S5P carbon monoxide bias relative to the ground-based reference data as a function of the measurement SZA. Figure 36 shows the S5P relative bias for the a priori aligned and smoothed case as a function of the measurement
SZA against the reference ground-based TCCON (top panel) and NDACC (bottom panel) stations. As mentioned in section 2.1,
the S5P carbon monoxide data are only available for SZA<80°. The upper limits of the plots therefore show values only till
80°. As explained in section 5.2, the high values of S5P relative bias are observed during winter (measurements performed
mostly at high SZAs) and the low values during summer (measurements performed mostly at low SZAs). We therefore observe
this in the S5P relative bias plotted against SZA. Figure 37 shows individual plots for a few sample stations. We observe that
the relative bias increases with increasing SZA of the measurement. This increase is about 10 % over the complete range of
measurements SZAs.

**6    Conclusions**

In this study, we have done the geophysical validation of Sentinel-5 Precursor operational methane and carbon monoxide data
sets (see Table 1 for version details) using reference ground-based TCCON and NDACC stations. A total of 28 TCCON stations
and 23 NDACC stations covering a wide latitudinal range (Eureka 80° N to Arrival Heights 77.8° S), various atmospheric
conditions (dry, humid, clean and polluted), various surface conditions (range of surface albedo), flat and high altitude terrains,
oceanic terrain) have been used in this study. Furthermore, the combined use of the near-infrared TCCON data and mid-infrared
NDACC data, as a whole network and at co-located stations, with their benefits helped to evaluate the Sentinel-5 Precursor
operational methane and carbon monoxide product's quality in our validation study.

We found that the systematic difference between the S5P standard $XCH_4$ and a priori aligned TCCON data is on average
-0.69±0.73 %. The systematic difference changes to a value of -0.25±0.57 % for the S5P bias-corrected $XCH_4$ data. The bias
for both S5P standard and bias-corrected $XCH_4$ data is well within the mission requirements for bias (systematic error) of
1.5 %. We also found that the random error is well below the mission requirements for a random error of 1 % for both standard (0.59±0.17 %) and bias-corrected (0.57±0.18 %) S5P $XCH_4$ data. Most stations show a correlation above 0.6, the poor
correlation at some sites are mostly dominated by the seasonal cycle or due to limited data sets available for the comparison.
The systematic difference between the S5P standard and bias-corrected $XCH_4$ against smoothed NDACC data are on aver-
age 0.±1.12 % and 0.64±0.77 %, respectively. This is well within the mission requirements. As the accuracy and precision of





NDACC $CH_4$ data are lower than TCCON, conclusions about the S5P systematic and random error are drawn based on TCCON validation results. The bias-correction of the S5P $XCH_4$ data being a function of the retrieved surface albedo acts differently at different locations. We observe high scatter in the relative bias for low surface albedo conditions. A seasonal dependency of the relative bias is seen. We observe a high bias during the springtime measurements at high SZAs for high latitude sites and a decreasing bias with increasing SZA for the rest of the year at all sites. The SZA dependence of the bias includes albedo correction and a priori difference from the true atmospheric state. We estimated the contribution of the a priori alignment uncertainty at the ground-based stations and found values up to ~4.8 ppb at a TCCON station with mean value of ~-2.7±1.3 ppb. The mean value of the smoothing uncertainty contribution at the NDACC stations is ~7±5.3 ppb with some stations showing high values of up to ~41.4 ppb. At the co-located TCCON and NDACC stations, we observed the highest contribution of the a priori alignment and smoothing uncertainty for mid-latitude TCCON stations, whereas for the NDACC stations we observe the highest contribution for the high latitude stations. The comparison with a priori alignment and taking smoothing effects into account is recommended as the preferred method for validation. We found that using the cone co-location criterion improves the co-location between the satellite and ground-based station by observing similar airmass. This is crucial for certain stations, which are located closer to emission sources or high latitude ones. Currently, we found seven TCCON and NDACC stations where the bias changed by more than 2 ppb between the circular and cone co-location settings. The cone criterion also significantly reduces the number of co-locations for some sites thereby making the statistics less reliable for those sites.

We found that the systematic difference between the S5P XCO and a priori aligned TCCON data is on average 9.14±3.33 %. Due to the uncertainty of the scaling slope of XCO in TCCON to tie the TCCON XCO measurements to WMO in situ scale, we have also used the unscaled TCCON XCO data (without application of the empirical scaling factor) for S5P XCO validation. We found that the systematic difference between the S5P XCO and a priori aligned TCCON unscaled XCO data is on average 2.36±3.22 %. Both results are within the mission requirements for bias (systematic error) of 15 %. We found that the difference of the relative bias using the TCCON unscaled XCO and the TCCON standard XCO data is on average -6.78±0.57 %. We estimated the contribution of the a priori alignment uncertainty in the validation and found that the magnitude of change between the a priori aligned and direct comparison is larger in the Southern Hemisphere than in the Northern Hemisphere except for sites located in polluted regions. The a priori alignment uncertainty contribution is significant at several sites as it is larger than the estimated TCCON XCO error. We observe a mean difference of 0.33±4.32 % across all TCCON stations with highest values of -17.43 % for Xianghe (due to very high a priori profile difference). We found that the systematic difference between the S5P CO column and the NDACC CO column data (excluding two stations which were obvious outliers) is on average 5.75±3.09 % (NDACC CO direct comparison), 3.18±4.5 % (NDACC CO smoothed by using S5P a priori as the common prior), and 6.44±3.79 % (NDACC CO profile with S5P a priori substituted and additionally smoothed with S5P column averaging kernel). The effect of the smoothing depends on the station location with a mean difference of 0.69±2.79 % across all NDACC stations and a maximum value of -6.89 % in relation to the direct comparison. The effect of smoothing by doing only a priori substitution in relation to the direct comparison gives a mean difference of -2.57±2.79 % across all NDACC stations and a maximum value of -9.38 %. The comparison with a priori alignment and taking smoothing effects into account is recommended as the preferred method. Most TCCON and NDACC stations show a correlation above 0.9 indicating





that the temporal variations in CO column captured by the ground-based instruments are reproduced very similarly by S5P. The few exceptions are due to the limited data sets available for the comparison. We also found that the S5P random error for the TCCON and NDACC validation results is well below the mission requirements for a random error of 10 % except for few stations where a co-location mismatch occurs during certain periods with high values of CO events occurring due to plumes

passing over/nearby the stations. A seasonal dependency of the relative bias is seen. We observe a high bias during the high CO event and low bias during the low CO event. We observed a mean difference of -0.02±0.24 % with a maximum difference of 0.6 % for TCCON validation results using the cone co-location criterion compared to the circular co-location criterion. The results of the cone selection criterion at the NDACC stations show higher values than for the TCCON stations. We observe a mean difference of -0.05±0.49 % with a maximum difference of -1.24 %. The high difference is observed mostly for high

latitude stations, where the cone co-location criterion following the line-of-sight of the ground-based FTIR is the best choice in finding co-located satellite pixels for validation. Furthermore, we observed that the validation results of the clear-sky and cloud cases of S5P pixels are comparable to the validation results including all pixels passing the filter criteria. The clear-sky or cloud cases are however useful for certain applications. We observe that the relative bias increases with increasing SZA of the measurement. We estimated this increase to be 10 % over the complete range of measurement SZAs.

Based on the validation results of the S5P operational methane and carbon monoxide data sets against the reference ground-based TCCON and NDACC data sets, we conclude that the S5P methane and carbon monoxide data fulfils the mission requirements.

## Appendix A: Reducing a priori and averaging kernel contribution in the validation

The S5P and ground-based FTIR instruments have different instrument sensitivities and use different a priori profiles to retrieve

the best representation of the true atmospheric state from the recorded spectra. The S5P uses a priori derived from the TM5 model, whereas the TCCON uses a daily a priori profile generated by a stand-alone program provided by Toon and Wunch (2017) and NDACC uses a single a priori profile from climatology of the Whole Atmosphere Community Climate Model Version 6 (WACCM V6; ftp://nitrogen.acom.ucar.edu/user/jamesw/IRWG/2013/WACCM/V6/). In order to make the quantitative comparison, the influence of the a priori contribution to the smoothing equation needs to be compensated/corrected by

adjusting the retrieval results to a common a priori (Rodgers and Connor, 2003). The S5P prior is used as the common prior. It is re-gridded to the FTIR grid using a mass conservation algorithm (Langerock et al., 2015). For the case where the satellite pixel elevation is above the ground-based site altitude, the S5P prior profile is extrapolated (i.e., a simple extension, the lowest vmr is taken as the vmr at the lowest ground-based grid) to the altitude of the ground-based instrument. The re-gridded S5P prior $x_{a\_S5P}$ is substituted in the FTIR retrieval.

$$x_{FTIR\_mod\_prior} = x_{FTIR} + (I - A_{FTIR})(x_{a\_S5P} - x_{a\_FTIR}), \qquad (A1)$$

where $x_{FTIR}$ is the original vmr profile, $x_{a\_FTIR}$ is the a priori profile used for the original FTIR retrieval ($x_{FTIR}$), $x_{FTIR\_mod\_prior}$ is the corrected FTIR-retrieved profile, $A_{FTIR}$ is the FTIR averaging kernel matrix and $I$ is the unity matrix.





This step reduces the total smoothing uncertainty on the column differences by eliminating the uncertainty on the FTIR a priori. Although Eq. A1 is only valid for NDACC profiles, it can be modified to be applied for TCCON column data as well. In that

case, the prior profiles should be transformed to partial column profiles and divided by the total column of FTIR dry air.

For NDACC profiles, to further reduce the smoothing uncertainty contribution introduced by the averaging kernel, we smooth the corrected FTIR-retrieved profile ($x_{FTIR\_mod\_prior}$) with the S5P column averaging kernel ($cA_{S5P}$). This requires the re-gridding of the corrected FTIR-retrieved profile to the S5P column averaging kernel grid before applying the smoothing equation:

$$c_{FTIR\_smoothed} = c_{a\_S5P} + cA_{S5P}[(x_{FTIR\_mod\_prior} - x_{a\_S5P}) \times n_{dry,air}], \tag{A2}$$

where $c_{a\_S5P}$ is the column values derived from the S5P a priori profile, $c_{FTIR\_smoothed}$ is the smoothed FTIR column associated with a co-located S5P pixel. The $n_{dryair}$ in Eq. A2 is the partial column profile calculated from the pressure difference ($\Delta P$) between the layer interfaces and the hydrostatic equation:

$$\Delta P = m_{wet,air} \times n_{wet,air} \times g \tag{A3}$$

For CH$_4$, the partial column of dry air is available in the S5P Level 2 files. For CO, we derive it using the pressure on the boundaries as described in Eq. A3. In the above Eq. A3, n$_{wet,air}$ is approximated by n$_{dry,air}$ and the molar mass of wet air is approximated by the molar mass of dry air as there is no $H_2O$ profile available in the S5P prior. We found that this approximation has only a small influence, e.g., the bias change at Paramaribo, a tropical site, is about 0.2% when compared to the case of using NCEP $H_2O$ profile. If the satellite pixel elevation is below the FTIR site altitude, the re-gridding of the

corrected FTIR-retrieved profile is done such that the FTIR profile is extended with the S5P a priori profile. This extension of the a priori profile cancels on the right hand side of Eq. A3 and the FTIR smoothed column coincides with the S5P a priori partial column for the region where the grids mismatch.

## Appendix B: S5P pixel altitude correction

An altitude correction is done for each S5P pixel in order to take into account the altitude difference between the S5P pixels

and the ground-based station. The correction can be significant for co-location with mountain stations where the satellite pixels can be picked up from locations around the station, which are at lower or higher altitudes than stations. The scaling factor ($f$) is calculated from the satellite a priori profile using the following equation:

$$f = \frac{c_{S5P}(FTIR\,altitude \rightarrow toa)}{c_{S5P}(S5P\,pixel\,altitude \rightarrow toa)}, \tag{B1}$$

where the numerator is the partial column from the FTIR station altitude to the top of the atmosphere (toa) and the denominator

is the total column from the pixel altitude to the top of the atmosphere. The scaling factor is less than 1 for cases where the satellite pixels are located below the altitude of the FTIR station. In certain cases, where the S5P pixels are above the FTIR station, the scaling factor goes above 1. The scaling factor is applied to the satellite data such that the co-located pairs are on





the same FTIR station altitude. Equation B1 is valid for satellite pixels < station altitude and we use the S5P prior profile. However, in the other case where satellite pixels > station altitude we extrapolate the satellite prior to compensate the small

altitude differences.

The S5P products are adapted to the altitude of the station by either cutting off the scaled mixing ratio profiles at the station altitude (for FTIR station at high altitude locations) or by extending the profile assuming a constant elongation of the mixing ratio up to the station altitude (for case where S5P pixel altitude is above the FTIR station). This method of S5P pixel altitude correction is applied when the satellite and ground-based columns are not calculated between the same boundaries, e.g., S5P

vs. TCCON, and S5P vs. NDACC without extra satellite smoothing.

*Data availability.* The S5P $CH_4$ and CO data used in this study are made available to the S5P Mission Performance Centre via the ESA expert Hub. The data since the release of the CO and $CH_4$ products are available at the Copernicus Open Access Hub (https://scihub.copernicus.eu, last access: 1 December 2020). The FTIR TCCON data are available via the TCCON Data Archive, hosted by CaltechDATA (https://tccondata.org, last access: 1 December 2020). The FTIR NDACC data can be provided in the public NDACC repository (ftp://ftp.cpc.ncep.noaa.gov/ndacc/station/,

last access: 1 December 2020) depending on the site PI's decision.

*Author contributions.* MKS and BL designed the study and produced the validation analysis and results. MKS wrote the first draft of the paper with support of BL. JL and AL have a joint responsibility for the S5P $CH_4$ prototype algorithm and operational processor. JL and TB have a joint responsibility for the S5P CO prototype algorithm and operational processor. All authors contributed in the generation of the data used in this study. All authors read the paper and provided comments.

*Competing interests.* The authors declare that they have no conflict of interest.

*Acknowledgements.* Part of this study has been carried out in the framework of the Copernicus Sentinel-5 Precursor Mission Performance Centre (S5P, MPC) contracted by the European Space Agency (ESA/ESRIN, Contract No. 4000117151/16/I-LG) and supported by the Belgian Federal Science Policy Office (BELSPO), the Royal Belgian Institute for Space Aeronomy (BIRA-IASB). Part of this study has also been supported by the S5P Validation Team (S5PVT) AO project TCCON4S5P (ID #28603, PI Mahesh Kumar Sha, BIRA-IASB) with

national funding from the BELSPO through the ESA ProDEx projects TROVA and TROVA-E2 (PEA 4000116692). This work contains modified Copernicus Sentinel-5 Precursor satellite data (2018–2020) post-processed by BIRA–IASB.

The authors wish to thank the instrument operators and scientists doing the retrieval for the delivery of the ground-based reference data, which were used for the validation study in this paper. We thank Olivier Rasson and José Granville for their support with the creation of S5P overpass level-2 files and download of data used in this study at BIRA-IASB.

The Eureka TCCON and NDACC measurements were made at the Polar Environment Atmospheric Research Laboratory by the Canadian Network for the Detection of Atmospheric Change, primarily supported by the Natural Sciences and Engineering Research Council of Canada





(NSERC), Environment and Climate Change Canada (ECCC), and the Canadian Space Agency. The Toronto NDACC measurements were made at the University of Toronto Atmospheric Observatory, supported by NSERC and ECCC. East Trout Lake is funded through grants from the CFI, ORF, NSERC, and the CSA. Funding for the Edwards TCCON stations is provided by NASA's Earth Science Division. Funding for the Darwin, Lamont, and Pasadena TCCON sites is provided by NASA's OCO-2/-3 projects (grants NNN12AA01C and 80NM0018D0004), while the Park Falls site plus TCCON (the network) is funded through NASA Carbon Cycle Science Program (grant NNX17AE15G). Measurements in Ny-Ålesund and Paramaribo are supported by the Alfred Wegener Institute Potsdam. We thank the AWIPEV station personnel in Ny-Ålesund, Cornelis Becker and Sukarni Mitro (Meteorologische Dienst Suriname) in Paramaribo and the Senate of Bremen for support. The observations at Ny-Ålesund are supported within the DFG, German Research Foundation Project-ID 268020496 - TRR 172 project (AC)3, Arctic Amplification, subproject E02. The Sodankylä site received financial support from the European Space Agency under the grant agreement number ESA-IPL-POE-LG-cl-LE-2015-1129. The Paris TCCON site has received funding from Sorbonne Université, the French research center CNRS, the French space agency CNES, and Région Île-de-France. TCCON sites at Tsukuba, Rikubetsu and Burgos are supported in part by the GOSAT series project. Burgos is supported in part by the Energy Development Corp. Philippines. The TCCON Nicosia site has received additional support from the European Union's Horizon 2020 research and innovation programme under grant agreement No. 856612 and the Cyprus Government, and by the University of Bremen. The Ascension Island TCCON station has been supported by the European Space Agency (ESA) under grant 4000120088/17/I-EF and by the German Bundesministerium für Wirtschaft und Energie (BMWi) under grants 50EE1711C and 50EE1711E. We thank the ESA Ariane Tracking Station at North East Bay, Ascension Island, for hosting and local support. The TCCON and NDACC sites at Réunion Island is operated by the Royal Belgian Institute for Space Aeronomy and local activities supported by LACy/UMR8105 and by OSU-R/UMS3365 – Université de La Réunion. The TCCON Réunion Island site is financially supported since 2014 by the EU project ICOS-Inwire and the ministerial decree for ICOS (FR/35/IC1 to FR/35/IC5).

The NDACC data used in this publication were obtained as part of the Network for the Detection of Atmospheric Composition Change (NDACC) and are publicly available (see http://www.ndacc.org). Rapid delivery data on NDACC is supported for selected sites by the CAMS27 project (https://cams27.aeronomie.be). The National Center for Atmospheric Research is sponsored by the National Science Foundation. The NCAR FTS observation programs at Thule, GR, Boulder, CO and Mauna Loa, HI are supported under contract by the National Aeronautics and Space Administration (NASA). The Thule work is also supported by the NSF Office of Polar Programs (OPP). We wish to thank the Danish Meteorological Institute for support at the Thule site and NOAA for support of the MLO site. The NDACC and TCCON stations Ascension Island, Bremen, Garmisch, Izaña, Ny-Ålesund, Paramaribo and Karlsruhe have been supported by the German Bundesministerium für Wirtschaft und Energie (BMWi) via DLR under grants 50EE1711A to E. The FTIR sites Garmisch, Izaña, Karlsruhe, Kiruna, and Zugspitze have been supported by the Helmholtz Society via the research program ATMO. The NDACC/TCCON Izaña data benefit from the financial support from the Ministerio de Economía y Competitividad from Spain for the project INMENSE (CGL2016-80688-P). We thank the International Foundation High Altitude Research Stations Jungfraujoch and Gornergrat (HFSJG, Bern) for supporting the facilities needed to perform the FTIR observations at Jungfraujoch. The University of Liège contribution has been supported primarily by the Fonds de la Recherche Scientifique - FNRS under grant J.0147.18, by the GAW-CH program of MeteoSwiss as well as by the CAMS project. Emmanuel Mahieu is a senior research associate of the F.R.S.-FNRS. The RUOA-network "Red Universitaria de Observatorios Atmosféricos de la Universidad Nacional Autónoma de México" are acknowledged for the support of the Altzomoni site. Omar Lopez and Delibes Flores Roman are acknowledged for technical support. Alejandro Bezanilla, Noemie Taquet are acknowledged for helping with measurements and data processing. For the Altzomoni site UNAM-DGAPA grants IN111418 & IN107417 and CONACYT 290589 are acknowledged. The St.Petersburg site of the NDACC is operated by Saint Petersburg State University with the financial support provided by the Russian Foundation for Basic Research (grant no. 18-05-00011). The scientific equipment is maintained by the Geomodel research center of SPbU. The



NDACC site at Rikubetsu is operated as parts of the joint research program of the Institute for Space-Earth Environmental Research (ISEE), Nagoya University and supported in part by the GOSAT series project. The Wollongong NDACC site is funded by the Australia Research Council grant DP160101598. Measurements at Lauder and Arrival Heights are core-funded by NIWA through New Zealand's Ministry of Business, Innovation and Employment Strategic Science Investment Fund. D. S. thanks Antarctica New Zealand for providing support for the measurements at Arrival Heights.



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





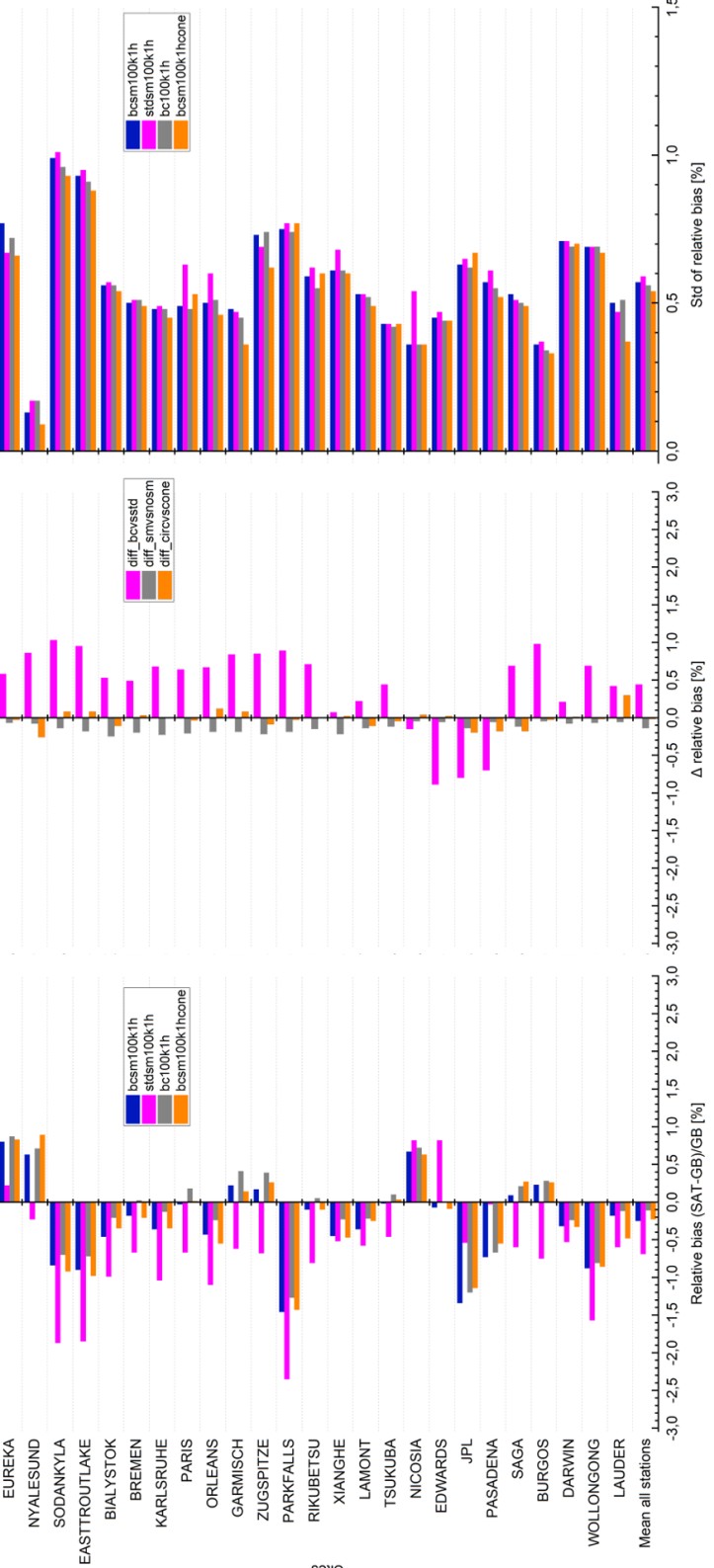

**Figure 1.** S5P XCH$_4$ validation results against TCCON XCH$_4$ data at 25 stations within the period between November 2017 and September 2020. Left: Bar chart of relative bias in percent; middle: Difference of the relative bias in percent; right: Standard deviation of the relative bias for validation cases (stdsm100k1h, bc100k1h, bcsm100k1hcone) against the reference case (bcsm100k1h) in percent. Spatial co-location with radius of 100 km or cone with 1° opening angle along the FTIR line-of-sight and time co-location of ±1 hour around the satellite overpass was used. The stations are sorted with decreasing latitude.

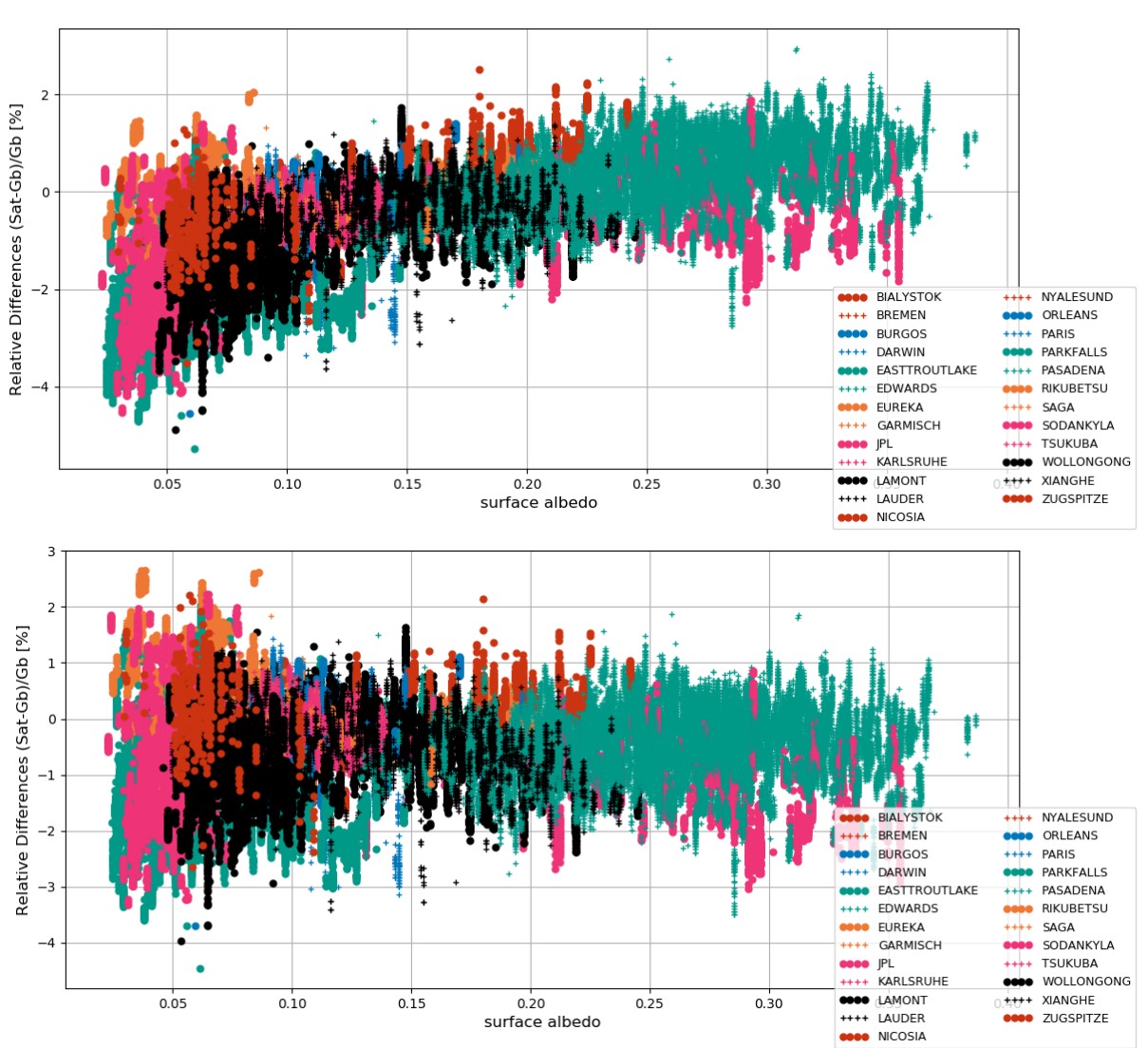

**Figure 2.** Relative biases between co-located S5P (standard $XCH_4$ product - top panel; bias-corrected $XCH_4$ product - bottom panel) and TCCON $XCH_4$ data with a priori alignment are plotted as a function of the surface albedo retrieved by S5P at 25 TCCON stations within the period between November 2017 and September 2020. Spatial co-location with radius of 100 km and time of ±1 hour around the satellite overpass was used.



**Figure 3.** Mosaic plots showing relative biases between co-located S5P (standard XCH$_4$ product - top panel; bias-corrected XCH$_4$ product - bottom panel) and TCCON XCH$_4$ data with a priori alignment at 25 TCCON stations within the period between November 2017 and September 2020. Spatial co-location with radius of 100 km and time of $\pm1$ hour around the satellite overpass was used. The stations are sorted with decreasing latitude.





**Figure 4.** XCH$_4$ time series for all TCCON data (grey), S5P bias-corrected data (light blue), S5P data co-located with TCCON data (blue) and co-located TCCON data with a priori alignment (black) at each site ordered with decreasing latitude. Spatial co-location with radius of 100 km and time of $\pm 1$ hour around the satellite overpass was used.







**Figure 5.** same as Fig. 4



**Figure 6.** Relative difference [(satellite - ground-based)/ground-based] of XCH$_4$ time series for all co-located S5P bias-corrected data and TCCON data with a priori alignment as the reference data at each site ordered with decreasing latitude as in Fig. 4. Spatial co-location with radius of 100 km and time of ±1 hour around the satellite overpass was used.



**Figure 7.** same as Fig. 6





**Figure 8.** Taylor diagram for daily mean differences between S5P and TCCON XCH$_4$ data with a priori alignment: standard (top) and bias-corrected (bottom) S5P XCH$_4$ data. The 25 TCCON stations are sorted with decreasing latitude. The data is within the period between November 2017 and September 2020. Spatial co-location with radius of 100 km and time of ±1 hour around the satellite overpass was used.

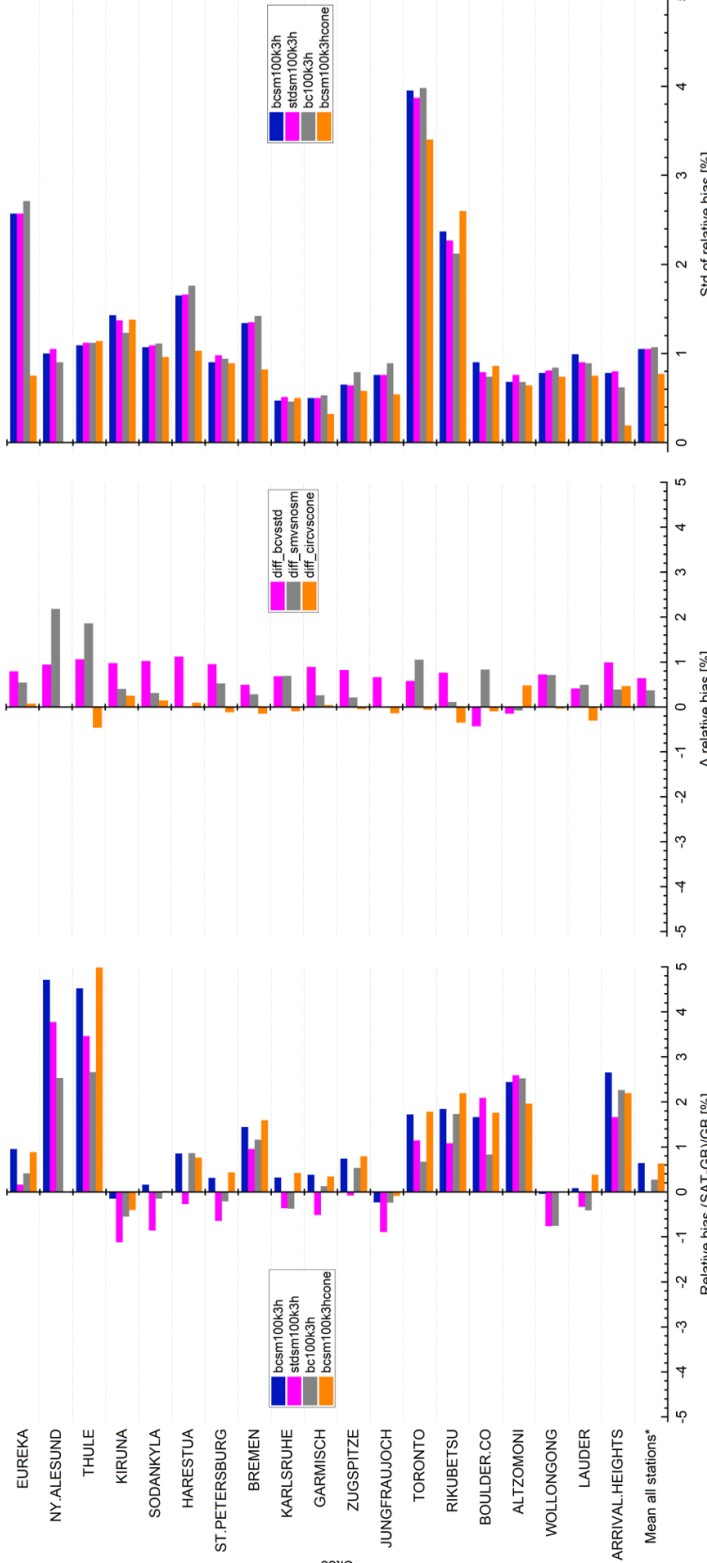

**Figure 9.** S5P XCH$_4$ validation results against NDACC XCH$_4$ data at 19 stations within the period between November 2017 and September 2020. Left: Bar chart of relative bias in percent; right: Standard deviation of the relative bias in percent; middle: Difference of the relative bias for validation cases (stdsm100k3h, bc100k3h, bcsm100k3hcone) against the reference case (bcsm100k3h) in percent. Spatial co-location with radius of 100 km or cone with 1° opening angle along the FTIR line-of-sight and time co-location of ±3 hour around the satellite overpass was used. The stations are sorted with decreasing latitude. The mean of all stations is calculated by excluding outliers which are stations with a low number of co-locations (Ny-Ålesund, Rikubetsu), high scatter in the ground-based data (Toronto), high unexpected bias (Thule, Arrival Heights).



**Figure 10.** Relative biases between co-located S5P (standard XCH$_4$ product - top panel; bias-corrected XCH$_4$ product - bottom panel) and NDACC XCH$_4$ data smoothed with S5P a priori and additionally smoothed with the S5P column averaging kernel are plotted as a function of the surface albedo retrieved by S5P at 19 NDACC stations within the period between November 2017 and September 2020. Spatial co-location with radius of 100 km and time of ±3 hour around the satellite overpass was used.





**Figure 11.** Mosaic plots showing relative biases between co-located S5P (standard XCH$_4$ product - top panel; bias-corrected XCH$_4$ product - bottom panel) and NDACC XCH$_4$ data smoothed with S5P a priori and additionally smoothed with the S5P column averaging kernel at 19 NDACC stations within the period between November 2017 and September 2020. Spatial co-location with radius of 100 km and time of $\pm 3$ hour around the satellite overpass was used. The stations are sorted with decreasing latitude.





**Figure 12.** XCH$_4$ time series for all NDACC data (grey), S5P bias-corrected data (light cyan), S5P data co-located with NDACC data (cyan) and co-located NDACC data smoothed with S5P a priori and additionally smoothed with the S5P column averaging kernel (black) at each site ordered with decreasing latitude. Spatial co-location with radius of 100 km and time of ±3 hour around the satellite overpass was used.







**Figure 13.** same as Fig. 12





**Figure 14.** Relative difference [(satellite - ground-based)/ground-based] of XCH$_4$ time series for all co-located S5P bias-corrected data and NDACC data smoothed with S5P a priori and additionally smoothed with the S5P column averaging kernel as the reference data at each site ordered with decreasing latitude as in Fig. 12. Spatial co-location with radius of 100 km and time of $\pm3$ hour around the satellite overpass was used.





**Figure 15.** same as Fig. 14





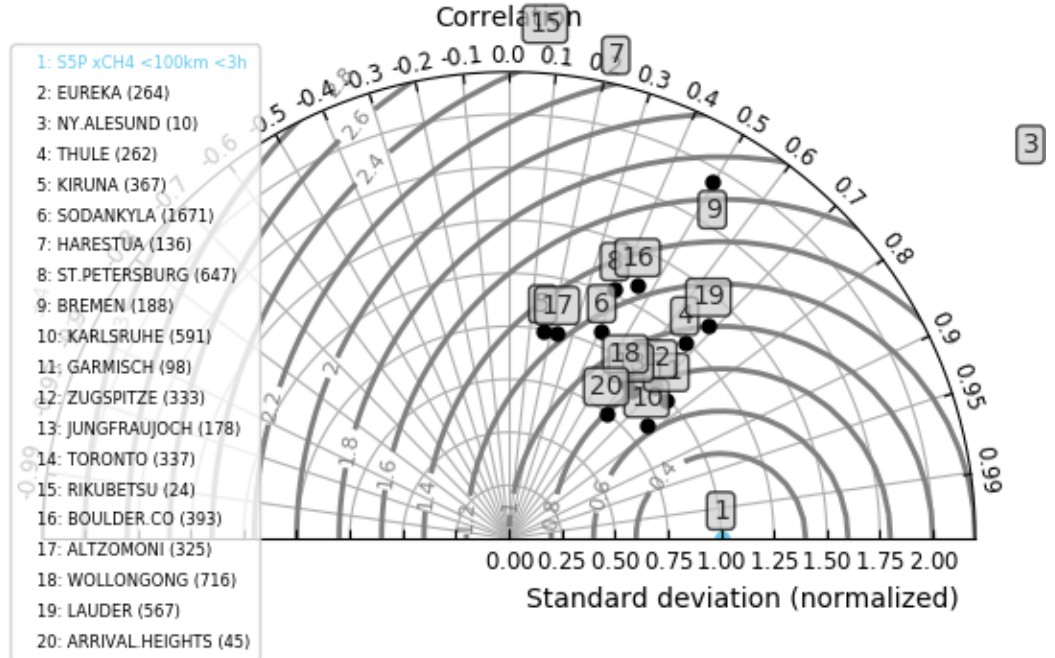

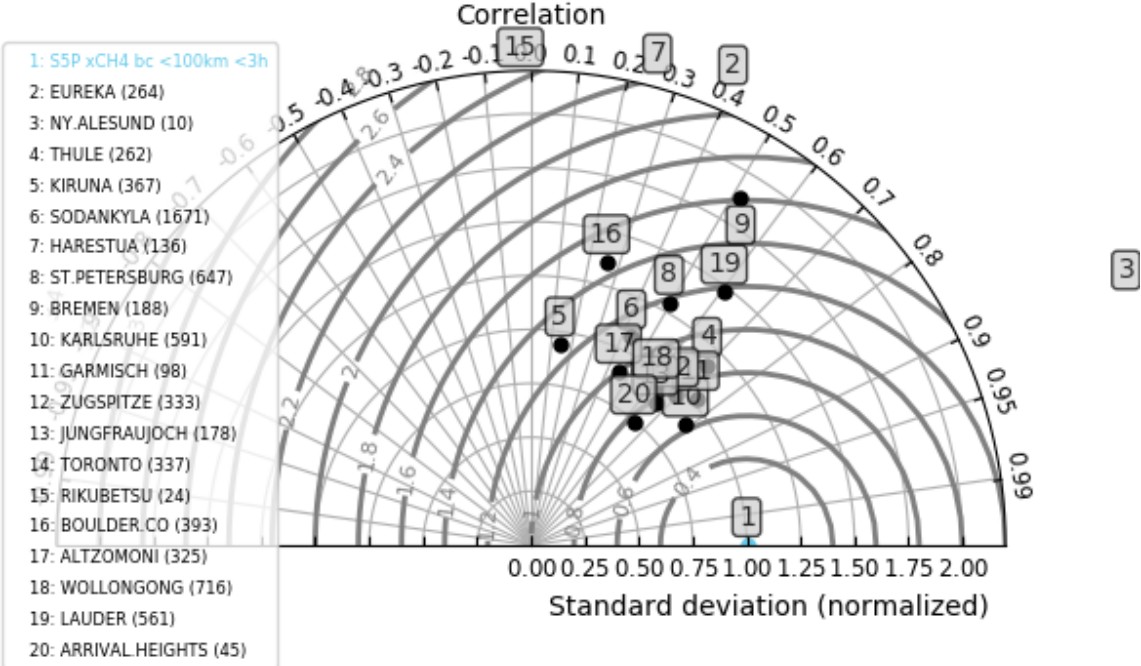

**Figure 16.** Taylor diagram for daily mean differences between S5P and NDACC XCH$_4$ data smoothed with S5P a priori and additionally smoothed with the S5P column averaging kernel: standard (top) and bias-corrected (bottom) S5P XCH$_4$ data. The 19 NDACC stations are sorted with decreasing latitude. The data are within the period between November 2017 and September 2020. Spatial co-location with radius of 100 km and time of ±3 hour around the satellite overpass was used.



**Figure 17.** Relative biases between co-located S5P bias-corrected XCH$_4$ and a priori aligned TCCON XCH$_4$ (top panel) as well as co-located S5P bias-corrected XCH$_4$ and smoothed NDACC XCH$_4$ (bottom panel) are plotted as a function of the S5P measurement solar zenith angles retrieved at the TCCON and NDACC stations within the period between November 2017 and September 2020. Spatial co-location with radius of 100 km and time of $\pm 1$ hour (TCCON) and $\pm 3$ hour (NDACC) around the satellite overpass was used.



**Figure 18.** Relative biases between co-located S5P bias-corrected XCH$_4$ and TCCON XCH$_4$ data with a priori alignment are plotted as a function of the S5P measurement solar zenith angles retrieved at a few TCCON stations within the period between November 2017 and September 2020. Spatial co-location with radius of 100 km and time of $\pm 1$ hour around the satellite overpass was used. The colours represent the different months from January (1) till December (12) of a year.



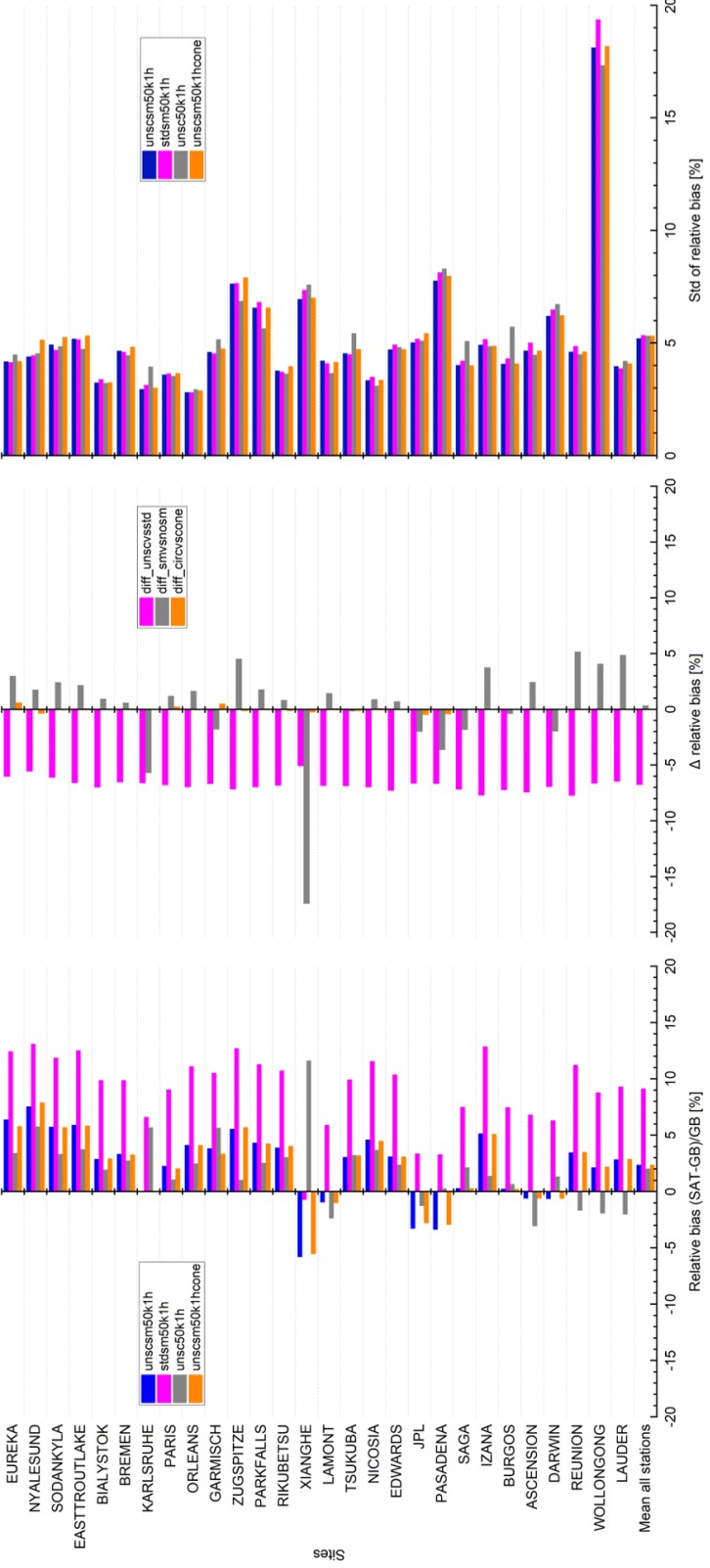

**Figure 19.** S5P XCO validation results against TCCON XCO data at 28 stations within the period between November 2017 and September 2020. Left: Bar chart of relative bias in percent; right: Standard deviation of the relative bias in percent; middle: Difference of the relative bias for validation cases (stdsm50k1h, unsc50k1h, unscsm50k1hcone) against the reference case (unscsm50k1h) in percent. Spatial co-location with radius of 50 km or cone with 1° opening angle along the FTIR line-of-sight and time co-location of ±1 hour around the satellite overpass was used. The stations are sorted with decreasing latitude.



**Figure 20.** XCO time series for all unscaled TCCON data (grey), all S5P data (light red), S5P data co-located with TCCON data (red) and co-located unscaled TCCON data with a priori alignment (black) at each site ordered with decreasing latitude. Spatial co-location with radius of 50 km and time of ±1 hour around the satellite overpass was used.



**Figure 21.** same as Fig. 20





**Figure 22.** Relative difference [(satellite - ground-based)/ground-based] of XCO time series for all co-located S5P data and unscaled TCCON data with a priori alignment as the reference data at each site ordered with decreasing latitude as in Fig. 20. Spatial co-location with radius of 50 km and time of ±1 hour around the satellite overpass was used.





**Figure 23.** same as Fig. 22





**Figure 24.** Mosaic plots showing relative biases between co-located S5P and TCCON XCO data with a priori alignment (standard - top panel; unscaled - bottom panel) at 28 TCCON stations within the period between November 2017 and September 2020. Spatial co-location with radius of 50 km and time of ±1 hour around the satellite overpass was used. The stations are sorted with decreasing latitude.

1: S5P-TCCON xCO unsc OFFL ALL smooth 1hr

2: EUREKA (10716)

3: NYALESUND (9495)

4: SODANKYLA (18723)

5: EASTTROUTLAKE (31198)

6: BIALYSTOK (4698)

7: BREMEN (1399)

8: KARLSRUHE (7990)

9: PARIS (12139)

10: ORLEANS (7462)

11: GARMISCH (5160)

12: ZUGSPITZE (1107)

13: PARKFALLS (16252)

14: RIKUBETSU (5164)

15: XIANGHE (9993)

16: LAMONT (17558)

17: TSUKUBA (10467)

18: NICOSIA (5259)

19: EDWARDS (34554)

20: JPL (4951)

21: PASADENA (30114)

22: SAGA (15288)

23: IZANA (8425)

24: BURGOS (18581)

25: ASCENSION (406)

26: DARWIN (8989)

27: REUNION (3892)

28: WOLLONGONG (10115)

29: LAUDER (29012)

**Figure 25.** Taylor diagram for daily mean differences between S5P and TCCON unscaled XCO data with a priori alignment at 28 TCCON stations within the period between November 2017 and September 2020. Spatial co-location with radius of 50 km and time of $\pm 1$ hour around the satellite overpass was used. The stations are sorted with decreasing latitude.



**Figure 26.** S5P CO column number density plotted around NDACC station at Altzomoni for one sample day. Top-left panel shows all available S5P pixels containing CO data in the overpass file. Top-right panel shows the co-located S5P pixels with 50 km radius selection criterion. Bottom panel shows the co-located S5P pixels with the cone co-location criterion with 1° opening angle of the cone at the highest altitude. The yellow line in the plots represent the line-of-sight of the ground-based FTIR at the time of the satellite overpass over the site.





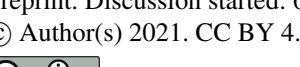

**Figure 27.** S5P CO column validation results against NDACC CO column data at 22 stations within the period between November 2017 and September 2020. Left: Bar chart of relative bias in percent; middle: Difference of the relative bias in percent. right: Standard deviation of the relative bias for validation cases (ALL50k3h, ALLap50k3h, ALLsmcone50k3h) against the reference case (ALLsm50k3h) in percent. Spatial co-location with radius of 50 km or cone with 1° opening angle along the FTIR line-of-sight and time co-location of ±3 hour around the satellite overpass was used. The stations are sorted with decreasing latitude.

**Figure 28.** CO column time series for all NDACC data (grey), all S5P data (light red), S5P data co-located with NDACC data (red) and co-located NDACC data smoothed with S5P a priori and additionally smoothed with the S5P column averaging kernel (black) at each site ordered with decreasing latitude. Spatial co-location with radius of 50 km and time of ±3 hour around the satellite overpass was used.





**Figure 29.** same as Fig. 28



**Figure 30.** Relative difference [(satellite - ground-based)/ground-based] of CO column time series for all co-located S5P data and NDACC data smoothed with S5P a priori and additionally smoothed with the S5P column averaging kernel as the reference data at each site ordered with decreasing latitude as in Fig. 28. Spatial co-location with radius of 50 km and time of ±3 hour around the satellite overpass was used.





**Figure 31.** same as Fig. 30



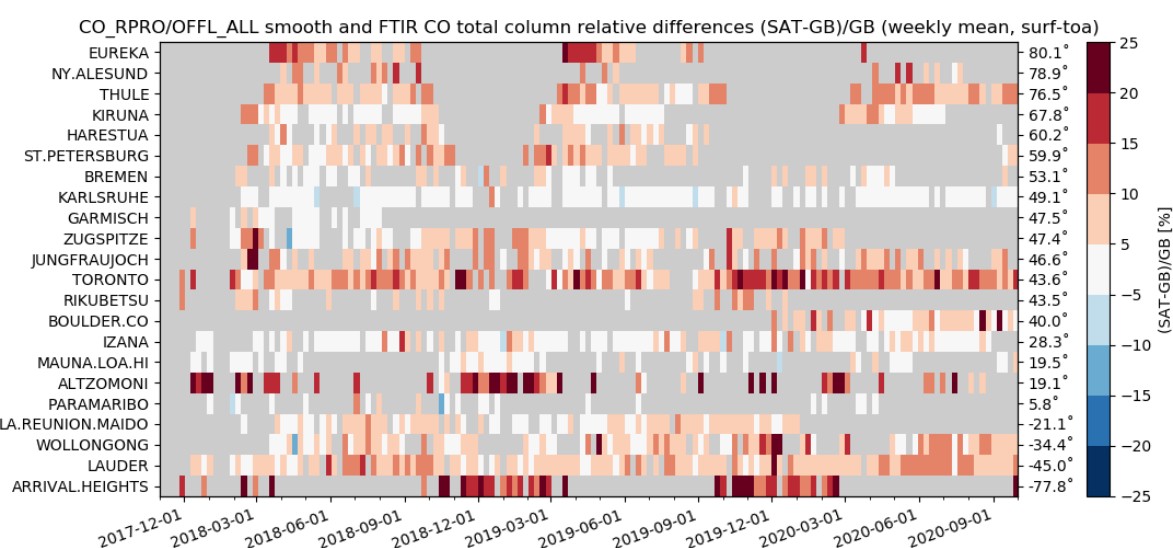

**Figure 32.** Mosaic plots showing relative biases between co-located S5P and NDACC CO column data smoothed with S5P a priori and additionally smoothed with the S5P column averaging kernel at 22 NDACC stations within the period between November 2017 and September 2020. Spatial co-location with radius of 50 km and time of $\pm 3$ hour around the satellite overpass was used. The stations are sorted with decreasing latitude.





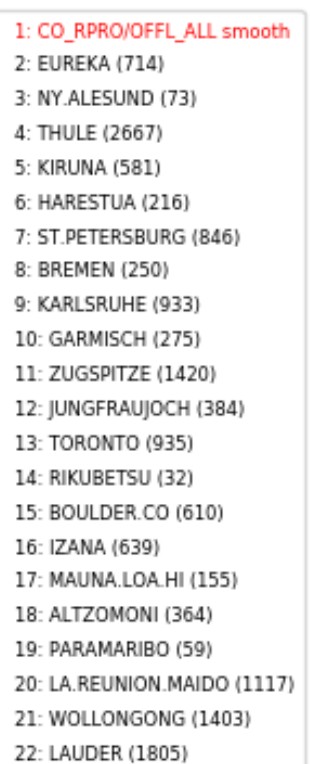

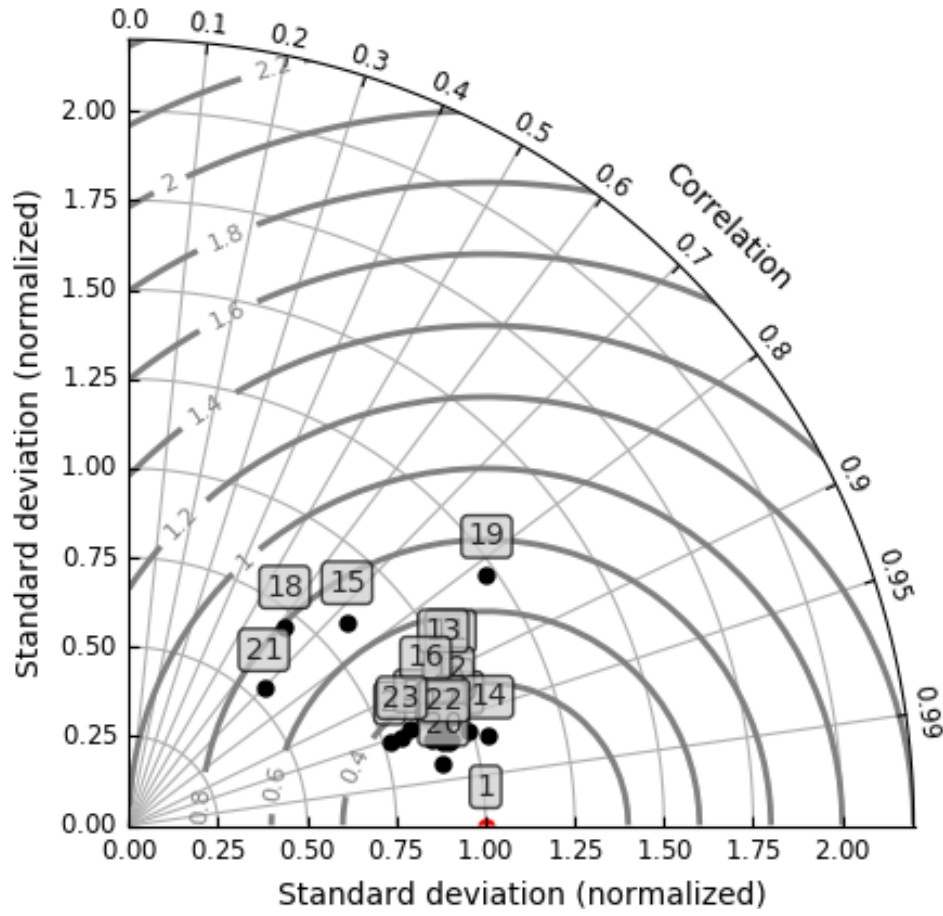

**Figure 33.** Taylor diagram for daily mean differences between S5P and NDACC CO column data smoothed with S5P a priori and additionally smoothed with the S5P column averaging kernel at 22 NDACC stations within the period between November 2017 and September 2020. Spatial co-location with radius of 50 km and time of $\pm3$ hour around the satellite overpass was used. The stations are sorted with decreasing latitude.



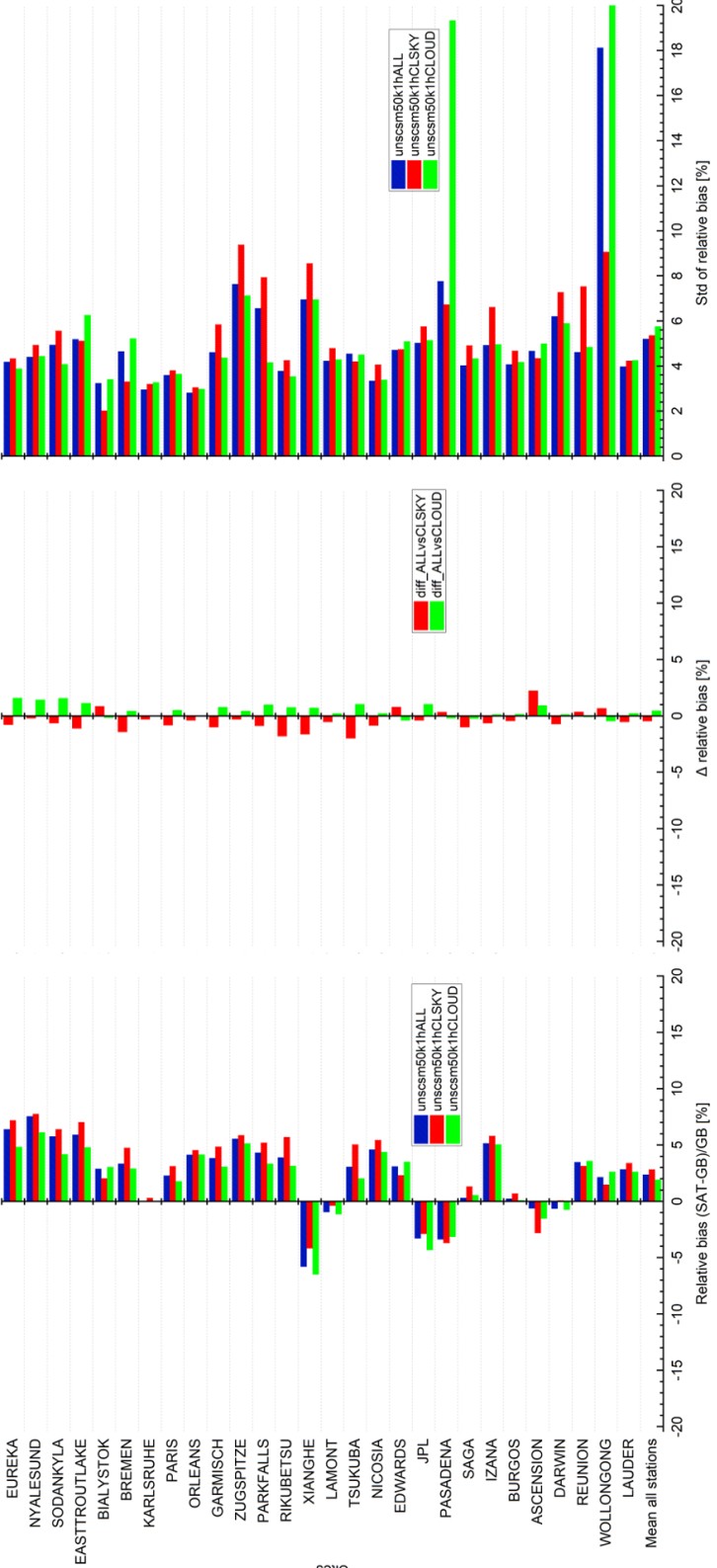

**Figure 34.** S5P XCO (ALL, CLSKY and CLOUD cases) validation results against TCCON XCO data at 28 stations within the period between November 2017 and September 2020. Left: Bar chart of relative bias in percent; right: Standard deviation of the relative bias in percent; middle: Difference of the relative bias for validation cases (unscsm50k1hCLSKY, unscsm50k1hCLOUD) against the reference case (unscsm50k1hALL) in percent. Spatial co-location with radius of 50 km and time co-location of ±1 hour around the satellite overpass was used. The stations are sorted with decreasing latitude.



**Figure 35.** S5P CO column (ALL, CLSKY and CLOUD cases) validation results against NDACC CO column data at 22 stations within the period between November 2017 and September 2020. Left: Bar chart of relative bias in percent; middle: Difference of the relative bias for validation cases (ALLsm50k3hCLSKY, ALLsm50k3hCLOUD) against the reference case (ALLsm50k3h) in percent. Spatial co-location with radius of 50 km and time co-location of ±3 hour around the satellite overpass was used. The stations are sorted with decreasing latitude.





**Figure 36.** Relative biases between co-located S5P XCO and a priori aligned TCCON unscaled XCO (top panel) as well as co-located S5P CO column and smoothed NDACC CO column (bottom panel) are plotted as a function of the S5P measurement solar zenith angles retrieved at the TCCON and NDACC stations within the period between November 2017 and September 2020. Spatial co-location with radius of 50 km and time of ±1 hour (TCCON) and ±3 hour (NDACC) around the satellite overpass was used.

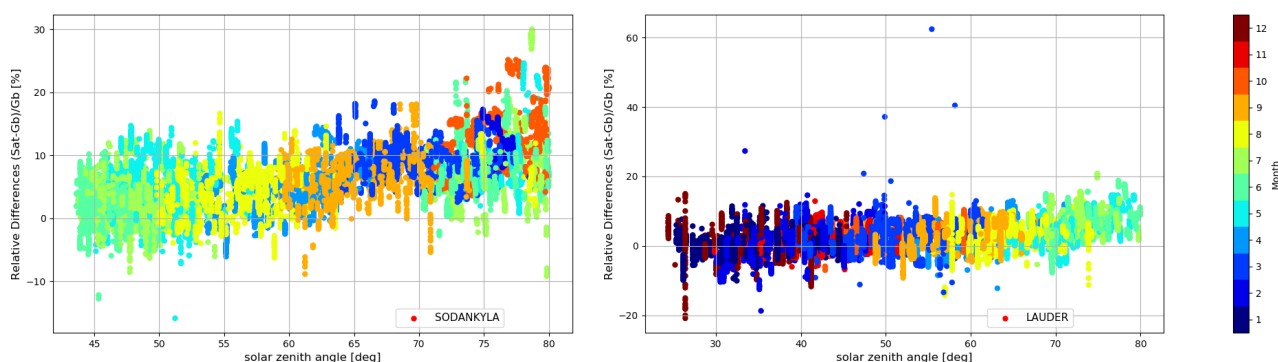

**Figure 37.** Relative biases between co-located S5P XCO and TCCON unscaled XCO data with a priori aligned are plotted as a function of the S5P measurement solar zenith angles retrieved at a few TCCON stations within the period between November 2017 and September 2020. Spatial co-location with radius of 50 km and time of ±1 hour around the satellite overpass was used. The colours represent the different months from January (1) till December (12) of a year.





**Table 1.** S5P CH$_4$ RPRO+OFFL data versions and CO RPRO+OFFL data versions used in the present work.

| Product ID | Stream | Version | In operation from (orbit no., date) | In operation until (orbit no., date) |
|---|---|---|---|---|
| L2_CH$_4$ | RPRO | 01.02.02 | 0657, 2017-11-28 | 5346, 2018-10-25 |
| | | 01.03.01 | 2818, 2018-04-30 | 5832, 2018-11-28 |
| | | 01.03.02 | 2463, 2018-04-04 | 2477, 2018-04-05 |
| | OFFL | 01.02.02 | 5833, 2018-11-28 | 7424, 2019-03-20 |
| | | 01.03.00 | 7425, 2019-03-20 | 7906, 2019-04-23 |
| | | 01.03.01 | 7907, 2019-04-23 | 8814, 2019-06-26 |
| | | 01.03.02 | 8812, 2019-06-26 | current version |
| L2_CO | RPRO | 01.02.02 | 5236, 2018-10-17 | 5346, 2018-10-25 |
| | | 01.03.01 | 2818, 2018-04-30 | 5832, 2018-11-28 |
| | | 01.03.02 | 2463, 2018-04-04 | 2477, 2018-04-05 |
| | OFFL | 01.02.00 | 5346, 2018-10-25 | 5832, 2018-11-28 |
| | | 01.02.02 | 5833, 2018-11-28 | 7424, 2019-03-20 |
| | | 01.03.00 | 7425, 2019-03-20 | 7906, 2019-04-23 |
| | | 01.03.01 | 7907, 2019-04-23 | 8814, 2019-06-26 |
| | | 01.03.02 | 8815, 2019-06-26 | current version |



**Table 2.** List of FTIR stations that are associated to TCCON and contributed to the present work by providing public and rapid delivery data. The stations marked with a star (*) are not yet associated to TCCON but perform observations and data analysis fully compatible with TCCON guidelines. Active dates correspond to the dates for which the measurements were provided from the satellite launch till the present work.

| Station | Latitude | Longitude | Altitude (km a.s.l.) | Active dates | Data reference |
|---------|----------|-----------|----------------------|--------------|----------------|
| EUREKA | 80.05° N | 86.42° W | 0.61 | Nov 2017 - Present | Strong et al. (2019) |
| NY-ÅLESUND | 78.90° N | 11.90° E | 0.02 | Nov 2017 - Present | Notholt et al. (2014b) |
| SODANKYLÄ | 67.37° N | 26.63° E | 0.19 | Nov 2017 - Present | Kivi et al. (2014); Kivi and Heikkinen (2016) |
| EAST TROUT LAKE | 54.35° N | 104.99° W | 0.50 | Nov 2017 - Present | Wunch et al. (2018) |
| BIAŁYSTOK | 53.23° N | 23.05° E | 0.18 | Nov 2017 - Oct 2018 | Deutscher et al. (2019) |
| BREMEN | 53.10° N | 8.85° E | 0.03 | Nov 2017 - Present | Notholt et al. (2014a) |
| KARLSRUHE | 49.10° N | 8.44° E | 0.12 | Nov 2017 - Present | Hase et al. (2015) |
| PARIS | 48.85° N | 2.36° E | 0.06 | Nov 2017 - Present | Té et al. (2014) |
| ORLÉANS | 47.97° N | 2.11° E | 0.13 | Nov 2017 - Present | Warneke et al. (2019) |
| GARMISCH | 47.48° N | 11.06° E | 0.74 | Nov 2017 - Present | Sussmann and Rettinger (2018a) |
| ZUGSPITZE | 47.42° N | 10.98° E | 2.96 | Nov 2017 - Present | Sussmann and Rettinger (2018b) |
| PARK FALLS | 45.95° N | 90.27° W | 0.44 | Nov 2017 - Present | Wennberg et al. (2017) |
| RIKUBETSU | 43.46° N | 143.77° E | 0.38 | Nov 2017 - Present | Morino et al. (2018c) |
| XIANGHE* | 39.75° N | 116.96° E | 0.05 | Nov 2017 - Present | Yang et al. (2019) |
| LAMONT | 36.60° N | 97.49° W | 0.32 | Nov 2017 - Present | Wennberg et al. (2016b) |
| TSUKUBA | 36.05° N | 140.12° E | 0.03 | Nov 2017 - Present | Morino et al. (2018a) |
| NICOSIA* | 35.14° N | 33.38° E | 0.19 | Aug 2019 - Present | Petri et al. (2019) |
| EDWARDS | 34.96° N | 117.88° W | 0.70 | May 2018 - Present | Iraci et al. (2016) |
| JPL | 34.20° N | 118.18° W | 0.39 | Nov 2017 - May 2018 | Wennberg et al. (2016a) |
| PASADENA | 34.14° N | 118.13° W | 0.23 | Nov 2017 - Present | Wennberg et al. (2015) |
| SAGA | 33.24° N | 130.29° E | 0.01 | Nov 2017 - Present | Kawakami et al. (2014) |
| IZAÑA | 28.30° N | 16.50° W | 2.37 | Nov 2017 - Present | Blumenstock et al. (2017) |
| BURGOS | 18.53° N | 120.65° E | 0.04 | Nov 2017 - Present | Morino et al. (2018b); Velazco et al. (2017) |
| ASCENSION | 7.92° S | 14.33° W | 0.01 | Nov 2017 - Present | Feist et al. (2014) |
| DARWIN | 12.46° S | 130.93° E | 0.04 | Nov 2017 - Present | Griffith et al. (2014a) |
| RÉUNION | 20.90° S | 55.49° E | 0.09 | Nov 2017 - Present | De Mazière et al. (2017) |
| WOLLONGONG | 34.41° S | 150.88° E | 0.03 | Nov 2017 - Present | Griffith et al. (2014b) |
| LAUDER | 45.04° S | 169.68° E | 0.61 | Nov 2017 - Present | Sherlock et al. (2014); Pollard et al. (2019) |





**Table 3.** List of FTIR stations that are associated to NDACC-IRWG and contributed to the present work by providing public and rapid delivery data. The stations marked with a star (*) are not yet associated to NDACC but perform observations and data analysis fully compatible with NDACC guidelines. The location of the stations and the teams involved are indicated for the respective stations.

| Station | Latitude | Longitude | Altitude (km a.s.l.) | Active dates | Teams |
|---|---|---|---|---|---|
| EUREKA | 80.05° N | 86.42° W | 0.61 | Nov 2017 - Present | U. of Toronto; Batchelor et al. (01 Jul. 2009) |
| NY-ÅLESUND | 78.90° N | 11.90° E | 0.02 | Nov 2017 - Present | U. of Bremen |
| THULE | 76.52° N | 68.77° W | 0.22 | Nov 2017 - Present | NCAR; Hannigan et al. (01 Sep. 2009) |
| KIRUNA | 67.84° N | 20.40° E | 0.42 | Nov 2017 - Present | KIT-ASF; IRF Kiruna |
| SODANKYLÄ* | 67.37° N | 26.63° E | 0.19 | Nov 2017 - Present | FMI; BIRA-IASB |
| HARESTUA | 60.20° N | 10.80° E | 0.60 | Nov 2017 - Present | Chalmers |
| ST.PETERSBURG | 59.88° N | 29.83° E | 0.02 | Nov 2017 - Present | SPbU; Makarova et al. (2015) |
| BREMEN | 53.10° N | 8.85° E | 0.03 | Nov 2017 - Present | U. of Bremen |
| KARLSRUHE* | 49.10° N | 8.44° E | 0.12 | Nov 2017 - Present | KIT-ASF |
| GARMISCH* | 47.48° N | 11.06° E | 0.74 | Nov 2017 - Present | KIT-IFU |
| ZUGSPITZE | 47.42° N | 10.98° E | 2.96 | Nov 2017 - Present | KIT-IFU |
| JUNGFRAUJOCH | 46.55° N | 7.98° E | 3.58 | Nov 2017 - Present | U. of Liège |
| TORONTO | 43.60° N | 79.36° W | 0.17 | Nov 2017 - Present | U. of Toronto; Wiacek et al. (01 Mar. 2007) |
| RIKUBETSU | 43.46° N | 143.77° E | 0.38 | Nov 2017 - Present | Nagoya U.; NIES |
| BOULDER | 40.04° N | 105.24° W | 1.61 | Nov 2017 - Present | NCAR; Ortega et al. (2019) |
| IZAÑA | 28.30° N | 16.50° W | 2.37 | Nov 2017 - Present | AEMET; KIT-ASF |
| MAUNA LOA | 19.54° N | 155.57° W | 3.40 | Nov 2017 - Present | NCAR |
| ALTZOMONI | 19.12° N | 98.66° W | 3.98 | Nov 2017 - Present | UNAM |
| PARAMARIBO | 5.81° N | 55.21° W | 0.03 | Nov 2017 - Present | U. of Bremen |
| LARÉUNION.MAÏDO | 21.08° S | 55.38° E | 2.16 | Nov 2017 - Present | BIRA-IASB |
| WOLLONGONG | 34.41° S | 150.88° E | 0.03 | Nov 2017 - Present | U. of Wollongong |
| LAUDER | 45.04° S | 169.68° E | 0.37 | Nov 2017 - Present | NIWA |
| ARRIVAL HEIGHTS | 77.82° S | 166.65° E | 0.20 | Nov 2017 - Present | NIWA |





**Table 4.** S5P XCH$_4$ validation results against TCCON XCH$_4$ data at 25 stations for the period between November 2017 and September 2020. Spatial co-location with radius of 100 km or cone with 1° opening angle along the FTIR line-of-sight and time co-location of ±1 hour around the satellite overpass was used. TCCON station (column 1) are sorted according to the decreasing latitude (column 2). The column with title 'No.' represents the number of co-located measurements, column title 'Std' represents the standard deviation of the time series of the ground-based data relative to the standard deviation of the time series of the S5P data, column title 'Corr' represents the correlation coefficient between the S5P and the reference ground-based data, column title 'Rel diff bias' represents the relative difference ((SAT-GB)/GB) bias in percent and column title 'Rel diff std' represents the standard deviation of the relative bias in percent.

| Site | Lat | S5P bc XCH$_4$ smooth 100 km 1 hr | | | | | S5P XCH$_4$ smooth 100 km 1 hr | | | | S5P bc XCH$_4$ 100 km 1 hr | | | | S5P bc XCH$_4$ smooth cone 100 km 1 hr | | | | |
|---|---|---|---|---|---|---|---|---|---|---|---|---|---|---|---|---|---|---|---|
| | | No. | Std | Corr | Rel diff bias (%) | Rel diff std (%) | Std | Corr | Rel diff bias (%) | Rel diff std (%) | Std | Corr | Rel diff bias (%) | Rel diff std (%) | No. | Std | Corr | Rel diff bias (%) | Rel diff std (%) |
| EUREKA | 80 | 1384 | 1 | 0.62 | 0.8 | 0.77 | 0.8 | 0.79 | 0.22 | 0.67 | 1 | 0.66 | 0.87 | 0.72 | 648 | 0.9 | 0.74 | 0.83 | 0.66 |
| NY-ÅLESUND | 78.9 | 113 | 1 | 0.97 | 0.63 | 0.13 | 1 | 0.95 | -0.23 | 0.17 | 0.9 | 0.95 | 0.71 | 0.17 | 11 | 2.8 | -0.56 | 0.89 | 0.09 |
| SODANKYLÄ | 67.4 | 5915 | 0.9 | 0.37 | -0.84 | 0.99 | 0.9 | 0.36 | -1.87 | 1.01 | 0.9 | 0.4 | -0.7 | 0.96 | 4681 | 0.9 | 0.42 | -0.92 | 0.93 |
| EAST TROUT LAKE | 54.3 | 12302 | 0.8 | 0.64 | -0.9 | 0.93 | 0.8 | 0.62 | -1.85 | 0.95 | 0.9 | 0.67 | -0.72 | 0.91 | 10358 | 0.9 | 0.65 | -0.98 | 0.88 |
| BIALYSTOK | 53.2 | 1821 | 0.7 | 0.46 | -0.46 | 0.56 | 0.8 | 0.37 | -0.99 | 0.57 | 0.8 | 0.49 | -0.21 | 0.56 | 1623 | 0.8 | 0.41 | -0.35 | 0.54 |
| BREMEN | 53.1 | 997 | 1 | 0.62 | -0.18 | 0.5 | 1.1 | 0.6 | -0.67 | 0.51 | 1.1 | 0.61 | 0.02 | 0.51 | 950 | 1 | 0.6 | -0.21 | 0.49 |
| KARLSRUHE | 49.1 | 4592 | 0.8 | 0.79 | -0.36 | 0.48 | 0.8 | 0.82 | -1.04 | 0.49 | 0.8 | 0.79 | -0.13 | 0.48 | 4114 | 0.8 | 0.81 | -0.35 | 0.45 |
| PARIS | 48.8 | 4999 | 1 | 0.67 | -0.03 | 0.49 | 0.8 | 0.55 | -0.67 | 0.63 | 1 | 0.69 | 0.18 | 0.48 | 4370 | 0.9 | 0.67 | 0.01 | 0.53 |
| ORLÉANS | 48 | 3689 | 0.8 | 0.58 | -0.43 | 0.5 | 0.6 | 0.62 | -1.1 | 0.6 | 0.8 | 0.56 | -0.24 | 0.51 | 3446 | 0.9 | 0.57 | -0.55 | 0.46 |
| GARMISCH | 47.5 | 1785 | 0.7 | 0.69 | 0.22 | 0.48 | 0.7 | 0.7 | -0.62 | 0.47 | 0.8 | 0.73 | 0.41 | 0.45 | 734 | 0.9 | 0.68 | 0.14 | 0.36 |
| ZUGSPITZE | 47.4 | 402 | 1.3 | 0.6 | 0.17 | 0.73 | 1.3 | 0.63 | -0.68 | 0.69 | 1.3 | 0.64 | 0.39 | 0.74 | 207 | 1.4 | 0.3 | 0.26 | 0.62 |
| PARK FALLS | 45.9 | 7201 | 0.8 | 0.7 | -1.46 | 0.75 | 0.8 | 0.67 | -2.35 | 0.77 | 0.8 | 0.71 | -1.27 | 0.74 | 6671 | 0.8 | 0.7 | -1.43 | 0.77 |
| RIKUBETSU | 43.5 | 2466 | 0.8 | 0.83 | -0.1 | 0.59 | 0.9 | 0.8 | -0.81 | 0.62 | 0.9 | 0.85 | 0.05 | 0.55 | 2206 | 0.8 | 0.83 | -0.1 | 0.6 |
| XIANGHE | 39.8 | 4395 | 1 | 0.87 | -0.45 | 0.61 | 1.2 | 0.81 | -0.52 | 0.68 | 1 | 0.87 | -0.23 | 0.61 | 3706 | 0.9 | 0.88 | -0.47 | 0.6 |
| LAMONT | 36.6 | 9608 | 0.9 | 0.84 | -0.36 | 0.53 | 0.9 | 0.84 | -0.58 | 0.53 | 0.9 | 0.84 | -0.22 | 0.52 | 8053 | 0.9 | 0.85 | -0.25 | 0.49 |
| TSUKUBA | 36 | 4655 | 0.9 | 0.84 | -0.02 | 0.43 | 1 | 0.84 | -0.46 | 0.43 | 1 | 0.86 | 0.1 | 0.42 | 4494 | 0.9 | 0.86 | 0.03 | 0.43 |
| NICOSIA | 35.1 | 1247 | 1 | 0.76 | 0.67 | 0.36 | 0.6 | 0.8 | 0.82 | 0.54 | 1 | 0.77 | 0.72 | 0.36 | 1158 | 1 | 0.73 | 0.63 | 0.36 |
| EDWARDS | 35 | 24350 | 0.9 | 0.85 | -0.07 | 0.45 | 1 | 0.83 | 0.82 | 0.47 | 0.9 | 0.85 | -0.01 | 0.44 | 22987 | 0.9 | 0.86 | -0.09 | 0.44 |
| JPL | 34.2 | 3233 | 1.1 | 0.33 | -1.34 | 0.63 | 1 | 0.4 | -0.54 | 0.65 | 1.1 | 0.34 | -1.2 | 0.62 | 1954 | 0.9 | 0.27 | -1.14 | 0.67 |
| PASADENA | 34.1 | 20646 | 0.9 | 0.79 | -0.73 | 0.57 | 0.9 | 0.74 | -0.03 | 0.61 | 0.9 | 0.81 | -0.67 | 0.55 | 15868 | 0.9 | 0.85 | -0.55 | 0.52 |
| SAGA | 33.2 | 2539 | 0.9 | 0.79 | 0.09 | 0.53 | 0.9 | 0.79 | -0.6 | 0.51 | 0.9 | 0.81 | 0.21 | 0.5 | 1943 | 0.9 | 0.83 | 0.27 | 0.49 |
| BURGOS | 18.5 | 417 | 1 | 0.76 | 0.23 | 0.36 | 0.9 | 0.77 | -0.75 | 0.37 | 0.9 | 0.77 | 0.28 | 0.34 | 269 | 0.8 | 0.76 | 0.26 | 0.33 |
| DARWIN | -12.5 | 3598 | 0.5 | 0.23 | -0.32 | 0.71 | 0.5 | 0.3 | -0.53 | 0.71 | 0.5 | 0.27 | -0.24 | 0.69 | 3267 | 0.5 | 0.24 | -0.33 | 0.7 |
| WOLLONGONG | -34.4 | 4765 | 0.9 | 0.55 | -0.88 | 0.69 | 0.8 | 0.56 | -1.57 | 0.69 | 0.8 | 0.54 | -0.81 | 0.69 | 4677 | 0.9 | 0.56 | -0.86 | 0.67 |
| LAUDER | -45 | 6708 | 1 | 0.8 | -0.18 | 0.5 | 0.9 | 0.84 | -0.6 | 0.47 | 0.9 | 0.8 | -0.12 | 0.51 | 1771 | 0.9 | 0.92 | -0.48 | 0.37 |
| Mean all stations | | | 0.9 | 0.68 | -0.25 | 0.57 | 0.9 | 0.68 | -0.69 | 0.59 | 0.9 | 0.69 | -0.11 | 0.56 | | 1 | 0.62 | -0.23 | 0.54 |




**Table 5.** S5P XCH$_4$ validation results against NDACC XCH$_4$ data at 19 stations for the period between November 2017 and September 2020. Spatial co-location with radius of 100 km or cone with 1° opening angle along the FTIR line-of-sight and time co-location of ±3 hour around the satellite overpass was used. NDACC station (column 1) are sorted according to the decreasing latitude (column 2). The column with title 'No.' represents the number of co-located measurements, column title 'Std' represents the standard deviation of the time series of the ground-based data relative to the standard deviation of the time series of the S5P data, column title 'Corr' represents the correlation coefficient between the S5P and the reference ground-based data, column title 'Rel diff bias' represents the relative difference ((SAT-GB)/GB) bias in percent and column title 'Rel diff std' represents the standard deviation of the relative bias in percent.

| Site | Lat | S5P bc XCH$_4$ smooth 100 km 3 hr | | | | | S5P XCH$_4$ smooth 100 km 3 hr | | | | S5P bc XCH$_4$ 100 km 3 hr | | | | S5P bc XCH$_4$ smooth cone 100 km 3 hr | | | | |
|---|---|---|---|---|---|---|---|---|---|---|---|---|---|---|---|---|---|---|---|
| | | No. | Std | Corr | Rel diff bias (%) | Rel diff std (%) | Std | Corr | Rel diff bias (%) | Rel diff std (%) | Std | Corr | Rel diff bias (%) | Rel diff std (%) | No. | Std | Corr | Rel diff bias (%) | Rel diff std (%) |
| EUREKA | 80.1 | 264 | 2.5 | 0.37 | 0.95 | 2.57 | 3.4 | 0.35 | 0.16 | 2.57 | 2.5 | 0.35 | 0.41 | 2.71 | 67 | 1.3 | 0.74 | 0.88 | 0.75 |
| NY-ÅLESUND | 78.9 | 10 | 3 | 0.92 | 4.71 | 1 | 3 | 0.82 | 3.77 | 1.05 | 2.9 | 0.94 | 2.53 | 0.9 | | | | | |
| THULE | 76.5 | 262 | 1.2 | 0.7 | 4.52 | 1.09 | 1.2 | 0.67 | 3.46 | 1.12 | 1.2 | 0.68 | 2.66 | 1.12 | 65 | 0.9 | 0.61 | 4.98 | 1.14 |
| KIRUNA | 67.8 | 367 | 0.9 | 0.14 | -0.15 | 1.43 | 1 | 0.16 | -1.12 | 1.37 | 0.8 | 0.3 | -0.55 | 1.23 | 256 | 1 | 0.2 | -0.4 | 1.38 |
| SODANKYLÄ | 67.4 | 1671 | 1.1 | 0.43 | 0.16 | 1.07 | 1.1 | 0.41 | -0.86 | 1.09 | 1.2 | 0.48 | -0.15 | 1.11 | 1281 | 1.1 | 0.46 | 0.02 | 0.96 |
| HARESTUA | 60.2 | 136 | 2.5 | 0.24 | 0.85 | 1.65 | 2.5 | 0.21 | -0.27 | 1.66 | 2.7 | 0.24 | 0.86 | 1.76 | 74 | 1.5 | 0.02 | 0.76 | 1.03 |
| ST.PETERSBURG | 59.9 | 647 | 1.3 | 0.49 | 0.31 | 0.9 | 1.3 | 0.39 | -0.64 | 0.98 | 1.4 | 0.51 | -0.21 | 0.94 | 529 | 1.3 | 0.5 | 0.43 | 0.89 |
| BREMEN | 53.1 | 188 | 1.9 | 0.52 | 1.44 | 1.34 | 1.9 | 0.5 | 0.95 | 1.35 | 2.1 | 0.56 | 1.16 | 1.42 | 182 | 1.1 | 0.68 | 1.59 | 0.82 |
| KARLSRUHE | 49.1 | 591 | 0.9 | 0.79 | 0.32 | 0.47 | 0.8 | 0.77 | -0.36 | 0.51 | 0.9 | 0.79 | -0.37 | 0.46 | 485 | 0.9 | 0.77 | 0.42 | 0.5 |
| GARMISCH | 47.5 | 98 | 1 | | 0.38 | 0.5 | 1 | 0.75 | -0.51 | 0.5 | 1.4 | 0.84 | 0.12 | 0.53 | 18 | 1.2 | 0.88 | 0.34 | 0.32 |
| ZUGSPITZE | 47.4 | 333 | 1 | 0.69 | 0.74 | 0.65 | 1 | 0.69 | -0.08 | 0.64 | 1.3 | 0.69 | 0.53 | 0.79 | 53 | 1.3 | 0.61 | 0.79 | 0.58 |
| JUNGFRAUJOCH | 46.6 | 178 | 0.9 | 0.66 | -0.23 | 0.76 | 0.9 | 0.64 | -0.89 | 0.76 | 1.1 | 0.64 | -0.24 | 0.89 | 31 | 1 | 0.74 | -0.09 | 0.54 |
| TORONTO | 43.6 | 337 | 3.5 | -0.13 | 1.72 | 3.95 | 3.6 | -0.08 | 1.14 | 3.87 | 3.6 | -0.12 | 0.67 | 3.98 | 235 | 2.5 | -0.18 | 1.78 | 3.4 |
| RIKUBETSU | 43.5 | 24 | 2.4 | -0.02 | 1.84 | 2.37 | 2.6 | 0.07 | 1.08 | 2.27 | 2.2 | 0.02 | 1.73 | 2.12 | 18 | 2.4 | -0.01 | 2.19 | 2.6 |
| BOULDER.CO | 40 | 393 | 1.4 | 0.26 | 1.66 | 0.9 | 1.3 | 0.45 | 2.09 | 0.79 | 1.1 | 0.34 | 0.83 | 0.74 | 289 | 1.4 | 0.42 | 1.76 | 0.86 |
| ALTZOMONI | 19.1 | 325 | 0.9 | 0.45 | 2.44 | 0.68 | 1 | 0.23 | 2.59 | 0.76 | 1 | 0.52 | 2.52 | 0.68 | 188 | 1.1 | 0.48 | 1.96 | 0.64 |
| WOLLONGONG | -34.4 | 716 | 0.9 | 0.62 | -0.04 | 0.78 | 0.9 | 0.6 | -0.76 | 0.81 | 1 | 0.57 | -0.75 | 0.84 | 603 | 0.9 | 0.65 | 0 | 0.74 |
| LAUDER | -45 | 561 | 1.5 | 0.61 | 0.08 | 0.99 | 1.4 | 0.68 | -0.33 | 0.9 | 1.3 | 0.61 | -0.41 | 0.89 | 41 | 1.4 | 0.71 | 0.38 | 0.75 |
| ARRIVAL.HEIGHTS | -77.8 | 45 | 0.7 | 0.64 | 2.65 | 0.78 | 0.7 | 0.62 | 1.66 | 0.8 | 1 | 0.81 | 2.26 | 0.62 | 2 | 0.3 | -1 | 2.19 | 0.19 |
| Mean all stations* | | | 1.3 | 0.5 | 0.64 | 1.05 | 1.4 | 0.49 | 0 | 1.05 | 1.4 | 0.53 | 0.27 | 1.07 | | 1.2 | 0.56 | 0.63 | 0.77 |

* The mean of all stations is calculated by excluding outliers which are stations with a low number of co-locations (Ny-Ålesund, Arrival Height, Rikubetsu), high scatter in the ground-based data (Toronto), high unexpected bias (Thule).

**Table 6.** S5P XCO validation results against TCCON XCO data at 28 stations for the period between November 2017 and September 2020. Spatial co-location with radius of 50 km or cone with 1° opening angle along the FTIR line-of-sight and time co-location of ±1 hour around the satellite overpass was used. TCCON station (column 1) are sorted according to the decreasing latitude (column 2). The column with title 'No.' represents the number of co-located measurements, column title 'Std' represents the standard deviation of the time series of the ground-based data relative to the standard deviation of the time series of the S5P data, column title 'Corr' represents the correlation coefficient between the S5P and the reference ground-based data, column title 'Rel diff bias' represents the relative difference ((SAT-GB)/GB) bias in percent and column title 'Rel diff std' represents the standard deviation of the relative bias in percent.

| Site | Lat | TCCON unsc XCO smooth 50 km 1 hr | | | | | TCCON std XCO smooth 50 km 1 hr | | | | TCCON unsc XCO 50 km 1 hr | | | | TCCON unsc XCO smooth cone 50 km 1 hr | | | | |
|---|---|---|---|---|---|---|---|---|---|---|---|---|---|---|---|---|---|---|---|
| | | No. | Std | Corr | Rel diff bias (%) | Rel diff std (%) | Std | Corr | Rel diff bias (%) | Rel diff std (%) | Std | Corr | Rel diff bias (%) | Rel diff std (%) | No. | Std | Corr | Rel diff bias (%) | Rel diff std (%) |
| EUREKA | 80 | 10716 | 0.8 | 0.95 | 6.4 | 4.18 | 0.8 | 0.95 | 12.44 | 4.15 | 0.8 | 0.94 | 3.41 | 4.48 | 10247 | 0.8 | 0.95 | 5.8 | 4.19 |
| NY-ÅLESUND | 78.9 | 9495 | 0.9 | 0.97 | 7.54 | 4.4 | 0.8 | 0.97 | 13.1 | 4.45 | 0.8 | 0.97 | 5.78 | 4.54 | 8569 | 0.9 | 0.96 | 7.91 | 5.14 |
| SODANKYLÄ | 67.4 | 18723 | 0.9 | 0.95 | 5.75 | 4.93 | 0.9 | 0.96 | 11.88 | 4.7 | 0.9 | 0.96 | 3.33 | 4.85 | 17395 | 0.9 | 0.95 | 5.71 | 5.26 |
| EAST TROUT LAKE | 54.3 | 31198 | 1 | 0.94 | 5.92 | 5.18 | 0.9 | 0.94 | 12.53 | 5.16 | 0.9 | 0.94 | 3.75 | 4.74 | 29856 | 1 | 0.94 | 5.85 | 5.33 |
| BIAŁYSTOK | 53.2 | 4698 | 0.9 | 0.97 | 2.88 | 3.24 | 0.9 | 0.97 | 9.89 | 3.39 | 0.9 | 0.97 | 1.94 | 3.22 | 4687 | 0.9 | 0.97 | 2.94 | 3.25 |
| BREMEN | 53.1 | 1399 | 1 | 0.92 | 3.33 | 4.65 | 0.9 | 0.93 | 9.87 | 4.61 | 1 | 0.93 | 2.74 | 4.46 | 1394 | 1 | 0.92 | 3.27 | 4.83 |
| KARLSRUHE | 49.1 | 7990 | 0.9 | 0.97 | -0.02 | 2.95 | 0.9 | 0.97 | 6.61 | 3.14 | 0.8 | 0.95 | 5.69 | 3.95 | 7905 | 0.9 | 0.97 | -0.03 | 3.01 |
| PARIS | 48.8 | 12139 | 1 | 0.93 | 2.27 | 3.59 | 1 | 0.94 | 9.06 | 3.64 | 1 | 0.93 | 1.06 | 3.53 | 11960 | 1 | 0.93 | 2.04 | 3.66 |
| ORLÉANS | 48 | 7462 | 0.9 | 0.97 | 4.13 | 2.81 | 0.9 | 0.97 | 11.12 | 2.81 | 0.9 | 0.97 | 2.49 | 2.94 | 7408 | 0.9 | 0.97 | 4.11 | 2.89 |
| GARMISCH | 47.5 | 5160 | 1 | 0.92 | 3.83 | 4.6 | 0.9 | 0.93 | 10.52 | 4.53 | 0.9 | 0.9 | 5.65 | 5.16 | 5091 | 1 | 0.91 | 3.35 | 4.75 |
| ZUGSPITZE | 47.4 | 1107 | 1.2 | 0.82 | 5.55 | 7.63 | 1.1 | 0.83 | 12.72 | 7.66 | 1.1 | 0.82 | 1.03 | 6.86 | 1093 | 1.2 | 0.81 | 5.71 | 7.91 |
| PARK FALLS | 45.9 | 16252 | 0.9 | 0.94 | 4.32 | 6.56 | 0.9 | 0.95 | 11.3 | 6.82 | 1 | 0.94 | 2.54 | 5.65 | 16101 | 0.9 | 0.94 | 4.26 | 6.57 |
| RIKUBETSU | 43.5 | 5164 | 1.1 | 0.97 | 3.89 | 3.77 | 1 | 0.97 | 10.73 | 3.72 | 1 | 0.96 | 3.06 | 3.63 | 5163 | 1.1 | 0.96 | 4.05 | 3.97 |
| XIANGHE | 39.8 | 9993 | 0.9 | 0.95 | -5.81 | 6.95 | 0.8 | 0.95 | -0.73 | 7.35 | 0.9 | 0.95 | 11.62 | 7.59 | 9934 | 0.9 | 0.95 | -5.56 | 7.01 |
| LAMONT | 36.6 | 17558 | 1 | 0.93 | -0.95 | 4.22 | 0.9 | 0.95 | 5.91 | 4.1 | 1 | 0.95 | -2.39 | 3.66 | 17478 | 1 | 0.94 | -1.03 | 4.16 |
| TSUKUBA | 36 | 10467 | 1 | 0.95 | 3.05 | 4.54 | 0.9 | 0.96 | 9.94 | 4.5 | 0.9 | 0.93 | 3.24 | 5.43 | 10434 | 1 | 0.95 | 3.21 | 4.73 |
| NICOSIA | 35.1 | 5259 | 0.9 | 0.95 | 4.59 | 3.34 | 0.8 | 0.95 | 11.57 | 3.5 | 1 | 0.96 | 3.69 | 3.1 | 5242 | 0.9 | 0.95 | 4.5 | 3.36 |
| EDWARDS | 35 | 34554 | 0.9 | 0.94 | 3.1 | 4.71 | 0.9 | 0.94 | 10.39 | 4.93 | 0.9 | 0.93 | 2.39 | 4.82 | 34510 | 0.9 | 0.94 | 3.11 | 4.72 |
| JPL | 34.2 | 4951 | 1.3 | 0.89 | -3.3 | 5.02 | 1.2 | 0.89 | 3.36 | 5.18 | 1.2 | 0.89 | -1.28 | 5.1 | 4875 | 1.3 | 0.87 | -2.81 | 5.43 |
| PASADENA | 34.1 | 30114 | 0.8 | 0.84 | -3.38 | 7.76 | 0.7 | 0.85 | 3.29 | 8.14 | 0.8 | 0.83 | 0.27 | 8.3 | 30002 | 0.8 | 0.84 | -2.95 | 7.98 |
| SAGA | 33.2 | 15288 | 1 | 0.97 | 0.3 | 4.02 | 1 | 0.98 | 7.5 | 4.21 | 0.9 | 0.96 | 2.14 | 5.08 | 15280 | 1 | 0.97 | 0.29 | 4 |
| IZAÑA | 28.3 | 8425 | 1 | 0.88 | 5.15 | 4.92 | 0.9 | 0.9 | 12.87 | 5.17 | 1 | 0.86 | 1.4 | 4.85 | 8425 | 1 | 0.88 | 5.11 | 4.88 |
| BURGOS | 18.5 | 18581 | 0.9 | 0.97 | 0.24 | 4.07 | 0.8 | 0.97 | 7.47 | 4.31 | 0.9 | 0.94 | 0.65 | 5.72 | 18580 | 0.9 | 0.97 | 0.22 | 4.09 |
| ASCENSION | -7.9 | 406 | 1.2 | 0.63 | -0.62 | 4.66 | 1.1 | 0.63 | 6.83 | 5.01 | 1.2 | 0.63 | -3.07 | 4.47 | 406 | 1.2 | 0.63 | -0.62 | 4.66 |
| DARWIN | -12.5 | 8989 | 1 | 0.92 | -0.65 | 6.2 | 1 | 0.92 | 6.3 | 6.49 | 0.9 | 0.9 | 1.34 | 6.73 | 8989 | 1 | 0.92 | -0.63 | 6.23 |
| RÉUNION | -20.9 | 3892 | 0.9 | 0.96 | 3.47 | 4.61 | 0.9 | 0.96 | 11.22 | 4.87 | 0.9 | 0.96 | -1.69 | 4.5 | 3892 | 0.9 | 0.96 | 3.5 | 4.62 |
| WOLLONGONG | -34.4 | 10115 | 0.8 | 0.82 | 2.14 | 18.12 | 0.8 | 0.82 | 8.8 | 19.37 | 0.8 | 0.82 | -1.94 | 17.33 | 10108 | 0.8 | 0.82 | 2.2 | 18.18 |
| LAUDER | -45 | 29012 | 1 | 0.97 | 2.84 | 3.97 | 0.9 | 0.98 | 9.3 | 3.87 | 1 | 0.97 | -2.02 | 4.2 | 28853 | 1 | 0.97 | 2.88 | 4.09 |
| Mean all stations | | | 1 | 0.92 | 2.36 | 5.2 | 0.9 | 0.93 | 9.14 | 5.35 | 0.9 | 0.92 | 2.03 | 5.32 | | 1 | 0.92 | 2.37 | 5.32 |





**Table 7.** S5P CO column validation results against NDACC CO column data at 22 stations for the period between November 2017 and September 2020. Spatial co-location with radius of 50 km or cone with 1° opening angle along the FTIR line-of-sight and time co-location of ±3 hour around the satellite overpass was used. NDACC station (column 1) are sorted according to the decreasing latitude (column 2). The column with title 'No.' represents the number of co-located measurements, column title 'Std' represents the standard deviation of the time series of the ground-based data relative to the standard deviation of the time series of the S5P data, column title 'Corr' represents the correlation coefficient between the S5P and the reference ground-based data, column title 'Rel diff bias' represents the relative difference ((SAT-GB)/GB) bias in percent and column title 'Rel diff std' represents the standard deviation of the relative bias in percent.

| Site | Lat | NDACC CO smooth 50 km 3 hr | | | | | NDACC CO 50 km 3 hr | | | | NDACC CO ap 50 km 3 hr | | | | NDACC CO smooth cone 50 km 3 hr | | | | |
|---|---|---|---|---|---|---|---|---|---|---|---|---|---|---|---|---|---|---|---|
| | | No. | Std | Corr | Rel diff bias (%) | Rel diff std (%) | Std | Corr | Rel diff bias (%) | Rel diff std (%) | Std | Corr | Rel diff bias (%) | Rel diff std (%) | No. | Std | Corr | Rel diff bias (%) | Rel diff std (%) |
| EUREKA | 80.1 | 714 | 0.8 | 0.95 | 12.96 | 4.56 | 0.8 | 0.96 | 11.64 | 3.72 | 0.8 | 0.96 | 11.65 | 3.79 | 636 | 0.8 | 0.95 | 12.18 | 4.83 |
| NY-ÅLESUND | 78.9 | 73 | 0.9 | 0.96 | 11.72 | 3.82 | 0.8 | 0.98 | 8.19 | 3.88 | 0.9 | 0.97 | 2.03 | 3.01 | 72 | 0.9 | 0.95 | 12.13 | 4.34 |
| THULE | 76.5 | 2667 | 0.9 | 0.95 | 9.44 | 4.79 | 0.9 | 0.96 | 7.67 | 3.67 | 0.9 | 0.95 | 4.28 | 4.04 | 2516 | 0.9 | 0.95 | 9.02 | 5.06 |
| KIRUNA | 67.8 | 581 | 0.8 | 0.95 | 3.51 | 4.77 | 0.9 | 0.97 | 4.38 | 3.68 | 0.9 | 0.97 | 2.42 | 3.68 | 563 | 0.8 | 0.94 | 3.59 | 5.07 |
| HARESTUA | 60.2 | 216 | 0.9 | 0.97 | 6.7 | 3.73 | 0.9 | 0.97 | 4.8 | 3.22 | 0.9 | 0.97 | 4.58 | 3.22 | 192 | 0.9 | 0.96 | 7.23 | 4.26 |
| ST.PETERSBURG | 59.9 | 846 | 0.9 | 0.96 | 6.67 | 3.87 | 0.9 | 0.95 | 5.08 | 3.74 | 0.9 | 0.95 | 3.99 | 3.75 | 845 | 0.9 | 0.95 | 6.53 | 3.95 |
| BREMEN | 53.1 | 250 | 0.9 | 0.97 | 5.12 | 3.31 | 0.9 | 0.95 | 5.93 | 3.81 | 0.9 | 0.97 | 4.38 | 3.05 | 244 | 0.9 | 0.97 | 5.16 | 3.36 |
| KARLSRUHE | 49.1 | 933 | 1 | 0.96 | -0.55 | 3.24 | 0.9 | 0.96 | 6.34 | 3.47 | 1 | 0.97 | -0.14 | 3.07 | 913 | 1 | 0.96 | -0.41 | 3.46 |
| GARMISCH | 47.5 | 275 | 0.9 | 0.95 | 1.26 | 4.25 | 0.9 | 0.97 | 2.2 | 3.37 | 1.1 | 0.97 | -7.18 | 3.06 | 267 | 1 | 0.94 | 0.87 | 4.34 |
| ZUGSPITZE | 47.4 | 1420 | 1 | 0.9 | 6.48 | 5.4 | 1.1 | 0.88 | 7.99 | 5.73 | 1.1 | 0.88 | 8.11 | 5.62 | 1326 | 1 | 0.89 | 7.32 | 6.08 |
| JUNGFRAUJOCH | 46.6 | 384 | 1 | 0.94 | 8.09 | 4.43 | 1 | 0.92 | 11.34 | 4.87 | 1 | 0.91 | 10.92 | 4.97 | 377 | 1 | 0.92 | 8.79 | 4.93 |
| TORONTO | 43.6 | 935 | 1 | 0.9 | 11.82 | 7.1 | 0.9 | 0.92 | 9.73 | 5.24 | 0.9 | 0.92 | 7.47 | 5.08 | 894 | 1 | 0.89 | 11.99 | 7.47 |
| RIKUBETSU | 43.5 | 32 | 1 | 0.97 | 7.77 | 3.73 | 1 | 0.96 | 3.42 | 3.3 | 1 | 0.96 | 2.12 | 3.37 | 32 | 1 | 0.97 | 7.42 | 3.62 |
| BOULDER.CO | 40 | 610 | 0.8 | 0.73 | 8.29 | 11.89 | 0.9 | 0.76 | 3.62 | 9.87 | 1 | 0.76 | -2.96 | 9.72 | 601 | 0.8 | 0.77 | 7.65 | 11.51 |
| IZAÑA | 28.3 | 639 | 0.9 | 0.92 | 2.54 | 4.24 | 1 | 0.9 | 3.7 | 4.36 | 1 | 0.9 | 3.66 | 4.35 | 638 | 0.9 | 0.92 | 2.52 | 4.37 |
| MAUNA.LOA.HI | 19.5 | 155 | 0.9 | 0.97 | 2.65 | 3.22 | 1 | 0.96 | 3.17 | 3.41 | 1 | 0.97 | 2.93 | 3.38 | 153 | 0.9 | 0.96 | 2.5 | 3.64 |
| ALTZOMONI | 19.1 | 364 | 0.7 | 0.62 | 20.6 | 10.73 | 0.8 | 0.56 | 20.32 | 11.2 | 0.8 | 0.56 | 20.35 | 11.2 | 358 | 0.6 | 0.63 | 20.26 | 11.79 |
| PARAMARIBO | 5.8 | 59 | 1.2 | 0.82 | 0.88 | 6 | 1.1 | 0.77 | -1.12 | 5.64 | 1.1 | 0.76 | -2.24 | 5.84 | 57 | 1.2 | 0.81 | 0.98 | 6.05 |
| LA.RÉUNION.MAIDO | -21.1 | 1117 | 0.9 | 0.98 | 6.44 | 4.62 | 1 | 0.98 | 5.96 | 4.54 | 1 | 0.98 | 5.78 | 4.58 | 1055 | 0.9 | 0.97 | 6.69 | 5.53 |
| WOLLONGONG | -34.4 | 1403 | 0.5 | 0.7 | 9.17 | 23.01 | 0.7 | 0.7 | 5.04 | 21.41 | 0.7 | 0.7 | 0.82 | 20.92 | 1382 | 0.5 | 0.71 | 8.84 | 21.88 |
| LAUDER | -45 | 1805 | 0.9 | 0.96 | 7.82 | 4.54 | 0.9 | 0.97 | 5.93 | 4.27 | 0.9 | 0.97 | 1.07 | 3.97 | 1776 | 0.9 | 0.96 | 7.9 | 4.74 |
| ARRIVAL.HEIGHTS | -77.8 | 174 | 0.8 | 0.95 | 18.3 | 5.63 | 0.8 | 0.96 | 15.68 | 4.93 | 0.9 | 0.95 | 11.99 | 5.34 | 138 | 0.8 | 0.93 | 19.54 | 7.13 |
| Mean all stations | | | 0.9 | 0.91 | 7.62 | 5.95 | 0.9 | 0.91 | 6.86 | 5.51 | 0.9 | 0.9 | 4.37 | 5.41 | | 0.9 | 0.9 | 7.67 | 6.25 |



**Table 8.** Validation of S5P XCO ALL, CLSKY and CLOUD data with TCCON XCO data at 28 stations for the period between November 2017 and September 2020. Spatial co-location with radius of 50 km and time co-location of $\pm 1$ hour around the satellite overpass was used. TCCON station (column 1) are sorted according to the decreasing latitude (column 2). The column with title 'No.' represents the number of co-located measurements, column title 'Std' represents the standard deviation of the time series of the ground-based data relative to the standard deviation of the time series of the S5P data, column title 'Corr' represents the correlation coefficient between the S5P and the reference ground-based data, column title 'Rel diff bias' represents the relative difference ((SAT-GB)/GB) bias in percent and column title 'Rel diff std' represents the standard deviation of the relative bias in percent.

| Sites | Lat | TCCON unsc XCO smooth 50 km 1 hr ALL | | | | | TCCON unsc XCO smooth 50 km 1 hr CLSKY | | | | | TCCON unsc XCO smooth 50 km 1 hr CLOUD | | | | |
|---|---|---|---|---|---|---|---|---|---|---|---|---|---|---|---|---|
| | | No. | Std | Corr | Rel diff bias (%) | Rel diff std (%) | No. | Std | Corr | Rel diff bias (%) | Rel diff std (%) | No. | Std | Corr | Rel diff bias (%) | Rel diff std (%) |
| EUREKA | 80 | 10716 | 0.8 | 0.95 | 6.4 | 4.18 | 9421 | 0.8 | 0.94 | 7.19 | 4.33 | 6019 | 0.9 | 0.92 | 4.82 | 3.88 |
| NY-ÅLESUND | 78.9 | 9495 | 0.9 | 0.97 | 7.54 | 4.4 | 7854 | 0.9 | 0.96 | 7.75 | 4.93 | 4637 | 0.9 | 0.97 | 6.12 | 4.43 |
| SODANKYLÄ | 67.4 | 18723 | 0.9 | 0.95 | 5.75 | 4.93 | 12972 | 0.9 | 0.94 | 6.4 | 5.56 | 7633 | 0.9 | 0.96 | 4.19 | 4.08 |
| EAST TROUT LAKE | 54.3 | 31198 | 1 | 0.94 | 5.92 | 5.18 | 18415 | 1 | 0.92 | 7.03 | 5.11 | 16283 | 0.9 | 0.91 | 4.79 | 6.26 |
| BIAŁYSTOK | 53.2 | 4698 | 0.9 | 0.97 | 2.88 | 3.24 | 1122 | 1 | 0.98 | 2.04 | 2.02 | 4110 | 0.9 | 0.97 | 3.04 | 3.41 |
| BREMEN | 53.1 | 1399 | 1 | 0.92 | 3.33 | 4.65 | 829 | 1 | 0.97 | 4.75 | 3.31 | 976 | 1 | 0.9 | 2.9 | 5.22 |
| KARLSRUHE | 49.1 | 7990 | 0.9 | 0.97 | -0.02 | 2.95 | 3885 | 0.9 | 0.96 | 0.29 | 3.2 | 5948 | 0.9 | 0.96 | 0.03 | 3.27 |
| PARIS | 48.8 | 12139 | 1 | 0.93 | 2.27 | 3.59 | 5703 | 1.1 | 0.92 | 3.11 | 3.8 | 8627 | 1 | 0.93 | 1.77 | 3.64 |
| ORLÉANS | 48 | 7462 | 0.9 | 0.97 | 4.13 | 2.81 | 3229 | 1 | 0.96 | 4.54 | 3.05 | 5976 | 0.9 | 0.97 | 4.14 | 2.98 |
| GARMISCH | 47.5 | 5160 | 1 | 0.92 | 3.83 | 4.6 | 3158 | 0.9 | 0.87 | 4.84 | 5.84 | 3609 | 0.9 | 0.93 | 3.07 | 4.36 |
| ZUGSPITZE | 47.4 | 1107 | 1.2 | 0.82 | 5.55 | 7.63 | 741 | 1.2 | 0.7 | 5.86 | 9.38 | 861 | 1.1 | 0.84 | 5.13 | 7.12 |
| PARK FALLS | 45.9 | 16252 | 0.9 | 0.94 | 4.32 | 6.56 | 9013 | 1 | 0.93 | 5.2 | 7.93 | 10553 | 0.9 | 0.94 | 3.33 | 4.15 |
| RIKUBETSU | 43.5 | 5164 | 1.1 | 0.97 | 3.89 | 3.77 | 2775 | 1 | 0.94 | 5.7 | 4.25 | 4398 | 1 | 0.97 | 3.15 | 3.53 |
| XIANGHE | 39.8 | 9993 | 0.9 | 0.95 | -5.81 | 6.95 | 4511 | 0.9 | 0.93 | -4.18 | 8.55 | 6655 | 0.9 | 0.95 | -6.51 | 6.95 |
| LAMONT | 36.6 | 17558 | 1 | 0.93 | -0.95 | 4.22 | 6513 | 0.9 | 0.88 | -0.4 | 4.79 | 15128 | 1 | 0.94 | -1.16 | 4.28 |
| TSUKUBA | 36 | 10467 | 1 | 0.95 | 3.05 | 4.54 | 5651 | 0.9 | 0.95 | 5.04 | 4.2 | 7887 | 1 | 0.95 | 2.03 | 4.5 |
| NICOSIA | 35.1 | 5259 | 0.9 | 0.95 | 4.59 | 3.34 | 4248 | 0.9 | 0.94 | 5.44 | 4.06 | 4744 | 0.9 | 0.95 | 4.37 | 3.38 |
| EDWARDS | 35 | 34554 | 0.9 | 0.94 | 3.1 | 4.71 | 17050 | 0.9 | 0.94 | 2.32 | 4.74 | 32387 | 0.9 | 0.93 | 3.5 | 5.09 |
| JPL | 34.2 | 4951 | 1.3 | 0.89 | -3.3 | 5.02 | 2626 | 1.2 | 0.84 | -2.9 | 5.75 | 3897 | 1.2 | 0.86 | -4.32 | 5.14 |
| PASADENA | 34.1 | 30114 | 0.8 | 0.84 | -3.38 | 7.76 | 19375 | 1 | 0.83 | -3.71 | 6.73 | 25831 | 0.4 | 0.57 | -3.16 | 19.33 |
| SAGA | 33.2 | 15288 | 1 | 0.97 | 0.3 | 4.02 | 7428 | 1 | 0.95 | 1.31 | 4.91 | 13487 | 1 | 0.97 | 0.54 | 4.33 |
| IZAÑA | 28.3 | 8425 | 1 | 0.88 | 5.15 | 4.92 | 3541 | 1 | 0.78 | 5.8 | 6.61 | 7707 | 1 | 0.88 | 5.04 | 4.96 |
| BURGOS | 18.5 | 18581 | 0.9 | 0.97 | 0.24 | 4.07 | 8442 | 0.9 | 0.94 | 0.69 | 4.67 | 17951 | 0.9 | 0.97 | 0.09 | 4.17 |
| ASCENSION | -7.9 | 406 | 1.2 | 0.63 | -0.62 | 4.66 | 126 | 2 | 0.88 | -2.83 | 4.34 | 383 | 1.1 | 0.59 | -1.54 | 4.99 |
| DARWIN | -12.5 | 8989 | 1 | 0.92 | -0.65 | 6.2 | 3866 | 1.1 | 0.91 | 0.08 | 7.27 | 8092 | 1 | 0.93 | -0.76 | 5.89 |
| RÉUNION | -20.9 | 3892 | 0.9 | 0.96 | 3.47 | 4.61 | 1055 | 0.9 | 0.9 | 3.13 | 7.53 | 3892 | 0.9 | 0.95 | 3.57 | 4.84 |
| WOLLONGONG | -34.4 | 10115 | 0.8 | 0.82 | 2.14 | 18.12 | 7141 | 1 | 0.89 | 1.48 | 9.06 | 7228 | 0.8 | 0.77 | 2.62 | 22.63 |
| LAUDER | -45 | 29012 | 1 | 0.97 | 2.84 | 3.97 | 19196 | 1 | 0.96 | 3.39 | 4.23 | 22116 | 1 | 0.97 | 2.62 | 4.25 |
| Mean all stations | | | 1 | 0.92 | 2.36 | 5.2 | | 1 | 0.91 | 2.83 | 5.36 | | 0.9 | 0.91 | 1.91 | 5.75 |





**Table 9.** Validation of S5P CO column ALL, CLSKY and CLOUD data with NDACC CO column data at 22 stations for the period between November 2017 and September 2020. Spatial co-location with radius of 50 km and time co-location of ±3 hour around the satellite overpass was used. NDACC station (column 1) are sorted according to the decreasing latitude (column 2). The column with title 'No.' represents the number of co-located measurements, column title 'Std' represents the standard deviation of the time series of the ground-based data relative to the standard deviation of the time series of the S5P data, column title 'Corr' represents the correlation coefficient between the S5P and the reference ground-based data, column title 'Rel diff bias' represents the relative difference ((SAT-GB)/GB) bias in percent and column title 'Rel diff std' represents the standard deviation of the relative bias in percent.

| Sites | Lat | NDACC CO smooth 50 km 3 hr ALL | | | | | NDACC CO smooth 50 km 3 hr CLSKY | | | | | NDACC CO smooth 50 km 1 hr CLOUD | | | | |
|---|---|---|---|---|---|---|---|---|---|---|---|---|---|---|---|---|
| | | No. | Std | Corr | Rel diff bias (%) | Rel diff std (%) | No. | Std | Corr | Rel diff bias (%) | Rel diff std (%) | No. | Std | Corr | Rel diff bias (%) | Rel diff std (%) |
| EUREKA | 80.1 | 714 | 0.8 | 0.95 | 12.96 | 4.56 | 597 | 0.8 | 0.95 | 12.23 | 4.08 | 300 | 0.8 | 0.96 | 11.72 | 4.64 |
| NY-ÅLESUND | 78.9 | 73 | 0.9 | 0.96 | 11.72 | 3.82 | 72 | 0.8 | 0.95 | 11.04 | 4.24 | 56 | 0.9 | 0.97 | 11.7 | 3.9 |
| THULE | 76.5 | 2667 | 0.9 | 0.95 | 9.44 | 4.79 | 2388 | 0.9 | 0.95 | 9.43 | 4.67 | 1609 | 0.9 | 0.94 | 7.8 | 5.14 |
| KIRUNA | 67.8 | 581 | 0.8 | 0.95 | 3.51 | 4.77 | 500 | 0.8 | 0.94 | 4.19 | 4.83 | 403 | 0.8 | 0.94 | 2.52 | 5.05 |
| HARESTUA | 60.2 | 216 | 0.9 | 0.97 | 6.7 | 3.73 | 159 | 0.9 | 0.95 | 6.47 | 4.45 | 126 | 1 | 0.97 | 6.81 | 3.69 |
| ST.PETERSBURG | 59.9 | 846 | 0.9 | 0.96 | 6.67 | 3.87 | 744 | 0.9 | 0.94 | 6.4 | 4.3 | 654 | 0.9 | 0.94 | 6.76 | 4.52 |
| BREMEN | 53.1 | 250 | 0.9 | 0.97 | 5.12 | 3.31 | 164 | 0.9 | 0.96 | 5.98 | 3.69 | 163 | 0.9 | 0.97 | 5.23 | 3.69 |
| KARLSRUHE | 49.1 | 933 | 1 | 0.96 | -0.55 | 3.24 | 506 | 1 | 0.96 | 0.19 | 3.27 | 795 | 1 | 0.96 | -0.8 | 3.42 |
| GARMISCH | 47.5 | 275 | 0.9 | 0.95 | 1.26 | 4.25 | 105 | 0.8 | 0.94 | 0.97 | 5.3 | 247 | 1 | 0.95 | 1.37 | 4.33 |
| ZUGSPITZE | 47.4 | 1420 | 1 | 0.9 | 6.48 | 5.4 | 984 | 1 | 0.89 | 6.25 | 6.01 | 992 | 1 | 0.93 | 6.67 | 5.63 |
| JUNGFRAUJOCH | 46.6 | 384 | 1 | 0.94 | 8.09 | 4.43 | 306 | 1 | 0.92 | 7 | 4.92 | 310 | 0.9 | 0.95 | 9.01 | 4.79 |
| TORONTO | 43.6 | 935 | 1 | 0.9 | 11.82 | 7.1 | 400 | 1 | 0.9 | 10.27 | 6.32 | 868 | 1 | 0.89 | 12.46 | 7.73 |
| RIKUBETSU | 43.5 | 32 | 1 | 0.97 | 7.77 | 3.73 | 22 | 1 | 0.96 | 7.36 | 3.8 | 26 | 1.1 | 0.96 | 9.11 | 5 |
| BOULDER.CO | 40 | 610 | 0.8 | 0.73 | 8.29 | 11.89 | 323 | 0.9 | 0.47 | 9.74 | 15.73 | 563 | 0.7 | 0.66 | 9.11 | 16.33 |
| IZAÑA | 28.3 | 639 | 0.9 | 0.92 | 2.54 | 4.24 | 261 | 1 | 0.85 | 2.67 | 5.85 | 597 | 0.9 | 0.93 | 2.54 | 4.17 |
| MAUNA.LOA.HI | 19.5 | 155 | 0.9 | 0.97 | 2.65 | 3.22 | 81 | 0.9 | 0.88 | 2.1 | 6.11 | 145 | 0.9 | 0.96 | 2.52 | 3.64 |
| ALTZOMONI | 19.1 | 364 | 0.7 | 0.62 | 20.6 | 10.73 | 212 | 0.7 | 0.73 | 22.05 | 13.44 | 338 | 0.8 | 0.65 | 19.51 | 10.3 |
| PARAMARIBO | 5.8 | 59 | 1.2 | 0.82 | 0.88 | 6 | | | | | | 59 | 1.2 | 0.82 | 1.01 | 5.98 |
| LA.RÉUNION.MAIDO | -21.1 | 1117 | 0.9 | 0.98 | 6.44 | 4.62 | 265 | 0.9 | 0.96 | 7.92 | 6.61 | 1011 | 0.9 | 0.98 | 6.57 | 4.64 |
| WOLLONGONG | -34.4 | 1403 | 0.5 | 0.7 | 9.17 | 23.01 | 1007 | 0.9 | 0.89 | 6.49 | 8.01 | 1050 | 0.5 | 0.7 | 10.7 | 26.87 |
| LAUDER | -45 | 1805 | 0.9 | 0.96 | 7.82 | 4.54 | 1132 | 0.9 | 0.96 | 7.91 | 4.58 | 1501 | 0.9 | 0.97 | 7.83 | 4.81 |
| ARRIVAL.HEIGHTS | -77.8 | 174 | 0.8 | 0.95 | 18.3 | 5.63 | 152 | 0.8 | 0.96 | 17.17 | 5.17 | 107 | 0.9 | 0.94 | 18.08 | 5.9 |
| Mean all stations | | | 0.9 | 0.91 | 7.62 | 5.95 | | 0.9 | 0.9 | 7.8 | 5.97 | | 0.9 | 0.91 | 7.65 | 6.55 |