# Peer review of "Validation of Methane and Carbon Monoxide from Sentinel-5 Precursor using TCCON and NDACC-IRWG stations"

_Atmospheric Measurement Techniques, 2021_

## Author Comment (AC1)

**Response to comments from the Editor (Dr. Andreas Richter)**
Black: Editor's comments; Blue: Authors' answers; Green: Changes in the manuscript

Dear Mahesh,

I'm pleased to accept your manuscript "Validation of Methane and Carbon Monoxide from Sentinel-5 Precursor using TCCON and NDACC-IRWG stations" for publication as AMT preprint. Please note that acceptance as AMT preprint does not guarantee later acceptance for AMT.

As you can see, both reviewers gave very positive assessments of your manuscript and I agree that this is a very relevant, detailed and well written study. I feel however that at least in parts, the article is written more like a report to ESA than like a scientific study. For example, the "mission requirements" have little relevance outside of ESA meetings and I would recommend to limit references to this quantity to a minimum. Also, the interesting point for readers is often not so much how large the differences are, but why they exist, and if S5P is to blame or the reference measurements or if it is rather linked to the details of the comparison. Finally, readers will also be interested in whether or not there is a way forward to further improve the S5P results.

Maybe you can keep these points in mind for the revisions after the public discussion.

Best regards,
Andreas

Dear Andreas,

We thank you for handling the manuscript and for providing the careful guidance.

We accept your comments and have included the suggested changes while doing the revision taking all comments made during the public discussion phase. In addition, we have added the validation results from a new NDACC type station at Porto Velho (Brazil), and added Corinne Vigouroux, who provided the data, as co-author. The co-located data cover few months between June 2019 and October 2019, which coincides with the peak of the CO emission seen for that year. This station is of high significance being the only station in the Amazonas and therefore we propose to include them in the paper. This will give confidence to the users of the S5P data. In addition, some station PIs have submitted missing data to complete the time series in our study period. We have updated the paper accordingly (incl. plots/tables/text). These modifications are small and does not change our previous conclusions.

Yours Sincerely
Mahesh Kumar Sha (on behalf of all authors)

**Response to comments from Referee 1**
**(see pages 2 – 12 of this document)**
Black: Referee's comments; Blue: Authors' answers; Green: Changes in the manuscript

We thank the referee for the review and for providing useful feedback, which we consider in the revised version of the paper.

Referee:
Manuscript "Validation of Methane and Carbon Monoxide from Sentinel-5 Precursor using TCCON and NDACC-IRWG stations" submitted to Atmos. Meas. Tech. by Sha et al. describes the validation of two atmospheric data products as generated operationally from the radiance spectra as measured by the TROPOMI instrument onboard of the Sentinel-5 Precursor (S5P) satellite. The manuscript covers an important topic appropriate for Atmos. Meas. Tech., contains new material and is well written. I therefore recommend publication after the comments listed below have been considered by the authors.

General comments
The manuscript is very (too?) long. It consists of 84 pages including 37 figures. I do not insist on shortening it but for readers (and reviewers) a split into two publications, one on methane and one on CO, would have been better, I think. Please consider this for a possible follow-on publication.

Authors' response:
Thanks for the suggestion. We will consider splitting the paper based on the species for potential follow-on publication(s).

Referee:
The paper covers several new and important aspects. Other validation papers addressing similar topics typically highlight the need (based on theoretical considerations) to apply one or more correction to be applied to the satellite and/or the ground-based data before they can be meaningfully compared. These corrections are primarily used to consider differences in vertical sensitivity (averaging kernels) and a priori profile assumptions in the retrieval. In addition, there are other corrections, especially corrections related to altitude, i.e., surface topography variations. Sha et al. also describe and apply these corrections but in addition they also present "direct comparison" results obtained without correction for vertical sensitivity and different a priori profiles (but still include an altitude correction). I like this very much for several reasons. One reason is that one needs to know how large a correction is to judge if the validation critically depends on (is dominated by) the correction or not. Another related reason is that many users will use the data "as is" and for this they probably need to know how good the data quality is without having applied more or less complex corrections. The information contained in this manuscript provides user with relevant information on this aspect, which is good.

Authors' response:
Thanks for this comment. This was exactly the reason why we presented the different cases and their influence at the respective ground-based reference sites.

Referee:
My interpretation of the results presented in this manuscript is that already a simple direct comparison gives meaningful validation results as the differences w.r.t. results obtained with correction are small. As shown in the paper (their Tables 4-7) the agreement between the satellite and the ground-based data are often (depending on metric and which data are compared) even better for the uncorrected data, i.e., for direct comparisons (e.g., Table 4 where it is shown that all three parameters (correlation coefficient, mean bias and standard deviation of the bias) are better for the direct comparison (see "Mean all stations"). This is not what one would expect for a meaningful correction. How can it be that a correction leads to an agreement between two data sets that is worse compared to a direct comparison without any correction? That this is the case (as shown in the paper) is very likely related to the fact that the correction is so small (compared to other more relevant effects) that the correction is essentially "not significant" (probably the error of the correction is on the same order or even larger than the correction itself). From the results shown in the paper I conclude that the effect of the correction is "plus/minus" in the sense that for certain aspects (or validation sites) agreement gets better but for other aspects agreement gets worse so that the overall effect is essentially zero. Do the authors agree with this interpretation of their results? Or in other words, do the authors agree with my conclusion that a meaningful validation of the S5P methane and CO products by comparisons with TCCON and NDACC is also possible via simple direct comparison, i.e., that a meaningful validation does not require to correct the data for altitude sensitivity and/or different a priori profiles? Please comment on this in the conclusion section of the manuscript as I think that statements related to this "correction aspect" would make the paper even more useful.

Authors' response:
The trace gas products from both satellite and ground-based remote sensing measurements provide the best estimate of the atmospheric state via a retrieval of the measured radiance spectra. However, there are several interfering parameters that are different for the two cases, which result in the respective uncertainties in their evaluation and comparison. Both methods, i.e., a direct comparison of satellite and ground-based reference data as well as a comparison with some corrections applied to one or the other data set, have their own advantages. In case of a direct comparison, we can get an estimate of the magnitude of differences due to some of the interfering parameters. In the latter case, we try to align the satellite and ground-based products as much as possible and then check for the differences.

The magnitude of the smoothing error or a priori alignment correction depends on the averaging kernels and the difference between the a priori profiles and the true profiles. In our a priori alignment case, we have used satellite prior as the common prior for comparison against both TCCON and NDACC data. For the methane comparison at the TCCON sites, this correction is small (< 0.3%) and a latitude dependence was observed. However, for the carbon monoxide

comparison at the TCCON sites the correction is quite significant and up to few percentages at high polluted sites and also sites located in the Southern Hemisphere. In the paper by Zhou et al. (2019), the influence of smoothing error when comparing TCCON and NDACC XCO data at six stations can be seen. The results with the adjusted TCCON and NDACC XCO data products, by adjusting the products towards a common optimal a priori profile, were consistent between the three North Hemispheric and three South Hemispheric stations.

In conclusion, we can state that it depends on the application if a direct comparison or comparison with correction should be applied. However, the latter case of applying the different corrections has the advantage that some of the interfering parameters can be aligned.

We have modified our previous statement in the conclusions section and added a line: "The comparison with a priori alignment and taking smoothing effects into account is recommended as the preferred method. However, the direct comparison of the satellite and reference data is useful to see the influence of the averaging kernel and a priori difference to the true profile."

Referee:
The Abstract contains this sentence starting at line 6: "In this paper, we present for the first time the S5P CH4 and CO validation results …". The sentence is long and lists a number of aspects so that it is not entire clear what is meant with "for the first time". Please modify this sentence to clarify if this statement only refers to a subset (or only one) or the many aspects listed. In any case this is not the first paper that reports on the validation of S5P CH4 and CO. The planned publication does not cite peer-reviewed publications relevant for the context of this publication. This is a major shortcoming and not easy to understand taking into account that many of the listed co-authors should be aware of relevant publications which are not cited (e.g., due to co-authorship). It needs to be mentioned that there is at least one other algorithm and corresponding S5P methane and CO data set, which is published, available, validated using TCCON and used to address important scientific applications, namely the S5P WFMD methane and CO data products generated by University of Bremen. To address this at least the following papers need to be cited (see References): Schneising et al., 2019, 2020a, 2020, Vellalassery et al., 2021.

Authors' response:
We have modified the sentence by adding that we do only validation of the S5P operational CH$_4$ and CO products against all TCCON and NDACC-IRWG stations. This information was present in the introduction but was missing in the abstract. Thanks for pointing this out.

Our focus was to validate the operational S5P CH$_4$ and CO products. However, we completely agree with the referee and have added a paragraph describing the TROPOMI/WFMD CH$_4$ and CO products in the Data section describing – S5P Methane and Carbon monoxide data sets. All references as suggested by the referee have been added.

"In addition to the operational S5P $CH_4$ and CO products, a scientific version of the products (TROPOMI/WFMD) using the Weighting Function Modified Differential Optical Absorption Spectroscopy (WFM-DOAS) has been developed by the University of Bremen (http://www.iup.uni-bremen.de/carbon_ghg/products/tropomi_wfmd/, last access 1 June 2021). The details of the TROPOMI/WFMD $CH_4$ and CO products, their validation against reference ground-based measurements and operational TROPOMI products, as well as use cases of the products to address important scientific applications can be found in (Schneising et al., 2019, 2020a, b; Vellalassery et al., 2021)."

Referee:
Because there is not only one S5P methane and CO product please add in the abstract which products (including version number) are validated in this publication.

Authors' response:
Done
"… the S5P operational $CH_4$ and CO products validation results (covering a period from November 2017 to September 2020, see Table 1 for version details) using global …"

Referee:
In this context also the paper of Lorente et al., 2021, is relevant. This paper is mentioned in the Sha et al. manuscript, which is good. Lorente et al., 2021, also report about a bias corrected S5P methane data product and shows detailed validation results. Is this Sha et al. manuscript using a methane data product with the same bias correction applied as also used by Lorente et al., 2021? If not, what is the difference between these products? Please add a short explanation of this so that it is clear for the readers what the relation between the two S5P bias corrected methane products is.

Authors' response:
We have used only the operational S5P $CH_4$ products in our study. In section 2.1 we stated "We provide a brief summary of the $CH_4$ bias correction here and the details of the bias correction can be found in section 5.6 of the Algorithm Theoretical Baseline Document (ATBD) for S5P methane retrieval (Hasekamp et al., 2019)."
The paper of Lorente et al., 2021 is discussing improvements that have been implemented to their RemoTec-S5P algorithm. This is a scientific product where all improvements are tested and then implemented in the operational processing chain.

We have modified our previous statement:
"Lorente et al. (2021) while analysing the improvements of their scientific S5P $XCH_4$ product found similar seasonality in the bias at the high latitude sites of Sodankylä and East Trout Lake and indicated correlations of high bias during spring time with the presence of snow (low surface albedo in the SWIR but high surface albedo in the NIR)."

Specific comments

Referee:
Abstract, page 2, line 20: Statement "We found that the required bias …". There is no "required bias" but only a requirement that the bias should be less than a certain value. Please modify this sentence.

Authors' response:
Done
"We found that the S5P carbon monoxide data over all surfaces for the recommended quality filtering in general fulfils the mission requirements of a bias (systematic error) less than 15% and a random error of <10%."

Authors' response:
Similar change is also done for the methane statement page 2 line 11.
"We found that the S5P standard and bias-corrected methane data over land surface for the recommended quality filtering fulfils the mission requirements of a bias (systematic error) less than 1.5% and a random error less than 1 %."

Referee:
Abstract, page 3, line 33 following: Sentence "The validation results for the clear-sky and cloud cases of S5P pixels are comparable to the validation results including all pixels …". This may be true for the validation results but is this really a statement about the quality of the S5P CO product for cloudy conditions taking into account that the ground-based data are limited to cloud free conditions (and the fact that S5P cannot look through clouds)?

Authors' response:
Following the product readme file for Carbon Monoxide, we have selected all pixels with QA_value of > 0.5 in our standard validation runs. This covers the clear-sky and clear-sky like observations, as well as the mid-level clouds. The latter also provides valid pixels over oceans (Borsdorff et al., 2018). In our validation runs we therefore compared the S5P results individually for the mid-level cloudy conditions (cloud optical thickness ≥ 0.5 & cloud height <5000 m, over land and ocean) and clear-sky conditions (cloud optical thickness <0.5 & cloud height <500 m, over land) in order to verify the co-incidence of the S5P pixels with the reference ground-based measurements. Note that the line-of-sight of the ground-based FTIR instruments is not necessarily the same as that of the satellite measurements. This plays a role in the comparison. Therefore, we investigated the influence of the cloudy and clear-sky S5P pixels at the reference ground-based sites. Our conclusion is based on mostly the background case. However, in case of fire events leading to large plumes we do observe a high scatter and bias w.r.t. ground-based stations, see example case for the Australian fire event discussed in section 5.1 of our discussion paper (page 17 line 498). We have therefore mentioned in the conclusion section "The clear-sky or cloud cases are however useful for certain applications."

We have modified our previous statement:
"The validation results for the clear-sky and cloud cases of S5P pixels are in general comparable to the validation results including all pixels with recommended quality filtering."

Referee:
Page 6, line 154 following: Concerning the 7% scaling factor and TCCON CO data use. I understand that the TCCON CO data used here are the products publicly available from the TCCON archive. Are these data already scaled and if yes how has the scaling factor be removed (e.g., is it a constant or does the product also contains the unscaled values?)? Please add this information on the content and use of the input data.

Authors' response:
Thanks for pointing this out. Yes, the publicly available TCCON data is already scaled. We have added the information about the creation of the unscaled XCO data in the paper:
"The unscaled XCO was calculated following Eq. 2 of Wunch et al. (2015), where the TCCON data without the scaling to the WMO scale were obtained from the site PIs."

The latter statement about the data availability is also included in the "*Data availability*" section.
"The FTIR TCCON data without the scaling to the WMO scale were obtained from the site PIs."

Referee:
Page 8, beginning of Sect. 3: The S5P operational CO product needs to be (has been) transformed to another product before it can be compared with TCCON. The transformation depends on space and time and is large and not simply a transformation between physical units. It is unclear for me if a meaningful validation is possible under these circumstances. At least it needs to be clearly mentioned that a direct comparison (validation) is not possible as the S5P and TCCON products differ significantly.

Authors' response:
We have modified our previous statement and added the information as suggested by the referee:
"S5P provides the total column density of CO, which can be directly validated against the NDACC CO total column density product. However, we need to calculate the corresponding XCO values in order to compare to the TCCON XCO products. The S5P XCO is calculated by taking the ratio of the total column of CO ($TC_{CO}$) divided by the total column of the dry air ($TC_{dry,air}$) (following Eq. 1 in Deutscher et al. (2010))."

Referee:
Page 8, line 197: Sentence "Ps and TCH2O are taken from the S5P files". Are these two quantities retrieved parameters or where do they come from? How accurate are they?

Authors' response:
Surface pressure (Ps) is an input parameter in the retrieval that is obtained from ECMWF (European Centre for Medium-Range Weather Forecasts) on SWIR pixel level, which is an estimate of the representative ground pressure for the observed ground scene and corrected based on the altitude difference between ECMWF and the surface elevation from a high resolution Digital Elevation Map (DEM). Following the ATBD document (Landgraf et al., 2018) a priori knowledge of the surface pressure within 1% is reasonable and they expect no critical performance issues due to uncertainties in the ground pressure.
TCH2O is a parameter calculated with meteorology from ECMWF. The ATBD document provides details on the sensitivity studies to the vertical distribution of water vapour.

Referee:
Page 9, line 230 following: I do not understand the colocation method. It is mentioned that an "effective location" of the FTIR sites is used depending on line-of-sight. Is this effective location a fix point on the Earth surface for a given satellite overpass? How is this location determined taking into account that the FTIR observations cover a certain period and there may be gaps at certain times due to clouds etc.? If not what does "a radius of 100 km" mean? What exactly are the "Co-located pairs"? Is this an average of several FTIR observations and an average of several S5P retrievals? If it corresponds to an average of several S5P retrievals (as mentioned in the paper) than the standard deviation of this difference cannot be directly related to the required random error or precision (as done in the paper, see line 253 following) as this requirement refers to the random error of single ground pixel retrievals but not to averages. Please clarify.

Authors' response:
The ground-based FTIR measures direct sunlight. The measured slanted airmass thus varies throughout the day. We determine the line-of-sight based on the instrument's location and the time of the measurement. The effective location is a point on this line-of-sight corresponding to an altitude of 5 km (free troposphere). The effective location thus changes from measurement to measurement and reflects the location of the probed airmass by the FTIR measurement more accurately.
A radius of 100 km means that TROPOMI pixels are co-located to an FTIR measurement if the pixel lies within a radius of 100 km around the effective location of the FTIR measurement (geodesic distance on the earth modelled as a sphere with radius r=(2*a_wgs84+b_wgs84)/3).
A co-located pair is a pair (FTIR measurement, SAT pixel) that satisfies the co-location conditions.
Given an FTIR measurement X, we consider the list of co-located pairs (X,P=SAT pixel) where P can vary. For each co-located pair we may apply Rodgers a priori alignment and or smoothing of the FTIR measurement with the SAT pixel column AVK and hence each co-located pair (X,P) is extended with a modified pair (X',P') depending on the validation settings. Averaging means that we average the modified SAT pixels and the modified FTIR measurement. In this method the random uncertainty on the pixels in the average is reduced by a factor √n (n=number of co-located pixels). The random uncertainty on the modified FTIR measurements X' is not reduced (as it is not independent since they originate from a single FTIR measurement).

We agree with the reviewer that alternative methods may be better suited to estimate this random uncertainty for a single pixel. The random uncertainty of the FTIR, which is present in the estimation using the std of the relative difference and therefore reflects the combined uncertainty FTIR and SAT, in which the SAT uncertainty is reduced by √n. Another method could be to use the "closest pixel" criterion without a priori alignment nor using the SAT AVK (see also the MPC validation server). Here the std of the relative differences is slightly above 1%, but this includes the FTIR uncertainty and the additional smoothing uncertainty due to a difference in prior's (see Rodgers). We are therefore quite confident that the S5P satisfies the mission requirements.

Referee:
Page 17, lime 515: Sentence "This result confirms the previously reported studies". What exactly is confirmed here? The exact value or only that it is negative? Please clarify.

Authors' response:
Done
"The result confirms the previously reported studies (Kiel et al., 2016; Sha et al., 2018b; Zhou et al., 2019) showing that the correction factor to tie the TCCON XCO data to WMO in situ scale is large and that TCCON XCO data is smaller than the uncorrected XCO data by about 7%."

Referee:
Page 24, line 758: Sentence starting with "We found that the systematic difference between the S5P ...":  The listed uncertainties of, for example, +/-0.57% for the bias corrected data, are not well justified, I think. These numbers are computed as mean values of the standard deviations of (averaged) S5P retrievals minus TCCON data per site. This uncertainty is more related to random errors and not so much to systematic errors. I think that the standard deviation of the biases as obtained for the various TCCON sites is more appropriate. I therefore recommend to add to all tables where the last row lists mean values ("Mean all stations") another row with "Standard deviation all stations" and to use these values as uncertainties.

Authors' response:
We have added a row showing the standard deviation of all stations to all tables (Tables 4 – 9). This standard deviation has been reported in the conclusion section along with the mean to give an indication of the overall performance for all stations in the network.

Referee:
Page 39: Fig. 1: I suggest to improve the x-axis annotation. If I understand correctly than for each co-location pair the relative difference is computed as (SAT-GB)/GB*100, i.e., as percentage difference. On the left of Fig. 1 then mean of this difference is shown (for each site) and on the fight the standard deviation of this difference. So "Mean difference [%]" and "Standard deviation of the difference [%]" would be clearer, I think. The annotation for the data in the middle is also potentially misleading as the unit percent may suggest that a percentage difference is shown but if I understand correctly the data show absolute differences of two

numbers, where each number has unit percent. Perhaps one can clarify this in the figure caption.

Authors' response:
Done
The x-axis annotations are "Mean relative bias [%], "Δ mean relative bias [%]", and "Std of the relative bias [%]". We have added the formula for relative bias (SAT-GB)/GB in the caption and included an explanation of how the Δ mean relative bias have been calculated.
We have made these changes to all figures showing bar charts.

Referee:
Page 41, caption Fig. 3: Please add that the time resolution of the data shown here is weekly.

Authors' response:
We have added the following sentence "The time resolution of the data shown here is weekly." to all Mosaic plots in the paper.

Typos

Referee:
No typos have been identified.

[revised manuscript text omitted]

Composition Change measured in Karlsruhe, Atmos. Meas. Tech., 9, 2223–2239, https://doi.org/10.5194/amt-9-2223-2016, 2016.

Sha, M. K., Langerock, B., De Mazière, M., Dils, B., Feist, D. G., Sussmann, R., Hase, F., Schneider, M., Blumenstock, T., Notholt, J., Warneke, T., Petri, C., Kivi, R., Te, Y., Wennberg, P. O., Wunch, D., Iraci, L. T., Strong, K., Griffith, D. W. T., Deutscher, N. M., Velazco, V. A., Morino, I., Ohyama, H., Uchino, O., Shiomi, K., Goo, T. Y., Pollard, D. F., Borsdorff, T., Hu, H., Hasekamp, O. P., Landgraf, J., Roehl, C. M., Kiel, M., Toon, G., TCCON team, and NDACC team: VDAF CO validation results & ESA AO # 28603 project TCCON4S5P first CO validation results using TCCON and NDACC data, in: S5P First products release workshop, ESA-Esrin, Frascati (Rome), Italy, 25 – 26 June, https://nikal.eventsair.com/QuickEventWebsitePortal/sentinel-5p-first-product-release-workshop/sentinel-5p, 2018.

---

## Author Comment (AC2)

**Response to comments from Referee 2**

Black: Referee's comments; Blue: Authors' answers; Green: Changes in the manuscript

We thank the referee for the review and for providing useful feedback, which we consider in the revised version of the paper.

**Referee:**

A. General Comments

The advantage of the TROPOMI measurement is its capability to cover entire globe in a single day with higher spatial resolution. Readers are interested in its validation for the data with large satellite zenith angles and how accurate the fast L2 retrieval algorithm is. The present manuscript looks like a technical report. The research paper must be concise and needs analysis for root causes of bias. The manuscript includes many redundant portions, which must be shortened. The abstract and the conclusion are also too long. Major revision is needed.

**Authors' response:**

Our intention of including S5P  $CH_4$  and CO validation results in one paper was to make use of the common description of the reference data sets and validation techniques description. We have provided possible reasons where large deviations in bias are observed.

The trace gas products from both satellite and ground-based remote sensing measurements provide the best estimate of the atmospheric state via a retrieval of the measured radiance spectra. However, there are several interfering parameters that are different for the two cases, which result in the respective uncertainties in their evaluation and comparison. Both methods, i.e., a direct comparison of satellite and ground-based reference data as well as a comparison with some corrections applied to one or the other data set, have their own advantages. In case of a direct comparison, we can get an estimate of the magnitude of differences due to some of the interfering parameters. In the latter case, we try to align the satellite and ground-based products as much as possible and then check for the differences. We believe, and also as pointed by Referee 1, that this information is relevant for the users of TROPOMI CH4 and CO data and therefore we find it useful to include these different cases in the paper.

As suggested by the referee, we have removed one sub-plot from each of Fig. 8 and Fig. 16, both-sub plots from Fig.17 and Fig. 36 and adapted the discussions in the main text accordingly to reduce the length of the paper.

**Referee:**

I have following suggestions.

(1) Match up condition

As TORPOMI has higher sampling density and spatial resolution, stricter match up conditions can be applied than for existing instruments such as GOSAT and OCO-2. 100 km or 50 km is too long for the sites located near urban area such as Saga.

**Authors' response:**

We agree with the reviewer that the co-location criteria for TROPOMI can be stricter as compared to the GOSAT and OCO-2 validation studies.

We found typical examples of co-location criteria used for GOSAT and OCO-2 validation studies: Noël et al. (2021) used a maximum spatial distance of satellite measurements from TCCON station of 500 km and maximum time difference of 2 h for GOSAT and GOSAT-2 validation. Parker et al. (2020) used all GOSAT soundings within  $\pm$  5° of a TCCON site for GOSAT XCH4 data validation. The co-location criterion used by Wunch et al. (2017) and O'Dell et al. (2018) for OCO-2 validation against TCCON requires the OCO-2 footprint to be within 2.5° latitude and 5.0° longitude of the TCCON station and the observations that occurred within 2 h of each other.

We have tested several co-location criteria and found that for CO, a co-location radius of 50 km gives robust results for the global stations in the networks (Sha et al., 2018; see extracted corresponding plot below - Figure 1).

However, when applying the same 50 km co-location radius we do not get robust statistics for CH4. This is due to the fewer TROPOMI CH4 pixels available, in comparison to CO, due to strict pixel filtering (clouds, SZA, ... ) and operational CH4 pixels currently available only over land. As a result, we have relaxed the co-location radius to 100 km for CH4 validation study.

Figure 1: XCO bias for Lauder as a function of the coincidence criterion of radius. Solid shapes are for coincidence > 4 S5P pixels and empty shapes are for >2 S5P pixels.

We agree with the reviewer that a stricter match up condition based on site characteristic might be useful. However, our goal was to use the same strict co-location criterion, which is valid for all stations in the network. This is also the reason why we have implemented a stricter

cone selection criterion where we follow the ground-based FTIR line-of-sight with a defined opening angle of the cone. This criterion is especially effective in pixel selection for stations located close to regions with high emission sources, where there are possible scenarios when the ground-based FTIR line-of-sight is not covering all pixels observed by the satellite using the circular co-location criterion (an example is given in Figure 26 of our discussion paper).

**Referee:**

**(2) Summary table**

There are several numbers of systematic errors in the abstract, the main text, and the conclusion. The summary table with numbers and conditions will help readers' understanding.

**Authors' response:**

Taking into account the comments of Referee 1, we have added the Standard deviation of all stations values along with the Mean of all stations to each validation settings in separate rows at the end of each table. In view of not increasing the length of the paper, we hope that these two columns at the end of each table will help the readers' understanding and to get a quick overview of the results for the respective settings.

**Referee:**

(3) NDAC

There are already plenty of match up data with TCCON. Explanation why NDAC data is additionally need, is required in more detail. Do authors need more data at high latitude stations with large solar angles?

**Authors' response:**

Although TCCON and the FTIR NDACC instruments use the same high-resolution FTIR spectrometer, the instruments are configured differently and the retrieval method is different. TCCON measures dry air averaged columns while NDACC measures vertical profiles with limited vertical resolution (typically 2 to 3 partial column are distinguished). Both networks have proven to be valuable in the comparison with satellite data and differences between the networks allow to better understand the subtleties in the comparison. Besides this, making use of the measurements from both networks allows a better coverage of different land types, geographic locations, atmospheric conditions, ... .

**Referee:**

(4) Geometry dependency

Authors mentioned solar zenith angle dependency. TROPOMI has wide cross-track coverage. Is there also viewing angle dependency? Is the bias due to forward calculation error by the radiative transfer model used in the L2 retrieval?

**Authors' response:**

The relative bias for CO plotted as a function of the viewing zenith angle (VZA) is shown in the figure below (Figure 2). We do not observe any significant dependence of the bias on the VZA.

---

## Author Response (AR2)

**Response to comments from the Editor (Dr. Andreas Richter)**
Black: Editor's comments; Blue: Authors' answers; Green: Changes in the manuscript

Dear Mahesh Kumar Sha,

I'm pleased to accept your manuscript "Validation of Methane and Carbon Monoxide from Sentinel-5 Precursor using TCCON and NDACC-IRWG stations" for publication in AMT subject to minor revisions as outlined below.

As you have seen, the two reviewers are satisfied with your changes even though you did not shorten the manuscript as suggested. I'm happy to see even more data and in particular from an important new station and I'm also satisfied with the slight change in tone. I still think that discussing more the why than the what would have made the manuscript even more interesting but it is already a relevant and quite big study as it is.

The only change I wold really like to see is a shortening of the abstract by roughly a factor of two - this will really help to give readers a quicker assessment of what are the main points of your study. It would also be good to shorten the summary at the end by removing duplications but the abstract is the most important point for me.

Best regards,

Andreas Richter

Dear Andreas,

Thank you for your comments. We have modified the abstract, which removes some of the duplications and therefore we decided to leave the conclusion chapter as it is. Please find attached the updated version of the manuscript for your evaluation (in track changes mode).

Yours Sincerely
Mahesh